# CREsted: modeling genomic and synthetic cell-type-specific enhancers across tissues and species

Niklas Kempynck [1,2,3,8], Seppe De Winter [1,2,3,8], Casper H. Blaauw [1,2,3,4], Vasileios Konstantakos [1,2,3], Eren Can Ekşi [1,2,3], Sam Dieltiens [1,2,3], Darina Abaffyová [1,2,3], Valérie Bercier [2,5], Ibrahim I. Taskiran [1,2,3,7], Gert Hulselmans [1,2,3,6], Katina Spanier [1,2,3], Valerie Christiaens [1,2,3,6], Ludo Van Den Bosch [2,5], Lukas Mahieu [1,2,3,6] & Stein Aerts [1,2,3,6] ✉

Sequence-based deep learning models have become the state of the art for analyzing the genomic regulatory code. Particularly for enhancers, these models excel at deciphering sequence grammar that underlies their activity. To enable end-to-end enhancer modeling and design, we developed a software package called CREsted (*cis*-regulatory element sequence training, explanation and design). It combines preprocessing and analysis of single-cell assay for transposase-accessible chromatin using sequencing data, modeling chromatin accessibility from sequence, sequence design and downstream analysis to decipher enhancer grammar. We demonstrate CREsted's functionality on a mouse cortex and a human peripheral blood mononuclear cell dataset. Additionally, we use CREsted to compare mesenchymal-like cancer cell states between tumor types, and we investigate different fine-tuning strategies of genomic foundation models within CREsted. Finally, we train a model on a zebrafish development atlas and use this to design and in vivo validate cell-type-specific enhancers. For varying datasets, we demonstrate that CREsted facilitates efficient training and analyses, enabling scrutinization of the enhancer logic and design of synthetic enhancers across tissues and species.

*Cis*-regulatory elements (CREs), particularly enhancers, are the primary drivers of cell-type identity[1]. These elements function by integrating specific combinations of activator and repressor transcription factor binding sites (TFBSs); the interplay of their strength, copy number and relative arrangement constitutes the 'enhancer code'. This logic is inherently degenerate: a single transcription factor (TF) can recognize diverse TFBSs and different combinations of TFBS sets can drive identical functional outputs[2]. Deep neural networks

have emerged as powerful tools to model these complex rules[3–11]. Sequence-to-function models[12] take in genomic sequences and predict a variety of genomic assays, including TF binding[9], chromatin accessibility[4,6–8,13], enhancer activity[14] and gene expression[5,10,15]. Among those, single-cell assay for transposase-accessible chromatin using sequencing (scATAC-seq) has demonstrated particular utility, as differential region accessibility between cell types provides a strong, cell-type-specific indicator of enhancer function in heteroge-

[1]Laboratory of Computational Biology, VIB Center for AI and Computational Biology (VIB.AI), Leuven, Belgium. [2]VIB-KU Leuven Center for Brain and Disease Research, Leuven, Belgium. [3]Department of Human Genetics, KU Leuven, Leuven, Belgium. [4]Oncode Institute, Hubrecht Institute-KNAW (Royal Netherlands Academy of Arts and Sciences) and University Medical Center Utrecht, Utrecht, the Netherlands. [5]Department of Neurosciences, KU Leuven, Leuven, Belgium. [6]Aligning Science Across Parkinson's (ASAP) Collaborative Research Network, Chevy Chase, MD, USA. [7]Present address: Illumina Artificial Intelligence Laboratory, Illumina, Foster City, CA, USA. [8]These authors contributed equally: Niklas Kempynck, Seppe De Winter. ✉e-mail: stein.aerts@kuleuven.be

neous cell populations[16–24]. Therefore, scATAC-seq atlases provide an established foundation for sequence-to-function models aimed at decoding enhancers[24–26].

Software packages have been introduced with the aim of streamlining the process of data processing, model training and sequence design[8,27–29]. For example, Selene[30] and EUGENe[28] offer frameworks for various predictive tasks; ChromBPNet[8] and scPrinter[27] focus on predicting TF footprinting; and Ledidi[31] is a tool kit to design edits to biological sequences. Lastly, gReLU[29] covers many steps in sequence modeling, such as data processing, model training, variant effect prediction and model-guided sequence design. However, these frameworks are not tailored to model enhancer codes across cell types, nor have they been validated on large-scale and complex scATAC-seq atlases in different biological systems, and they often lack comprehensive cell-type-specific enhancer code analysis tools.

Here we present CREsted, a Python package compatible with 'scverse'[32] providing user-friendly deep learning modeling of scATAC-seq data combined with cell-type-specific enhancer code analyses. By examining various biological systems, such as the mouse motor cortex, human peripheral blood mononuclear cells (PBMCs), human cancer cell states and the developing zebrafish, we demonstrate that CREsted is versatile across species and tissues and capable of handling large atlases. We validate the identification of relevant TFBSs, highlight cross-species prediction capabilities and in vivo validate enhancer design methods.

## CREsted is a software package for efficient enhancer modeling and design

CREsted consists of four modules: data preprocessing, model training, cell-type-specific enhancer code interpretation and synthetic enhancer design (Fig. 1).

Preprocessing builds on the output of established scATAC-seq analysis pipelines[33–35] and has two modes: topic modeling[34,36] and pseudobulk peak aggregation. Topics are latent representations of matched probability distributions over peaks and cells. Pseudobulk peak aggregations are cell-type-specific counts over a set of consensus peaks (a set of peaks merged across cell types). When using topic modeling, either a regression or a classification model can be trained. The regression model directly predicts peak-topic probabilities, while for the classification model, these probabilities are first binarized[13,24–26]. When using pseudobulks, a scalar value per cell type is retrieved for each consensus peak. Standard counts-per-million (CPM) normalization introduces bias against cell types with higher peak counts. To correct this, we rescale CPM-normalized values using constitutive peaks with high absolute values and low variability (Gini index < mean + 1 standard deviation). This aligns baseline accessibility across cell types (Supplementary Fig. 1a), functioning as a min–max normalization analogous to ArchR's ReadsInTSS method[16,33].

We recommend a two-step training procedure by first training on all consensus peaks, then fine-tuning on cell-type-specific peaks. Cell-type-specific peaks can be defined both manually or, in the case of peak regression, by selecting regions with high variability based on their Gini index. Before training, regions are split into training, validation and test sets, either by splitting on chromosomes or by randomly dividing regions according to user-defined proportions.

For training a classification model, binary cross-entropy is used as a loss function. In regression modeling[16,26], we use the sum of the average cosine similarity and the log mean squared error (MSE) between the region predictions over all cell types and their target values. Optionally, both parts of the loss function can be scaled dynamically during training. Multiple architectures can be chosen, all of which are inspired by previously validated models[6,8,10,11] (Fig. 1). Additionally, the large-scale Enformer or Borzoi foundation models[10,15] can be used as pretrained

models or with transfer learning to predict chromatin accessibility of new cell types.

CREsted allows for nucleotide-level explanations using gradient-based methods[37] and in silico mutagenesis (ISM)[3,6], revealing nucleotides of high importance for the prediction of a given cell type. CREsted interfaces with tfmodisco-lite[38,39] and tangermeme[40] to identify cell-type-specific TFBSs[26] followed by matching to TF candidates through single-cell RNA-sequencing data (if available).

As a final module, CREsted provides an enhancer design toolkit using in silico evolution (ISE) or through optimal implantation of TFBS instances, as described by Taskiran et al.[41]. Additionally, we provide a new cost function that makes use of the L2 distance helping to ensure cell-type specificity, by designing cell-type-specific sequences while keeping enhancer activity low in other cell types.

## CREsted provides detailed insights into enhancer codes of mouse cortical cell types

We investigated a mouse motor cortex scATAC-seq dataset[42] that was recently used to benchmark in vivo enhancer activity predictions[16]. We obtained subclass-level pseudobulk chromatin accessibility tracks (Fig. 2a) and validated peak scaling by comparing cell-type-specific peak scalars to average peak heights of 3,985 promoters from mouse housekeeping genes[43] (Supplementary Fig. 1b–e and Supplementary Note 1).

Next, we trained a peak regression model, DeepBICCN2, by first training on 440,993 consensus peaks followed by fine-tuning on 73,326 cell-type-specific regions. On test regions from held-out chromosomes, we obtained Spearman ($\rho$) and Pearson ($r$) correlation coefficients of 0.79 and 0.82 between log-transformed predictions and peak heights, respectively (Fig. 2b). The correlation is always the highest with the target cell type (Fig. 2c). Cell types that have similar accessibility profiles, such as glutamatergic neurons (Supplementary Fig. 1f), also exhibit high prediction correlation. We compared the predictions of our base and fine-tuned models against a default (6 million parameters) and large (22 million parameters) model trained with the gReLU framework[29]. The predictions from CREsted were significantly ($P$ value < 0.001) more accurate both on all and on cell-type-specific test peaks compared to the gReLU models (Fig. 2d). We also assessed the performance of CREsted in various settings, finding that direct training on cell-type-specific peaks results in worse performance than fine-tuning and that both altering chromosomal splits and including non-peak regions does not affect the performance notably (Supplementary Figs. 1g,h, and 2a–d and Supplementary Note 2).

Encouraged by DeepBICCN2's strong performance, we further extended predictions to genomic loci. For example, we evaluated the *Chsy3* gene, which is specifically expressed in L6 corticothalamic cells (L6CT), for the L6CT class with a 100-bp step sliding window (Fig. 2e). We observed a high correlation ($r = 0.75$) between predicted and ground-truth accessibility tracks, indicating that DeepBICCN2 not only generalizes well to unseen peaks, but also to inaccessible regions. We then extended this concept to a locus of another species. For example, we have recently shown that the enhancer code of mouse and bird interneurons is strongly conserved[26]. We scored the chicken *UACA* gene (specifically expressed in Parvalbumin (Pvalb) interneurons) locus using the mouse Pvalb class, also resulting in a strong correlation ($r = 0.62$). This illustrates the possibility of scoring genomic loci in species without scATAC-seq data to identify and decode candidate enhancers.

Next, we used DeepBICCN2 to score 171 in vivo-validated cell-type-specific enhancers[17]. It classified these enhancers with an average precision of 0.77 and recall of 0.79 in a multi-label classification setting[16] (Supplementary Fig. 2e,f). We highlight three examples (AiE2428m, AiE2587m and AiE0391h; Supplementary Fig. 2g,h), with their contribution scores showing a variety of TFBSs (Fig. 2f). To link

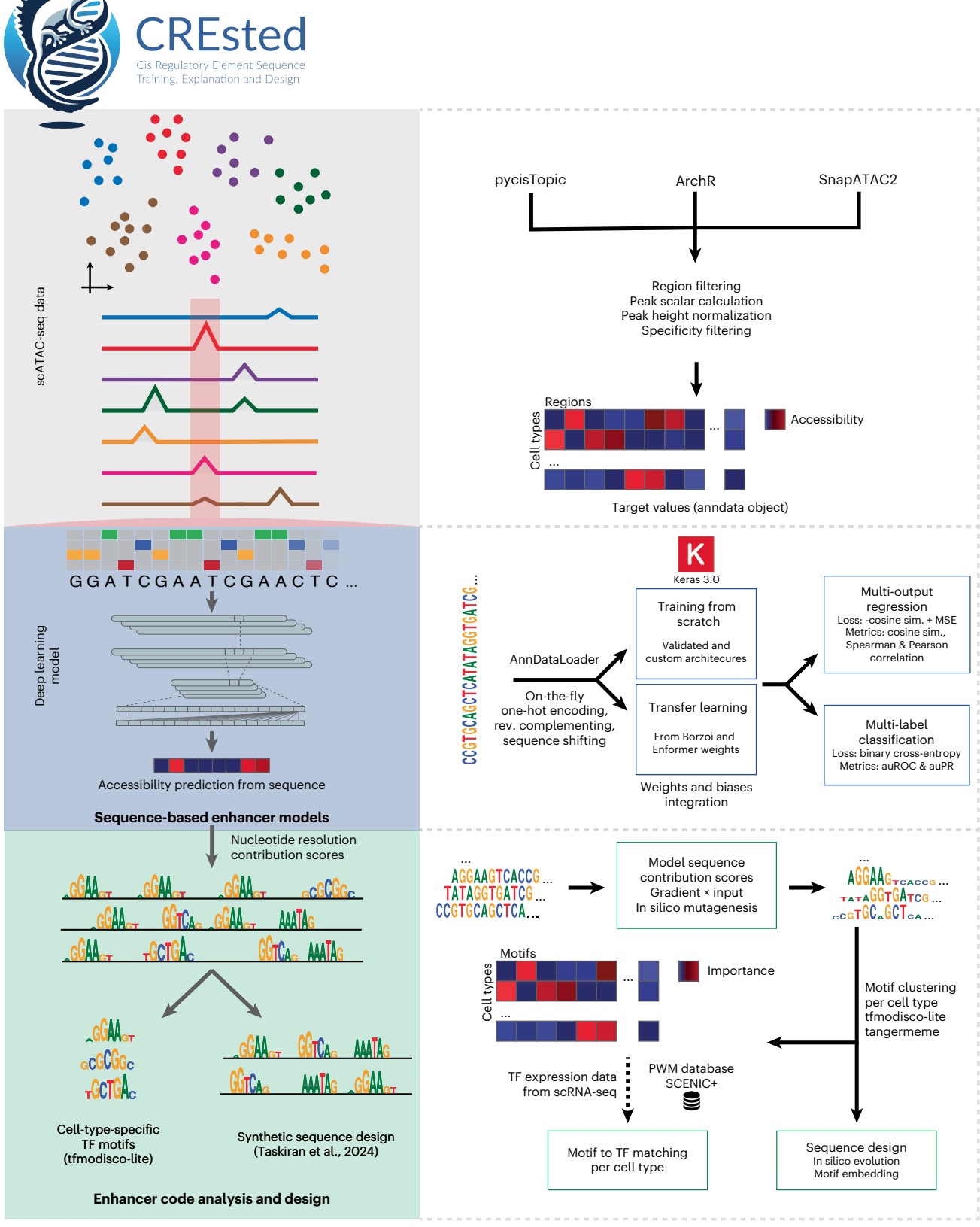

**Fig. 1 | CREsted package overview.** CREsted is a software package for training enhancer models, analyzing cell-type-specific enhancer codes and designing synthetic enhancer sequences. From preprocessed scATAC-seq data, we apply additional steps to obtain accessibility values per cell type. Those are used as target values to train sequence-based enhancer models that can be trained from scratch or use transfer learning from large-scale models. Both multi-output regression and multi-label classification are possible. From the trained models, we obtain insights into cell-type-specific enhancer codes by looking at nucleotide contribution scores to identify motifs and match them to TF candidates. Finally, CREsted allows for synthetic sequence design using the obtained enhancer code insights.

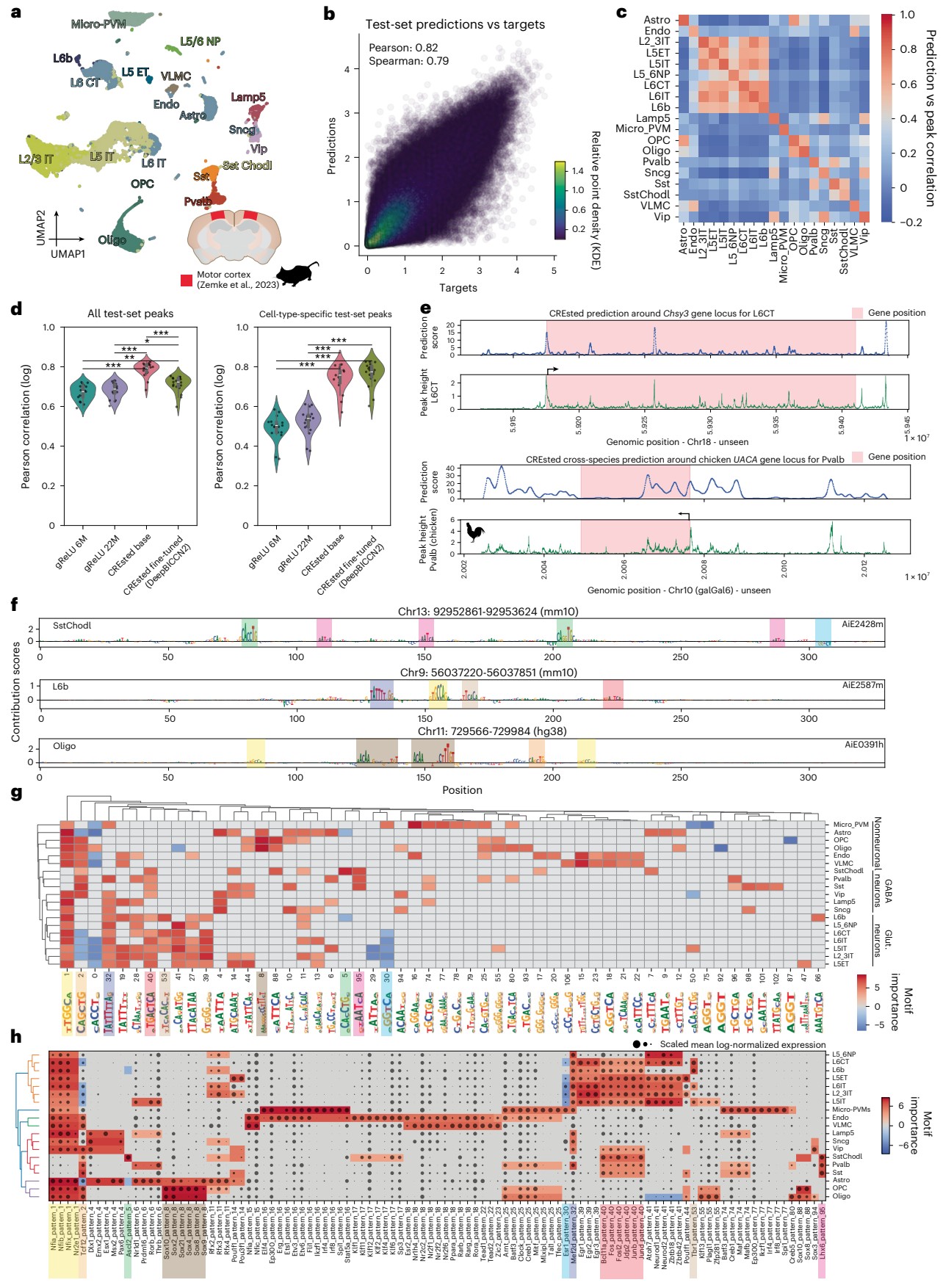

**Fig. 2 | CREsted applied to mouse motor cortex scATAC-seq data. a**, scATAC-seq uniform manifold approximation and projection (UMAP) of the Zemke et al.[42] mouse motor cortex data. Adapted with permission from ref. 26, AAAS. **b**, Scatterplot of log-transformed predicted peak heights and scATAC target peak heights for test-set regions over all cell types for nonzero peak values, generated with 'crested.pl.scatter.class_density'. KDE, kernel density estimation. **c**, Heat map of Pearson correlations for log-transformed cell-type-specific test-set nonzero peak heights and predictions, separated per cell type, generated with 'crested.pl.heatmap.correlations_predictions'. **d**, Comparison of Pearson correlation between log-transformed peaks and predictions per cell type ($n = 19$ classes) for gReLU and CREsted models, on all test-set consensus peaks (left) and cell-type-specific peaks (right). $P$ values were derived using a two-sided $t$-test and corrected using the Benjamini–Hochberg method (*$P < 0.05$, **$P < 0.01$, ***$P < 0.001$). Left: gReLU 6M–gReLU 22 M, $P = 0.33$; gReLU 6M–CREsted base, $P = 7.7 \times 10^{-11}$; gReLU 6M–CREsted fine-tuned, $P = 0.003$; gReLU 22M–CREsted base, $P = 3.9 \times 10^{-10}$; gReLU 22M–CREsted fine-tuned, $P = 0.02$; CREsted base–CREsted fine-tuned, $P = 1.4 \times 10^{-6}$. Right: gReLU 6M–gReLU 22 M, $P = 0.15$; gReLU 6M–CREsted base, $P = 1.1 \times 10^{-13}$; gReLU 6M–CREsted fine-tuned, $P = 7.6 \times 10^{-15}$; gReLU 22M–CREsted base, $P = 3.2 \times 10^{-12}$; gReLU 22M–CREsted fine-tuned, $P = 1.3 \times 10^{-13}$; CREsted base–CREsted fine-tuned, $P = 0.27$. **e**, Gene locus scoring with the fine-tuned CREsted model on a gene located in a held-out chromosome

for the L6CT class, compared to its scATAC track (top), and cross-species scoring on a chicken gene locus for the Pvalb class, compared to its chicken scATAC track[26] (bottom), generated with 'crested.pl.hist.locus_scoring'. **f**, Contribution scores of three in vivo-validated enhancers for their corresponding classes, generated with 'crested.pl.patterns.contribution_scores'. Colored boxes were added for different identified motifs. Groups of nonneuronal cell types, GABAergic (GABA) neurons and glutamatergic (glut.) neurons are highlighted. **g**, Clustermap of identified motifs in the top 2,000 regions per cell type. The color scale indicates motif importance, represented by the log-transformed pattern count. Negatively contributing motifs were given negative importance values. This plot was generated with 'crested.pl.patterns.clustermap_with_pwm_logos'. Colored boxes were added for matching motifs found in **f**. **h**, Clustermap of scaled mean log-normalized TF expression over cell types, indicated by dot size. TFs have matching binding sites with a motif in **g**. Motif importances for the matched patterns are indicated by color. This plot was generated with 'crested.pl.patterns.clustermap_tf_motif'. Colored boxes were added for matching motifs in **f** and **g**. For violin plots in **d**, the inner box plots show the median (center line) and the lower and upper quartiles (bottom and top hinges); whiskers extend from each hinge to the most extreme data point within 1.5 times the interquartile range. Individual points represent values for each cell type ($n = 19$). Micro-PVMs, microglia–perivascular macrophages; VLMC, vascular leptomeningeal cell.

those to TFs, we identified TFBS patterns on contribution scores for the 2,000 most cell-type-specific regions, clustered frequently occurring patterns based on motif similarity and counted instances of those pattern clusters in regions across cell types (Fig. 2g). The resulting cell-type clustering shows expected grouping of nonneuronal, glutamatergic and GABAergic cell types. Of note, we observe an E-box motif that not only positively influences interneurons, but also negatively impacts excitatory neuron predictions, potentially stemming from model leakage. Next, for each pattern cluster we identified annotated TFs[34] following a similar expression profile compared to the motif importance across cell types. We find previously observed[26] specific factors: TBR1 and NFI for deep-layer glutamatergic neurons; RFX3 and EGR for upper-layer glutamatergic neurons; SOX10 and CREB5 for oligodendrocytes; SPI1, IRF and MAFB for microglia–perivascular macrophages; and LHX6 and MAFB for medial ganglionic eminence interneurons. Interestingly, we identified an E-box motif with the consensus sequence CAGGTG that is unique to somatostatin-chondrolectin (SstChodl) cells. Mutating such instances in the SstChodl AiE2428m enhancer to the more canonical CAGCTG form in all interneurons, changes its predictions to be generally accessible in medial ganglionic eminence interneurons (Supplementary Fig. 2i,j). This highlights the high resolution of CREsted models, from finding global motifs representing cell-type-specific enhancer codes, to explaining cell-type-specific effects of single-nucleotide variations in TFBSs.

Here we illustrated a complete run-through of the CREsted pipeline, highlighting its capabilities of interpreting mouse cortex enhancer codes and its strong predictive capabilities at region and locus levels.

## A CREsted human PBMC model captures validated TFBSs

To illustrate the capability of CREsted models to also function in other tissues and species, we trained DeepPBMC, a human PBMC peak regression model, reaching an $r$ of 0.71 on cell-type-specific test peaks (Supplementary Fig. 3a–c). An embedding of cell-type-specific regions using DeepPBMC's prediction scores reveals clear separation per cell type (Fig. 3a and Supplementary Fig. 3d).

Next, we assessed two enhancers for which TFBSs were previously experimentally validated[44,45]: a *CD79A* enhancer in B cells and a *TCRα* enhancer in T cells. DeepPBMC could recover all TFBSs for both enhancers (Fig. 3b) but also identifies extra TFBSs such as for IRF8 and REL in the *CD79A* enhancer and an ETS-like TFBS for the *TCRα* enhancer that were also confirmed using Borzoi (Supplementary Fig. 3e,f). Next, we investigated the dendritic cell-specific interferon-β (*IFNB1*) enhanceosome. This enhancer consists of multiple overlapping TFBSs in a 50-bp window and has been structurally resolved together with all bound TFs[46,47]. DeepPBMC retrieves a large part of the enhanceosome's complexity, only missing TFBSs for p50, c-Jun and one of the four IRF TFBSs (Fig. 3c). Compared to Borzoi, DeepPBMC identifies substantially more (34%) important nucleotides (Supplementary Fig. 3g).

Using CREsted's pattern clustering analysis (Fig. 3d), we recovered motifs for previously established key cell-type-specific TFs, such as EBF1, PAX5 and POU2F2 in B cells[34], ETS1, RUNX1 and GATA3 in T cells[48–50] and CEBPA and SPI1 in monocytes[51,52]. In this context, we investigated whether model input size affects motif identification and found this parameter to have limited effects (Supplementary Fig. 4).

**Fig. 3 | A human PBMC CREsted model identifies functional TFBSs. a**, $t$-distributed stochastic neighbor embedding ($t$-SNE) of region model embeddings for the top 1,000 specific regions per PBMC type. The assigned class determines the coloring. **b**, Contribution scores of the *CD79A* (hg38 chr19: 41876056–41878170; top) and *TCRα* core (hg38 Chr14: 22555403–22557517; bottom) enhancers for the B cell and CD4+ T cell class, respectively. The center 200 bp of both enhancers is shown. **c**, Contribution scores for both strands of the *IFNB1* enhanceosome (hg38 Chr9: 21076963–21079077, zoomed to center 200 bp) for the dendritic cell class. The enhanceosome's location is highlighted with the black box. In **b** and **c**, colored boxes indicate found motifs. Validated TFBSs are underlined; proposed TFBSs are not. **d**, Clustermap of identified motifs in the top 1,000 regions per cell type from **a**. The color scale indicates motif importance, represented by the log-transformed pattern count. Negatively contributing motifs were given negative importance values. Motifs were manually annotated

with the results from **b** and **c** and motif databases. **e**, Precision–recall curve on comparing model-identified seqlets and ChIP–seq peaks in the top 1,000 peaks for the corresponding cell type. Thresholding is done on the average seqlet contribution score. Average precision (AP) and average recall (AR) over the thresholds are indicated in the legend. **f**, Bar plot of recall of identified seqlets inside UniBind sites in the top 1,000 peaks for the corresponding cell type. **g**, Average ChIP peak height of different sets of proposed TFBSs for CEBPA in the top 1,000 most specific CD14+ monocytes. **h**, Cross-species scoring on the mouse *Tcf3* gene locus in B cells before (blue) and after (red) EBF1 TFBS mutation (top), compared to mouse scATAC tracks before (green) and after (red) EBF1 degradation in mouse precursor B cells[58]. **i**, Contribution scores for two highlighted regions before binding site mutation in **h** for the B cell class. EBF1 sites are highlighted in green; other proposed PAX5 and MEF2 TFBSs are also highlighted.

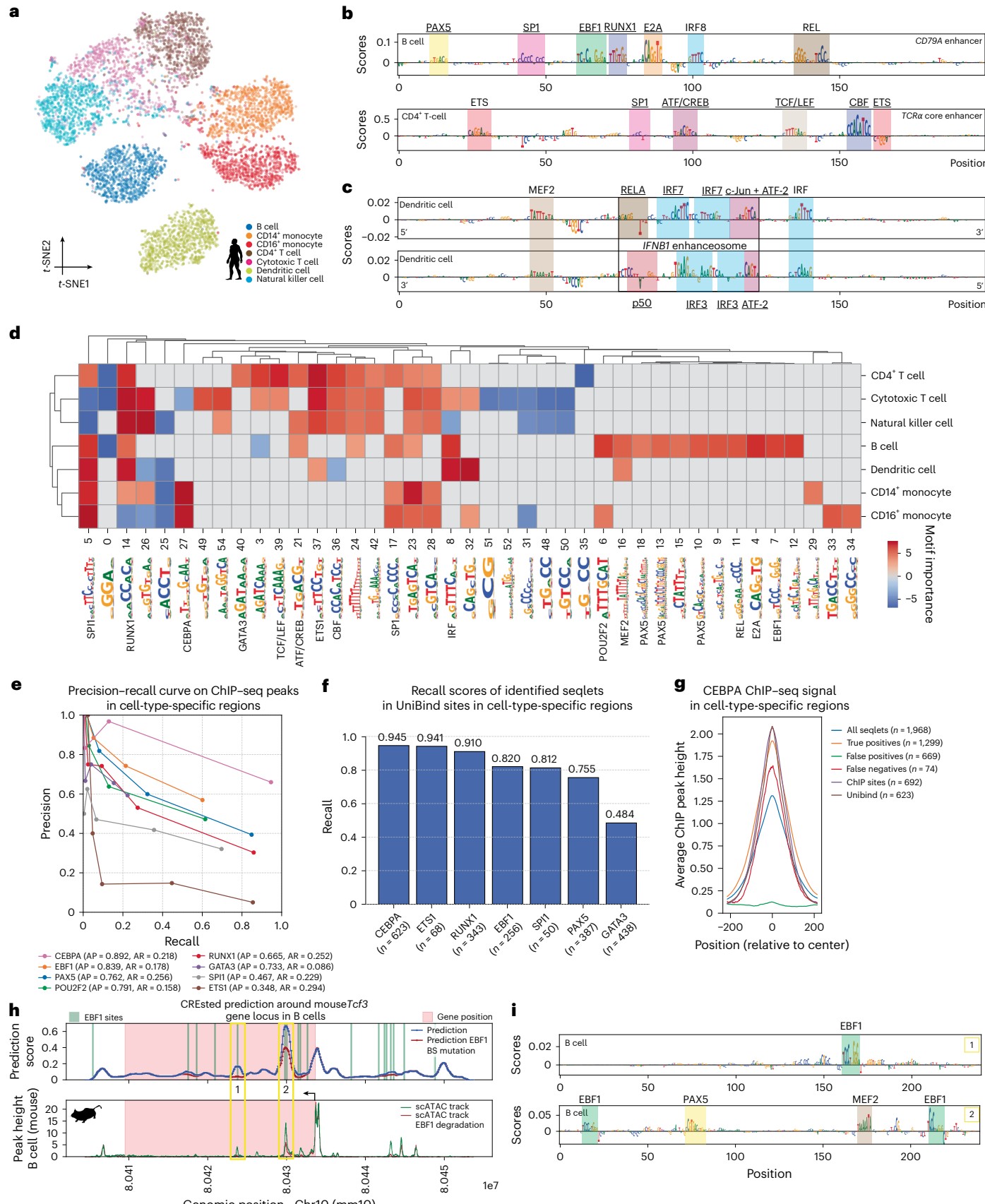

To validate the identified TFBSs, we used publicly available chromatin immunoprecipitation sequencing (ChIP–seq) data (EBF1, PAX5, POU2F2 in B cells, CEBPA and SPI1 in CD14+ monocytes, and RUNX1, ETS1 and GATA3 in CD4+ T cells[50–54]) and observed an overall high precision (average precision over all thresholds for all TFs of 0.75) that increased with prediction score (Fig. 3e) but a low recall. The latter could be attributed to ChIP–seq peaks with indirect TF binding[55]. For this reason, we next focused on directly bound ChIP–seq peaks as predicted by UniBind[56]. This resulted in a much stronger recall, with an exception for GATA3 (Fig. 3f and Supplementary Fig. 5a). Here we also compared gradient-based versus ISM-based nucleotide contributions, and observed that gradient-based methods are more appropriate for motif recovery (Supplementary Fig. 5b,c). The high concordance between predicted TF binding site and ChIP–seq signal can also be observed by aggregating ChIP–seq signal over all seqlets (Fig. 3g and Supplementary Fig. 5d,e). Finally, we compared the motifs identified through CREsted to those identified using classical motif enrichment analysis tools (pycisTarget[34] and pyChromVar[57]). CREsted identified similar motifs, but with overall higher precision and recall for identifying binding instances (Supplementary Fig. 6).

The functional relevance of CREsted-predicted TFBSs is further confirmed by simulating the effect of TF degradation. In particular, we assessed the effect of mutating EBF1 instances in the mouse *Tcf3* locus on B cell chromatin accessibility compared to the actual chromatin accessibility profile after EBF1 protein degradation[58]. For both control and treated cells, we observed a high correlation between the predicted and actual chromatin accessibility profile (*r* of 0.55 and 0.60, respectively; Fig. 3h). Furthermore, the two regions that are most strongly influenced by EBF1 protein degradation indeed contain EBF1 TFBSs according to DeepPBMC (Fig. 3i).

These results strongly support that TFBSs identified by CREsted are biologically relevant, both for known enhancers, and globally through the identification of TFBS instances per cell type.

## CREsted identifies high similarity between MES enhancer codes in cancer

Cancer cell states have been enigmatic to robustly compare between patient biopsy samples, due to strong patient-specific epigenomes and transcriptomes[59,60]. The same states can recur across different tumor types, such as the mesenchymal-like (MES) cell states reported in melanoma, glioblastoma (GBM) and others[61,62,63]. Previously, we already modeled MES-like enhancers in melanoma using DeepMEL[13] and DeepMEL2 (ref. 64). However, MES-like regulatory programs have not yet been compared across cancer types at the enhancer-code level. We hypothesized that CREsted models could make an abstraction of patient-specific genomic aberrations and enable a direct comparison of regulatory programs active across cancer types.

For this purpose, we compared the melanoma and GBM MES-like states using CREsted. Furthermore, cancer is often modeled using cancer cell lines, which do not necessarily reflect in vivo cancer states. To address this, we compared the enhancer code of GBM cell lines and patient biopsy samples.

We trained a peak regression model, DeepCCL, on melanoma cell lines (two MES-like, MM029 and MM099, and one melanocytic-like, MM001)[34], GBM cell lines (the MES-like A172 and M059J lines, and the pro-neural LN229 line[65,66]) and other cancer cell lines (HepG2 and GM12878) used as a negative (non-MES-like) control. Deep-CCL groups MES-like states across cancer types and can partially distinguish between them, although shared regulatory programs remain evident (Fig. 4b and Supplementary Fig. 7a,b). We confirmed these findings using an ensemble of ChromBPNet models, although this model had a lower predictive and explanatory performance (Fig. 4c and Supplementary Fig. 7a–c). Next, we compared the enhancer codes learned for both MES-like states. For example, a candidate *AXL* enhancer[64] relied on AP-1, TEAD and ZEB TFBSs in both contexts (Fig. 4d and Supplementary Fig. 7d). Globally, we observed shared contributions from AP-1, TEAD, RUNX, NFI and ATF/CREB (Fig. 4e).

Next, to investigate whether this MES-like program observed in cell lines is also active in tumor biopsy samples, we reused a scATAC-seq dataset of human gliomas[67]. To avoid any biases in cell-type annotation, we identified topics using pycisTopic[36], yielding both patient-specific and shared topics across patients (Fig. 4f and Supplementary Fig. 8a). Next, we trained a CREsted topic classification model, DeepGlioma (Fig. 4a and Supplementary Fig. 8b). By comparing cell line accessibility and predictions from DeepCCL with topic accessibility, we identified a set of potential MES-like biopsy topics (topics, 8, 21, 25 and 20) and three candidate oligodendrocyte progenitor cell/neural progenitor cell (OPC/NPC)-like topics (topics 14, 18 and 19; Supplementary Fig. 8c). To compare the enhancer codes obtained from cell line (DeepCCL) and patient biopsy (DeepGlioma) models, we calculated the pairwise correlation between contribution scores of both models on cell-type-specific regions, revealing biopsy topic 8 to correlate the strongest with the MES-like cancer cell lines (Fig. 4g) and topics 14, 18 and 19 correlating the most with pro-neural cell line LN229. This correlation is clearly lower compared to any of the pairwise similarities found between the cell lines. We reasoned that this approach, compared to using prediction scores directly, reduces the influence of non-sequence-mediated changes, such as copy number variations (CNVs). Consistent with this interpretation, we find that contribution score correlations between models remain stable across CNV and neutral regions, whereas accessibility/prediction correlations vary more strongly with CNV state (Supplementary Fig. 8d).

**Fig. 4 | Comparing MES-like states across cancers in cell lines and biopsy samples. a**, Overview of the DeepCCL and DeepGlioma model trained on human cancer cell lines and GBM biopsy data, respectively. Created in BioRender; Aerts, S. https://BioRender.com/77yqidq (2025). **b**, Heat map of Pearson correlations for log-transformed cell-type-specific test-set peak heights and predictions from DeepCCL, separated per cell line. MES-like states are highlighted. **c**, Comparison of Pearson correlation between log-transformed peaks and predictions per cell line (*n* = 8 classes) for DeepCLL (base model and fine-tuned) and an ensemble ChromBPNet model, on all test-set consensus peaks (left) and cell-type-specific peaks (right). *P* values were derived using a two-sided *t*-test and corrected using the Benjamini–Hochberg method (*P < 0.05, **P < 0.01, ***P < 0.001). The exact *P* values between the pairs (DeepCCL Base, DeepCCL), (DeepCCL Base, ChromBPNet ensemble) and (DeepCCL, ChromBPNet ensemble) for all test-set peaks and cell-type-specific test-set peaks were 0.00037, 1.07609 × 10⁻⁷, 0.74659 and 0.03053, 0.07177, 0.00016, respectively. **d**, Highlighted example intronic region in *AXL*, showing multiple ChIP–seq tracks, cell line ATAC tracks and biopsy topic scATAC tracks (top). Contribution scores for MES-like model classes and an

OPC/NPC-like class from the DeepCCL and DeepGlioma models, with ChIP–seq matched motifs highlighted. **e**, Comparison of log counts of CREsted-identified grouped motif sets between GBM MES-like and melanoma MES-like cell lines for the DeepCCL base model (top) and DeepCCL (bottom). Negatively contributing patterns were given a negative count. **f**, *t*-SNEs based on pycisTopic's topic-cell contributions of human biopsy cells, colored by patient sample (left) and topic probability for a set of topics (right). **g**, Contribution score Spearman correlation between contribution scores obtained from the DeepCCL and DeepGlioma model per model class for the top 1,000 most specific regions per cell line and per topic. **h,i**, Heat maps of identified motifs in top 2,000 regions for DeepCCL and DeepGlioma classes. The color scale indicates motif importance, represented by the log-transformed pattern count. Negatively contributing motifs were given negative importance values. Motifs were manually annotated. For box plots in **c**, the top/lower hinge represents the upper/lower quartile, and whiskers extend from the hinge to the largest/smallest value no further than 1.5 times the interquartile range from the hinge, respectively. The median is used as the center.

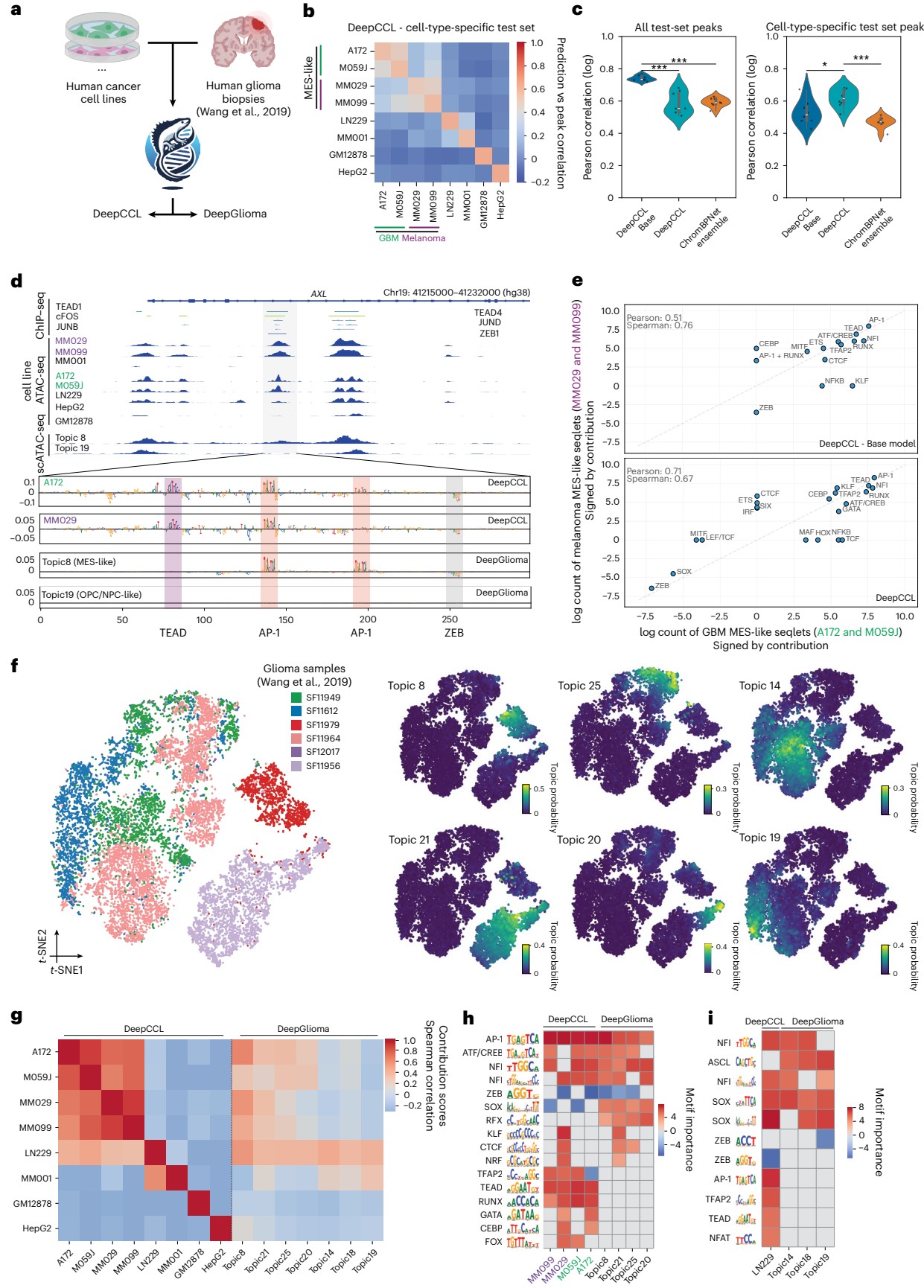

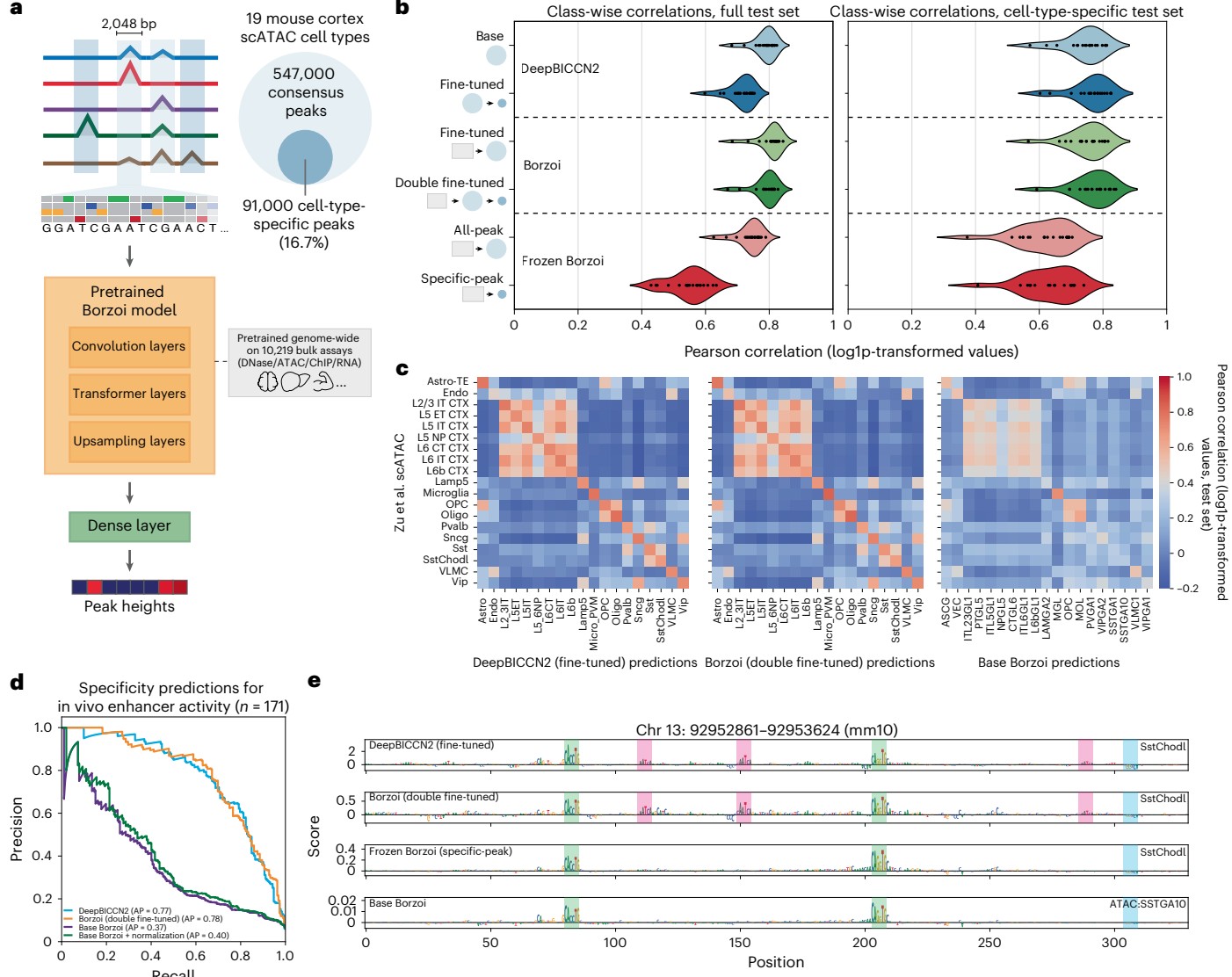

**Fig. 5 | Transfer learning from and benchmarking against a large pretrained sequence-to-function model. a**, Schematic overview of the Borzoi transfer learning process, fine-tuning and evaluating a modified Borzoi model on all consensus peaks, cell-type-specific peaks, or both. **b**, Comparison of mean $r$ between log-transformed predictions and peak heights per cell type ($n = 19$ cell types) with the test dataset for the DeepBICCN2 model (Fig. 2), the fine-tuned Borzoi and a fine-tuned Borzoi with the model trunk ('pretrained Borzoi model') frozen. Icons denote the datasets these models were successively trained on (original Borzoi training data, mouse cortex consensus peaks, or mouse cortex cell-type-specific peaks), matching **a**. **c**, Heat maps showing the correlation of model predictions with peak heights for matching classes from the Zu et al. mouse brain atlas[19], evaluated on the cell-type-specific peaks in each model's corresponding test set. **d**, Precision–recall curve on specificity predictions on a set of in vivo-validated enhancers. **e**, Contribution scores for a validated SstChodl enhancer[17] (AiE2428m) for the specific models and a matching class from the base Borzoi model.

Analyzing the motifs underlying cell line and biopsy MES-like states revealed AP-1 and CREB/ATF motifs to be shared, while TEAD motifs are specific to cell lines, and SOX and RFX motifs are specific to biopsy samples (Fig. 4d,h). For the OPC/NPC-like states, we identified SOX motifs to be shared across cell lines and biopsy enhancers (Fig. 4i) and certain TFBSs to be specific to biopsy enhancers, like ASCL (Fig. 4i). Notably, LN229 still shares a part of the identified MES-like program (for example, AP-1, TEAD) making it more accurately defined as a mixed pro-neural-mesenchymal cell line from an accessibility perspective[68].

In conclusion, we show that CREsted can be used to compare cancer cell states across biopsy samples and cell lines, highlighting that the often-found MES-like enhancer logic in cell lines is not fully recapitulated in the tumor. Moreover, we highlight a relevant use case for CREsted topic classification models on the more continuous glioma scATAC-seq data.

## CREsted-trained models outperform large, pretrained models on cell-type-specific chromatin accessibility predictions

Besides training enhancer models from scratch, an alternative strategy involves transfer learning from large pretrained models, such as Enformer[10] and Borzoi[15]. The preexisting knowledge base of these models could help with the training of dataset-specific models[69–72]. To accommodate this, CREsted provides functionalities to update the architecture of pretrained models to allow input sequences of any specified length, and to add a new dense layer after the final flattened embedding to predict peak heights for cell types in a new dataset (Fig. 5a).

As an illustration, we fine-tuned Borzoi to predict cell-type-level chromatin accessibility in the mouse cortex by double fine-tuning on all peaks followed by specific peaks (Fig. 5a and Supplementary Note 3). We also trained a version of this model with the pretrained section

of the model frozen, training only the new dense layer on either the consensus regions or only the specific set.

We observed that DeepBICCN2 generally matches the performance of the substantially larger fine-tuned Borzoi models, although those retain slightly better generalizability on consensus peaks following cell-type-specific fine-tuning (Fig. 5b, Supplementary Note 3 and Supplementary Fig. 9a,b). These results demonstrate that simple, parameter-efficient architectures can effectively compete with large-scale pretrained models in predicting region accessibility.

To compare the fine-tuned Borzoi models and DeepBICCN2 to the base Borzoi model, we evaluated their performance on an external, unseen dataset[73], containing cell types present in all tested models. On this data, the fine-tuned Borzoi and DeepBICCN2 have similar performance and both outperform the base Borzoi model (Fig. 5c). Base Borzoi has particular difficulties in distinguishing between different subclasses of GABAergic and glutamatergic neurons. Although the glutamatergic neurons in particular have similar chromatin accessibility profiles, correlations between the measured peaks show these subclasses are still distinct (Supplementary Fig. 9c,d). In line with this result, base Borzoi performs substantially worse in classifying 171 in vivo-validated mouse cortex enhancers[16,17], achieving much lower average precision compared to DeepBICCN2 and the double fine-tuned Borzoi (Fig. 5d and Supplementary Fig. 9e). These results indicate that the base Borzoi model does not perform well at increased cell-type resolution and that fine-tuning is necessary to achieve comparable results to models trained from scratch. Focusing on sequence explanations, we observe that a subset of motifs are identified by all models but that the fine-tuned models were able to identify additional motifs (Fig. 5e and Supplementary Fig. 10), for example, a LHX6-like TFBS in a validated SstChodl enhancer.

Next to large, supervised models like Borzoi, self-supervised genomic language models (gLMs) have also been trained to serve as foundations for downstream tasks. To assess how these models compare to fine-tuning Borzoi, we fine-tuned two gLMs, HyenaDNA[74] and the Nucleotide Transformer[75], following the same approach as the one used for fine-tuning Borzoi. This resulted in worse-performing models for both gLMs as compared to either fine-tuning Borzoi or training CREsted models from scratch (Supplementary Fig. 9f), despite in some cases much higher parameter counts and corresponding computational cost of training.

In summary, CREsted allows for training of new models that can outperform large pretrained models, whether through transfer learning or from scratch. We show that the base pretrained models appear to fail at differentiating cell types at high resolution in the mouse motor cortex.

## CREsted designs enhancers specifically active in targeted cell types of a developing zebrafish

To assess the use of CREsted for the design of organism-level cell-type-specific enhancers, we trained a peak regression model, DeepZebrafish, on a scATAC-seq dataset of the developing zebrafish embryo[76] comprising 20 developmental stages, and 639 cell type–timepoint combinations (Fig. 6a). We evaluated the performance of the model to predict the cell-type specificity of 54 validated enhancers[76]; 74% showed high and specific predictions for the corresponding cell-type class (Fig. 6b).

Next, we designed enhancers specific for endothelial cells and two closely related cell types: cardiac and somatic muscle cells. ISE was used to optimize randomly initialized DNA sequences toward cell-type-specific enhancers[41]. To ensure cell-type specificity, we defined a cost function to minimize the Euclidean distance (L2 distance) between the model's prediction vector and a vector with targeted levels of chromatin accessibility. In this case, we aimed for high levels in the target cell type and no accessibility in other cell types.

We performed 30 iterations of ISE (Fig. 6c–e), and for each target cell type we tested a set of three candidate enhancers using in vivo enhancer reporter assays in zebrafish embryos. All enhancers designed for either cardiac or somatic muscle cells were specifically active in their target cell type (Fig. 6f,g). However, the efficiency of cardiac muscle enhancers was overall lower. Only one of three enhancers designed for endothelial cells had strong and specific activity, while another endothelial enhancer was active in the correct cell type, but the signal was weak (Fig. 6h).

We next generated dual-specificity enhancers with predicted accessibility ratios for cardiac versus somatic muscle cells: 1/1, 0.5/1 and 1/0.5. While sequences converged to targets (Fig. 6i,j) and most cardiac$_1$/somatic$_1$ and cardiac$_1$/somatic$_{0.5}$ designs yielded dual activity, expression efficiency was notably reduced compared to single-target counterparts. Conversely, only one of three cardiac$_{0.5}$/somatic$_1$ enhancers achieved dual activity and these enhancers showed overall higher somatic muscle labeling efficiency (Fig. 6m). We conclude that Deep-Zebrafish enables dual-code enhancer design, although precisely controlling expression magnitude remains nontrivial.

Finally, we investigated the enhancer code underlying these cell types. Both cardiac and somatic muscle enhancers have MEF, TEAD and E-box TFBS motifs (Fig. 6n,o), while GATA and NKX TFBSs are additionally identified in cardiac muscle enhancers (Fig. 6n,o). Lastly, for endothelial enhancers, we found nuclear receptor, SOX, ETS and MEF motifs.

In conclusion, CREsted scales to large-scale developmental scATAC-seq datasets and can be used to design enhancers that are specific to a target cell type across an entire organism.

## Discussion

High-throughput identification and characterization of CREs have been long-standing challenges in genomics, hampered by high false-positive rates[77]. Sequence-to-function models have emerged as promising methods to overcome this limitation[5,8–10] (Supplementary Note 4). Differential chromatin accessibility has been shown to be an especially

**Fig. 6 | DeepZebrafish can be used to design cell-type-specific enhancers in the whole zebrafish over development. a**, Schematic overview of 48 hours post fertilization (48 hpf) zebrafish embryo and UMAP of scATAC-seq cells with cell types of interest highlighted. **b**, Heat map of prediction score of DeepZebrafish on validated zebrafish enhancers and percentage of embryos that were active for the enhancer. **c–e**, Prediction score on designed enhancers for each iteration of the design process (box plot; $n = 200$ sequences) and prediction score of tested enhancers for each iteration (lines) for enhancers designed for: cardiac muscle (**c**), somatic muscle (**d**) and endothelium (**e**). **f–h,k–m**, Brightfield and fluorescence images (overlaid) of enhancer reporter assays in 48 hpf zebrafish embryos for enhancers designed for: cardiac muscle (**f**), somatic muscle (**g**) endothelial cells (**h**), cardiac and somatic muscle in a 1:1 ratio (**k**), cardiac and somatic muscle in 2:1 ratio (**l**) and cardiac and somatic muscle in 1:2 ratio (**m**). Scale bar, 500 μm. Fractions indicate the number of embryos that look similar to the representative one over the total number of embryos that were imaged per condition. Experiments were performed in multiple embryos (see fractions) and in three replicates (different injection days and clutches). **i**, Prediction score of tested dual-code enhancers for both body muscle and cardiac muscle class for each iteration. **j**, Prediction score for body muscle (left) and cardiac muscle (right) class for all designed enhancers ($n = 200$ sequences) at last iteration. **n**, Contribution scores for one synthetic enhancer per set of enhancers. **o**, Heat map of identified patterns in enhancers designed for cardiac and/or somatic muscle cells showing the natural log of the number of seqlets per class over all designed enhancers. **p**, Patterns identified in enhancers designed for endothelial cells. BV, blood vessel; Brn, brain; Endo, endothelial; Hrt, heart; Intest., intestine; Mus., muscle; NC, neural crest; Neu., neuron; Neu. h., neuron (head); Neu. t., neuron (trunk); Olf. b., olfactory bulb; Olf. p., olfactory placode; Ot., otic; RP, roof plate. For the box plots in **c–e** and **j**, the top/lower hinge represents the upper/lower quartile and whiskers extend from the hinge to the largest/smallest value no further than 1.5 times the interquartile range from the hinge, respectively. The median is used as the center.

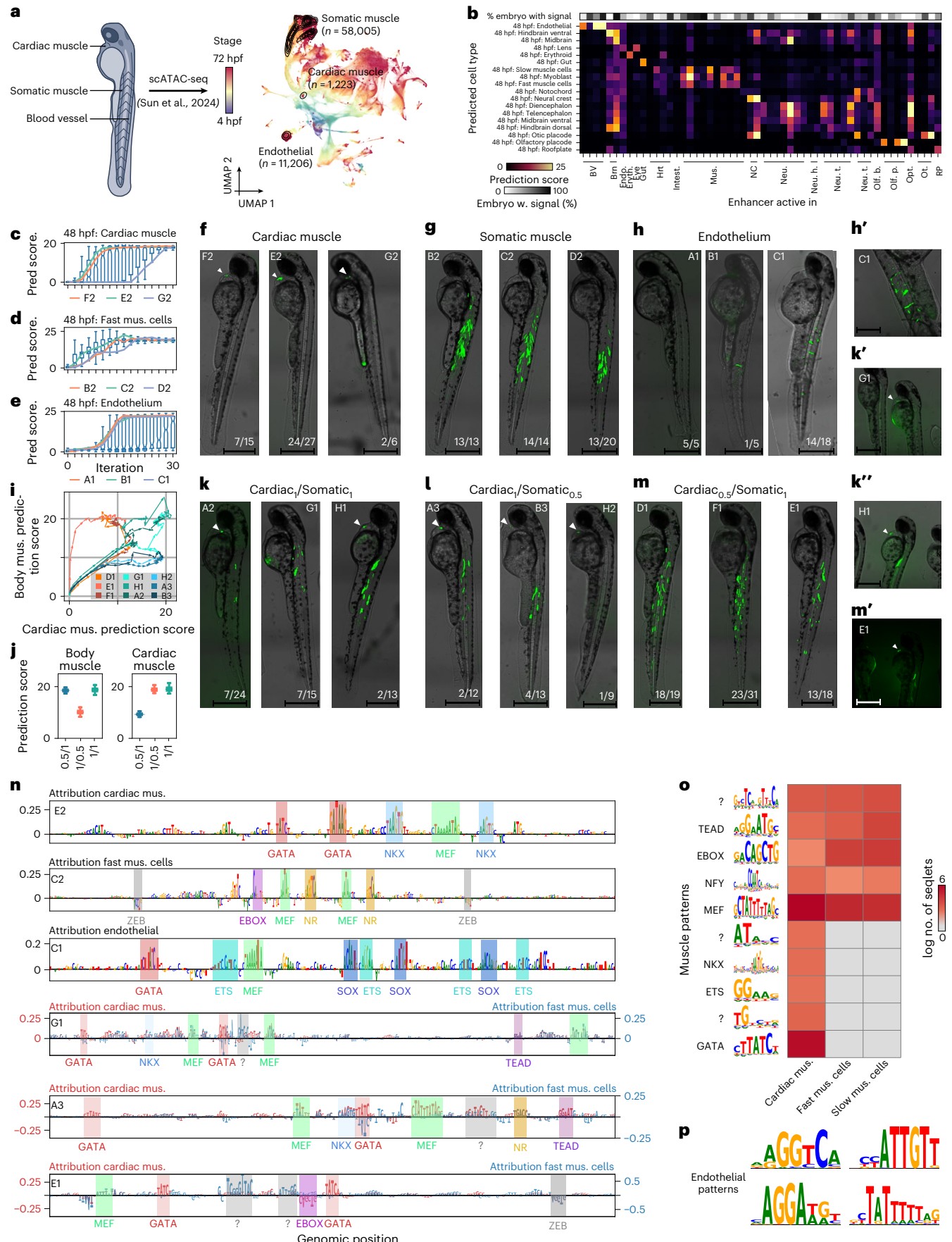

powerful modality to model cell-type-specific enhancers and is currently the most reliable indicator of enhancer activity in heterogeneous cell populations[16–18,26,41]. To fully exploit enhancer modeling on scATAC-seq data, we have optimized the training and downstream analyses of sequence-to-function enhancer models that predict chromatin accessibility across cell types, in a robust and user-friendly manner, by developing CREsted.

We applied CREsted to a wide range of biological systems and showed that it can: (1) accurately predict cell-type-specific chromatin accessibility, including for unseen sequences from other species; (2) identify the nucleotide-level determinants of cell-type-specific chromatin accessibility, revealing TFBSs that we validated using ChIP−seq data, summarized in enhancer-code tables and linked to TFs based on gene expression data; (3) enable fine-tuning of large, pretrained models; (4) be used to model large-scale scATAC-seq atlases containing many cell types[26] and; (5) design synthetic enhancers that exhibit cell-type-specific activity across an entire organism.

Across these analyses, we gained novel insight into enhancer codes for: (1) mouse cortical cell types, revealing cell-type-specific regulators at the *cis*-regulatory level; (2) PBMC types, including the explanation of nucleotide contributions of the densely packed *IFNB1* enhanceosome; (3) MES-like states across melanoma and GBM, revealing similar regulators across both cancers and; (4) organism-level single and dual cell-type-specific synthetic enhancers of two closely related cell types in zebrafish that share a large portion of their enhancer code while still being able to encode cell-type-specific enhancer activity (Supplementary Note 5).

CREsted is positioned within a rapidly expanding domain of enhancer and gene locus modeling tools, targeting different aspects of the *cis*-regulatory code. One major direction focuses on the prediction of gene expression data[5,10,15,71,72,78], typically using multi-class models covering hundreds of tissues and cell lines with large input windows that can integrate information from multiple CREs. Such models[10,15,78] can be used for inference and variant effect prediction and be fine-tuned to unseen cell types[71,72]. Packages such as gReLU[29] facilitate their usage. Although these approaches provide global models of gene regulation across diverse tissues, they are computationally expensive and not well suited for decoding enhancers at cell-type-specific resolution (as we showed for mouse cortex cell types) or enhancer design. Another direction entails local enhancer modeling using smaller-scale accessibility-based models. For example, ChromBPNet[8] and scPrinter[27] can be trained to accurately predict accessibility at base-pair resolution and allow for TF footprinting analyses of a single cell type/line at a time. Such models reveal important TFBSs underlying chromatin accessibility, but multiple models need to be trained to model enhancers across cell types. The CREsted package is optimized for training multi-class models encompassing many cell types from scATAC-seq atlases. Compared to general-purpose packages like gReLU, where similar models can be trained (as we illustrated on mouse cortex), CREsted focuses specifically on modeling of cell-type-specific enhancers through optimized data preprocessing and track normalization, training modalities with fine-tuning on cell-type-specific peaks, and an improved loss function to focus on cell-type-specific signals. Likewise, it contains optimized downstream functions for enhancer design and to link TFBSs to candidate TFs. The multi-class paradigm used in CREsted also enables straightforward scaling to large scATAC-seq datasets across entire tissues and organisms, as we illustrated here on whole-zebrafish development. A downside of multi-class models, however, is that they may misuse positive sequence features (for example, TF activator binding sites) of one class to negatively influence predictions of another, leading to certain motifs being assigned incorrect negative contributions (thus appearing as TF repressor sites), as exemplified in Fig. 2g. Regardless, multi-output models can still identify repressive chromatin factors[41].

In summary, we present CREsted, a user-friendly software package aimed at modeling, analyzing and designing enhancers from scATAC-seq data. Its source code and all models we developed are publicly available. The CREsted model repository also contains all the legacy models from our previous studies, including human, mouse and chicken brain[16,26], mouse liver[25], fly brain[24] and human melanoma models[13,64].

## Online content

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

## Methods

### CREsted workflow

CREsted (RRID: SCR_026617) consists of four main modules: data preprocessing, model training, enhancer code analysis and enhancer design. CREsted can be found at https://crested.readthedocs.io/ and https://github.com/aertslab/CREsted/.

**Data preprocessing.** CREsted preprocessing has two main modes: cell-type-specific peak-height modeling and topic modeling[34,36] (RRID: SCR_026618).

*Peak-height preprocessing.* A peak-height matrix is generated over consensus peaks using CPM-normalized pseudobulk BigWig tracks per cell type using pybigtools[79] (RRID: SCR_026627), producing the max, mean, sum or logarithm of the sum of either cut sites or coverage per cell type. Optionally, peak widths can be resized while ensuring that they stay within chromosome boundaries. By default, a peak width of 2,114 bp is used and the peak height is calculated from the center 1,000 bp. The matrix is stored in an 'anndata' object[32,80] (RRID: SCR_018209).

*Peak normalization.* A min–max normalization is applied by using cell-type-specific scaling factors based on constitutive peaks[16]. Constitutive peaks are automatically identified as those with high accessibility in a cell type (top 1% by default) that are nonspecific (Giniindex $< \mu_{\text{Giniindex}-\text{s.d.}_{\text{Giniindex}}}$; $\mu_{\text{Giniindex}}$ and s.d.$_{\text{Giniindex}}$ are the average Gini index and standard deviation of the Gini index across all peaks, respectively) and the average values of those peaks, per cell type, are used as cell-type-specific scaling factors (Supplementary Note 1).

*Topic modeling preprocessing.* Either a binary matrix (topic classification) or a probability matrix is generated over consensus peaks for each topic containing either a binary label indicating regions belonging to topics or the probability of each region for each topic. This matrix is stored in an anndata[32,80] (RRID: SCR_018209) object.

*Dataset splitting.* Regions are split into training, validation and test sets either based on a defined chromosomal split or by randomly shuffling regions following predefined proportions.

**Model training.** CREsted implements training of multi-label models—predicting accessibility in multiple cell types from one sequence—from scratch, in both classification and regression settings, and by transfer learning from Enformer[10] (RRID: SCR_024805) and Borzoi[15] (RRID: SCR_026619). Keras 3.0 (RRID: SCR_026159) is used as the model training framework, allowing both TensorFlow (RRID: SCR_016345) and PyTorch (RRID: SCR_018536) backends.

*Regression models.* By default, the sum of the cosine similarity between predicted and ground-truth peak-height vectors and the MSE between the log-transformed predicted and ground-truth peak-height vectors is used as the loss function. By default, the cosine similarity value is dynamically weighted based on the MSE magnitude. Other loss functions implemented in CREsted include: cosine similarity, Poisson loss and multinomial loss. By default, the Adam optimizer is used with a learning rate of $1 \times 10^{-3}$, which is decayed with a factor of 4 after five epochs without a decrease in validation loss, and stopping is applied after ten epochs without a decrease in validation loss.

*Fine-tuning to cell-type-specific regions.* Regression models can be fine-tuned on cell-type-specific peaks with a lower learning rate than their base models. Cell-type-specific peaks are defined as those with Giniindex $> \mu_{\text{Giniindex}} + \text{s.d.}_{\text{Giniindex}}$; $\mu_{\text{Giniindex}}$ and s.d.$_{\text{Giniindex}}$ with $\mu_{\text{Giniindex}}$ and s.d.$_{\text{Giniindex}}$ are the average Gini index and standard deviation of the Gini index across all peaks, respectively.

*Classification models.* By default, the binary cross-entropy loss function is used with an Adam optimizer and learning rate of $1 \times 10^{-3}$. The learning rate is decayed with a factor 4 after five epochs without a decrease in validation loss and stopping is applied after ten epochs without a decrease in validation loss.

*Model architectures.* Multiple architectures are implemented in CREsted. By default, a dilated convolution neural network (CNN) is used for regression and a regular CNN for topic classification.

*Transfer learning from pretrained supervised models.* For transfer learning from Enformer[10] and Borzoi[15], the models' input size is shrunk (Supplementary Note 3), and they are fine-tuned to predict cell-type-specific peak heights in two rounds: one on consensus peaks and another on cell-type-specific peaks with a lowered learning rate.

**Enhancer code analysis.** *Contribution score calculations.* Both gradient-based and prediction-based methods are used for contribution calculation, inspired by the tfomics package[37].

$$\text{IG}_i(x, x') = \int_{\alpha=0}^{1} \frac{\partial f(x' + \alpha(x - x'))}{\partial x_i} \partial \alpha \tag{1}$$

$$\text{IG}_i(x, x' = 0) = \int_{\alpha=0}^{1} \frac{\partial f(0 + \alpha(x - 0))}{\partial x_i} \partial \alpha = \int_{\alpha=0}^{1} \frac{\partial f(\alpha x)}{\partial x_i} \partial \alpha \tag{2}$$

$$\text{EIG}_i(x) = \frac{1}{m} \sum_{n=1}^{m} (\text{IG}(x_i, S(x_i))) \tag{3}$$

With:
$x$, the sequence to explain;
$x'$, the baseline sequence;
$f(x)$, the model to explain with;
$\alpha$, the progress along the integration path;
$S(x)$, a function randomly permuting $x$ along the sequence axis; and
$m$, the number of permuted baselines to use.

In integrated gradients[81] (IGs; equation (1)), gradients are computed along a path from a baseline to the input sequence being explained. By default, a zeroed-out sequence is used as baseline (equation (2)) and the integral is approximated using the Riemann integral over gradients for 26 steps ($\alpha$), including the baseline and final sequence. In expected integrated gradients[82] (EIGs; equation (3)), inspired by expected gradients[83], the input sequence is permuted 25 times ($m$; by default) and this is used as the baseline.

In silico saturation mutagenesis mutates each input nucleotide to all other nucleotides and quantifies the effect of each mutation on the model's prediction.

*TF-MoDISco analysis.* Contribution scores, for the relevant class(es), are calculated on cell-type-specific regions and tfmodisco-lite[38,39] (RRID: SCR_024811) is performed to obtain motifs from those regions. We provide a motif database[34], which is matched to the found motifs in a report.

*Pattern clustering.* Tomtom motif similarities are calculated for all patterns across TF-MoDISco runs per cell type using tangermeme[40] (v.0.4.0; RRID: SCR_026620). Groups of patterns passing a certain similarity score threshold are merged and represented by the pattern with the highest information content. Single-instance patterns with an information content below a defined threshold are discarded.

*Pattern to TF matching.* Identified motif patterns are matched to TF candidates based on a motif-to-TF database[34], and the Pearson correlation between the importance vector of the motif pattern over all cell

types and the average expression per cell type is calculated. Only TFs that pass a correlation threshold (0.2 by default) are kept.

**Enhancer design.** *Seed sequence generation.* The enhancer design methods in CREsted are inspired by Taskiran et al.[41]. Random DNA sequences of a given length are generated that follow the GC content of genomic sequences. To this end, the fraction of each nucleotide of genomic sequences is calculated for each position, and nucleotides are sampled according to this position-dependent distribution.

*Optimization function.* Two optimization functions are implemented in CREsted. The default function optimizes a weighted difference between a target class of interest and all other classes (equation (4)).

$$\text{cost}_{S_i} = [X_{ct}(S_i) - X_{ct}(S_{i-1})] - \frac{1}{N}\sum_{bg}^{B}(w_{bg}[X_{bg}(S_i) - X_{bg}(S_{i-1})]) \quad (4)$$

With:
$ct$, the target class;
$S_i$, the designed sequence at iteration $i$;
$X_{ct}(S)$, the prediction score of the model on sequence (S) for cell type $ct$;
$N$, the total number of classes;
$B$, the set of classes other than the target class; and
$w_{bg}$, the user-defined weight for class 'bg' (defaults to 1).
Another function optimizes the Euclidean distance between a predicted vector of chromatin accessibility to a user-defined target one (equation (5)).

$$\text{cost}_s = \sqrt{\sum_{ct}^{C}(X_{ct}(s) - t_{ct})^2} \quad (5)$$

With:
$C$, the set of cell types of interest;
$X_{ct}(s)$, the prediction score of the model for cell type $ct$ on sequence $s$; and
$t_{ct}$, the target prediction score for cell type $ct$.
Custom optimization functions can be defined by the user.

*ISE.* A set of seed sequences is optimized by making all possible nucleotide substitutions for each sequence and selecting the most optimal substitution according to an optimization function. This process is repeated for a user-defined number of iterations[41].

*Motif embedding.* A set of seed sequences is optimized by sequentially placing a set of user-defined TFBSs in each sequence and selecting the most optimal location according to an optimization function, while ensuring that the newly placed binding site does not overlap previously placed binding sites[41].

**Enhancer models**
**Model training.** *CREsted mouse cortex peak regression model, Deep-BICCN2.* Mouse motor cortex was downloaded from ENCODE[42] (GSE229169), and cut-site BigWigs (RRID: SCR_007708) were generated per cell type[16]. Preprocessing and training (pretraining and fine-tuning) were performed using default parameters, except using the top 3% of peaks per cell type to calculate peak-normalization scaling factors. Regions of chromosomes 8 and 10 were used for validation (56,064 consensus and 9,951 cell-type-specific peaks), 9 and 18 for test (49,936 consensus and 8,198 cell-type-specific peaks) and the remaining for training (440,993 consensus and 73,326 cell-type-specific peaks). Stochastic 3-bp sequence shift and reverse-complement augmentation were used during training with a batch size of 256 for pretraining and 64 for fine-tuning.

*gReLU mouse cortex peak regression model.* A default ($6 \times 10^6$ parameters using 512 convolutional filters) and large ($22 \times 10^6$ parameters using 1,024 convolutional filters) multi-class regression model was trained with the gReLU package[29] (v.1.0.3; RRID: SCR_026621), following the 3_train.ipynb tutorial notebook. The same cut-site BigWigs (RRID: SCR_007708) and consensus peaks from the Zemke et al. dataset[42] and the same train–validation–test splits and the same peak-height scalars per cell type as used for training the CREsted peak regression model were used.

*CREsted human PBMC peak regression model, DeepPBMC.* Coverage BigWigs (RRID: SCR_007708) were downloaded from https://ucsc-tracks.aertslab.org/papers/scatac_benchmark (ref. 84). Preprocessing and training (pretraining and fine-tuning) were performed using default parameters, except using the top 3% of peaks per cell type to calculate peak-normalization scaling factors. Regions of chromosomes 8 and 10 were used for validation (29,385 consensus and 6,070 cell-type-specific peaks), 9 and 18 for test (19,892 consensus and 4,303 cell-type-specific peaks) and the remaining for training (278,687 consensus and 51,644 cell-type-specific peaks). Stochastic 3-bp sequence shift and reverse-complement augmentation were used during training with a batch size of 256 for pretraining and 64 for fine-tuning (using $1 \times 10^{-6}$ as the learning rate).

*CREsted cancer cell line peak regression model, DeepCCL.* Coverage BigWigs (RRID: SCR_007708) were generated for the cell lines HepG2, GM12878, MM029, MM099, MM001, A172, M059J and LN229 using deepTools[85] (RRID: SCR_016366) and WiggleTools[86] (RRID: SCR_001170) and peaks were called using MACS[87] (v.3; RRID: SCR_013291) and merged into consensus peaks using pycisTopic[34] (v.2; RRID: SCR_026618). Preprocessing and training (pretraining and fine-tuning) were performed using default parameters, except using the top 3% of peaks per cell type to calculate peak-normalization scaling factors, defining cell-type-specific regions with coefficient of variation of the coverage across all regions greater than 0.9, using the DilatedCNN model architecture with Swish activation function, a filter size of 11 for first convolutional layer and a dropout of 0.2, using the PoissonLoss function over log-transformed values and using the Lion optimizer[88]. Regions of chromosomes 7, 8, 9 and 10 were used for validations (80,674 consensus and 39,585 cell-type-specific peaks), 5 and 6 for test (48,268 consensus and 27,020 cell-type-specific peaks) and the remaining for training (285,790 consensus and 140,761 cell-type-specific peaks). Stochastic 5-bp sequence shift and reverse-complement augmentation were used during training with a batch size of 512, a learning rate of $5 \times 10^{-5}$ and a weight decay of $1 \times 10^3$ for pretraining, and a batch size of 64, a learning rate of $5 \times 10^{-6}$ and a weight decay of $1 \times 10^{-1}$ for fine-tuning.

*ChromBPNet cancer cell line model.* For each cell line, a separate ChromBPNet[8] model was trained using the instructions provided at the corresponding GitHub page, using the previously called peaks and the same train–validation–test split as the CREsted model. After training, the models were ported to CREsted by keeping only the count output head and integrating all eight models into one ensemble. Finally, the ChromBPNet ensemble was fine-tuned to cell-type-specific regions in a similar way to the DeepCCL training regime. The Pearson correlation coefficient calculated on log-transformed ground-truth peak heights and model predictions (48,268 and 27,020 test-set samples for the base and cell-type-specific comparison, respectively) were used to evaluate the performance of the models for each class.

*CREsted glioma biopsy topic model, DeepGlioma.* BAM files were downloaded from the European Genome-phenome Archive repository, under EGAS00001003845 and EGAD00001005314, converted to fragment files and used as input for pycisTopic[36] (v.2; RRID: SCR_026618). Pseudobulk ATAC-seq profiles were generated and peaks were called

using MACS[87] (v.2; RRID: SCR_013291; $q$ value = 0.001), which were merged into consensus peaks with 500-bp width. Barcodes with less than 1,000 unique fragments, with transcription start site enrichment below 5 or that were recognized as doublets were removed. Leiden clustering (resolution of 0.4) was performed, resulting in 14 clusters that were manually annotated based on gene activity. Clusters annotated as healthy cells were removed, and the same pycisTopic workflow was rerun resulting in 14,275 cells and 311,118 regions. Twenty-five topics were selected based on topic metrics and cell-topic probabilities were corrected for batch effects using Harmony[89] (v0.0.10; RRID: SCR_022798). Region-topic probabilities were binarized using Otsu thresholding. One topic (topic 3) included only 21 regions and was excluded from the subsequent analysis. On the remaining 24 topics, a CREsted classification model was trained. Regions of chromosomes 8, 9 and 10 were used for validation (33,229 peaks), 5 and 6 for test (24,889 peaks) and the remaining for training (192,256 peaks). Stochastic 50-bp sequence shift and reverse-complement augmentation were used during training with a batch size of 128.

*CREsted zebrafish peak regression model, DeepZebrafish.* Fragment file and cell-level metadata file (containing cell-type and development timepoint annotations) were downloaded from NCBI's Gene Expression Omnibus (GEO; GSE243256). The data were preprocessed using SnapATAC2 (ref. 35; v.2.6.4; RRID: SCR_026622). Consensus peaks were called by first calling peaks per cell type–timepoint combination (snap.tl.macs3) followed by snap.tl.merge_peaks using default parameters. Next, a cut-site consensus peak count matrix was generated (snap.pp.make_peak_matrix) setting the counting_strategy option to 'insertion'. Finally, a normalized matrix containing counts per cell type–timepoint combination was made (aggregate_X) using the 'RPM' normalization method. The resulting matrix was used to train (pretrain and fine-tune) the CREsted peak regression model, using default parameters, except using 1,024 convolutional filters. Regions of chromosomes 8 and 10 were used for validation (70,993 consensus and 7,518 cell-type-specific peaks), 9 and 18 as test (77,948 consensus and 7,521 cell-type-specific peaks) and the remaining for training (793,273 consensus and 89,637 cell-type-specific peaks). Stochastic 3-bp sequence shift and reverse-complement augmentation were used during training with a batch size of 128 for both training phases and learning rate of $2 \times 10^{-6}$ for fine-tuning.

*CREsted Borzoi transfer learning.* A CREsted-ported Borzoi model (replicate 0) was initialized, with its input size reduced to 2,048 bp, cropping layer disabled and final head layers (Conv1D + Softplus) disregarded (Supplementary Note 3). This base model's weights were frozen for the 'frozen Borzoi models'. The 'Borzoi CNN models' followed the same architecture, except that only the convolutional tower was kept (Supplementary Note 3). The trunk model was then extended with Flatten, Dense and Softplus layers to predict 19 pseudobulk classes. Models were trained on the Zemke et al. dataset[42], following the same data and training protocol as DeepBICCN2, except for the 2,048-bp input size, a batch size of 32 and learning rate of $5 \times 10^{-5}$ for full-model first-round fine-tuning (on all consensus peaks) and $1 \times 10^{-5}$ for second-round fine-tuning (on cell-type-specific peaks) and frozen model fine-tuning. The 'frozen Borzoi models' were fine-tuned for one round only on either all consensus peaks or cell-type-specific peaks, as a single Dense layer does not benefit from multiple rounds of fine-tuning.

For all models, the lowest validation loss epoch was kept as the final model.

*Fine-tuning gLMs.* HyenaDNA[74] (RRID: SCR_027471, hyenadna-small-32k, https://huggingface.co/LongSafari/hyenadna-small-32k-seqlen) and Nucleotide Transformer[75] (RRID: SCR_027472, nucleotide-transformer-500m-1000g, https://huggingface.co/InstaDeepAI/nucleotide-transformer-500m-1000g) models were retrieved from the Hugging Face

repository using the Transformers library (v.4.54.1; RRID: SCR_027381) and were fine-tuned for cell-type-specific chromatin accessibility. For both models, the last layer's embeddings were mean pooled, and a Dense layer followed by Softplus activation was added and an input of 2,048 bp was used. Instead of one-hot encoding the input, they were passed through the respective models' tokenizers. PyTorch Lightning (v.2.5.2; RRID: SCR_027468), combined with CREsted (v.1.5.0) using the PyTorch backend (v.2.6.0+cu124) were used for model training. Both models were fine-tuned for two rounds using the same data and loss function as the Borzoi fine-tuning, stopping when their validation loss plateaued. The Nucleotide Transformer was fine-tuned with original learning rates of $5 \times 10^{-5}$ and $1 \times 10^{-5}$ for the first-round and second-round fine-tuning, with a batch size of 4 and a maximum of 20 epochs. HyenaDNA was trained with a learning rate of $1 \times 10^{-5}$ for the first round and $5 \times 10^{-6}$ for the second round, a batch size of 32 and a maximum of 50 epochs.

**Model evaluation.** *CREsted and gReLU performance comparison.* The mean Pearson correlation between log-transformed ground-truth peak heights and model predictions (49,936 consensus peaks and 8,198 cell-type-specific peaks) for each class ($n = 19$ classes) were used to evaluate model performance, and a two-sided Welch's $t$-test was used to test for significant differences between the correlation values per class across models. We report the $t$-statistic, degrees of freedom and two-sided $P$ value, which were corrected for multiple testing using the Benjamini–Hochberg procedure.

*Comparison of base-tuned, fine-tuned and scratch models.* Three models were trained on the Zemke et al. mouse cortex dataset[42]: (1) exclusively on consensus peaks (base), (2) first on consensus peaks followed by fine-tuning on cell-type-specific peaks (fine-tuned) and (3) exclusively on cell-type-specific peaks (scratch). Models were evaluated on test-set regions and using locus scoring. We trained baseline models on all peaks to establish a generalizable foundation.

*Chromosomal split benchmarking.* Models were trained on the mouse cortex dataset[42] using ten different chromosomal splits to cover all chromosomes in the test set at least once following default CREsted training parameters. Using all models, contribution scores were calculated on 171 functionally validated, on-target, mouse and human genomic enhancers[17] for the corresponding target classes. The pairwise Spearman correlation of their contribution score across models of different chromosomal splits was evaluated.

*Closed regions benchmarking.* The mouse motor cortex training set[42] was supplemented with an additional 550,000 randomly selected non-exonic, non-peak regions and together with the consensus peaks from this dataset was used to train a CREsted peak regression model ('Nonpeakextended' models) followed by fine-tuning on cell-type-specific regions. Model performance on both test-set peaks and test-set closed regions (defined as regions with aggregate signal equal to zero) were evaluated for both the base and fine-tuned models.

*Gene locus predictions.* Gene loci (50 kb upstream of the gene's transcription start site to 25 kb downstream of the 3' untranslated region) were scored using a 2,114-bp sliding window with 100-bp shifts and, for overlapping positions, the average prediction score was used.

*CREsted predictions on validated mouse cortex enhancers.* A set of 171 functionally validated, on-target, mouse and human genomic enhancers[17] were evaluated using DeepBICCN2. The sequences were zero-padded to fit the 2,114-bp model input. Similarly, the peak heights were evaluated directly and values were zero-padded to 1,000 bp to fit the model's output. Precision and recall were calculated using as

ground truth the cell type where the enhancer is active in (including secondary cell type if present) using the relevant classes.

For base Borzoi, the enhancers were evaluated in their genomic context by centering on the enhancer sequence and using the surrounding genomic DNA sequence to come to an input size of 524,288 bp and scored for the relevant mouse cortex classes taking as predicted values the center 32 bins, corresponding to 1,024 bp. These Borzoi predictions (for 19 relevant classes matched between Borzoi and DeepBICCN2) were transformed to obtain their counts without soft-clipping (that is, prediction scores > 96 become $96 + (\text{prediction score} - 96)^2$). An additional peak-scaling component was applied using the average prediction on housekeeping promoter regions (centered, and extended by genomic flankings) used in Supplementary Fig. 1 as scaling factors.

*EBF1 degradation analysis in B cells.* Exact hits of CCCCTGGG, CCCTAGGG and TCCCTGGG were identified and considered as potential EBF1 TFBSs and replaced by N's, resulting in a zero input. The *TCF3* gene locus (20 kb upstream and 5 kb downstream) was evaluated using 'crested.tl.score_gene_locus' and compared to chromatin accessibility data of wild-type B cells and B cells after EBF1 degradation[58].

*CREsted predictions against topics derived from the cisTopic analysis of human Gliomas.* Pseudobulk topic BigWigs (RRID: SCR_007708) were generated using the 'export_pseudobulk' function in pycisTopic[36] (v.2; RRID: SCR_026618) after binarizing cell-topic distributions. DeepCCL predictions and topic-pseudobulk accessibility were evaluated on the top 10,000 differentially accessible regions per cluster by averaging the predictions and accessibility of GBM and melanoma MES-like cell lines, and applying Spearman correlation to identify potential biopsy topics that share a similar MES-like regulatory landscape. Similarly, the predictions and accessibility of LN229 were used to find candidate OPC/NPC-like topics.

*GBM CNV analysis.* epiAneufinder[90] v.1.1.3 (RRID: SCR_026269) was applied directly to the count matrix of 14,275 cells and 311,118 regions derived from the pycisTopic analysis of glioma biopsy samples using default parameters and a 100-kb window size. The resulting dataframe was filtered to include only cells belonging to the cell state of interest (MES-like) or to the relevant topics (topic 8 and topic 21). Windows were then classified as either neutral (>99.9% of cells labeled as normal) or CNV (>2% of cells labeled with either gain or loss)

*CREsted predictions on validated zebrafish enhancers.* A set of 54 tested and active enhancers (Supplementary Table 6 in Sun et al.[76]) were rescaled to 2,114 bp centered on the middle of each enhancer and scored using DeepZebrafish.

*Comparisons of CREsted-trained models with base Borzoi.* Pseudobulk BigWigs (RRID: SCR_007708) were downloaded[19] (http://catlas.org/wholemousebrain) for selected cell types corresponding to those of Zemke et al.[42]. To this end, each peak was padded to 524,288 bp using the mm10 genome sequence and zeros where it fell outside chromosome boundaries and the core 32 bins (corresponding to 1,024 bp) were used for the comparison.

**Motif analysis.** *Mouse cortex motif analysis.* Contribution scores for each cell type were calculated on the 2,000 most cell-type-specific regions (obtained based on the average of chromatin accessibility and prediction score per region) and 'tfmodisco-lite' was applied using the function: crested.tl.modisco.tfmodisco(window = 1000, max_seqlets = 20000) followed by clustering of patterns using: crested.tl.modisco.process_patterns(sim_threshold = 4, trim_ic_threshold = 0.025, discard_ic_threshold = 0.2) and crested.tl.modisco.create_pattern_matrix(normalize = False, pattern_parameter = 'seqlet_count_log').

*Human PBMC motif analysis.* Contribution scores, for each cell type, were calculated on the top 1,000 most cell-type-specific regions (obtained based on the average of chromatin accessibility and prediction scores per region) and 'tfmodisco-lite' was applied using the function: crested.tl.modisco.tfmodisco(window = 1000, max_seqlets = 20000) followed by clustering of patterns using crested.tl.modisco.process_patterns(sim_threshold = 4.25, trim_ic_threshold = 0.05, discard_ic_threshold = 0.2) and crested.tl.modisco.create_pattern_matrix(normalize = False, pattern_parameter = 'seqlet_count_log').

*Human PBMC ChIP–seq comparison.* ChIP–seq peaks and BigWigs (RRID: SCR_007708) were downloaded from ENCODE for PAX5 (ENCFF827VVQ and ENCFF914QGY), EBF1 (ENCFF895MHN and ENCFF810XRY) and POU2F2 (ENCFF803HIP and ENCFF934JFA) and from ChIP-Atlas (https://chip-atlas.org/) for GATA3 (SRX4705120), RUNX1 (SRX1492212), ETS1 (SRX015825), CEBPA (SRX097095) and SPI1 (SRX4001818). For each pattern identified in the PBMC motif analysis, TFs were assigned based on Tomtom[40] (v.0.4.0; RRID: SCR_026620) similarity with our motif database[34]. Precision and recall was calculated, using multiple contribution score thresholds, where true positives indicate seqlets inside a ChIP peak, false-positive seqlets that are not inside the peaks and false-negative ChIP peaks without seqlets.

*Human PBMC UniBind comparison.* Unibind[56]-predicted direct ChIP–seq peaks were downloaded for: PAX5 and EBF1 in B cells (https://unibind.uio.no/factor/ENCSR000BHD.GM12878_female_B-cells_lymphoblastoid_cell_line.PAX5/ and https://unibind.uio.no/factor/ENCSR000DZQ.GM12878_female_B-cells_lymphoblastoid_cell_line.EBF1/); for GATA3, RUNX1 and ETS1 in CD4+ T cells (https://unibind.uio.no/factor/GSE76181.Jurkat_T-cells.GATA3/, https://unibind.uio.no/factor/GSE76181.Jurkat_T-cells.RUNX1/ and https://unibind.uio.no/factor/EXP000299.Jurkat_E6_1_T-cells.ETS1/); and for CEBPA and SPI1 in CD14+ monocytes (https://unibind.uio.no/factor/EXP000946.U937_adult_acute_monocytic_leukemia.CEBPA/ and https://unibind.uio.no/factor/EXP047756.MDMmonocyte_derived_macrophages.SPI1/). The overlap of those sites and TFBS instances from DeepPBMC, calculated using the tangermeme[40] (RRID: SCR_026620) recursive_seqlets function with a *P*-value parameter of 0.05, was assessed. Overlaps were only considered when at least 50% of the instance overlapped with the Unibind site. This analysis was performed for both the top 1,000 most cell-type-specific regions and all consensus peaks.

*Human PBMC motif enrichment analysis using pycisTarget.* A cisTarget motif database was generated on the PBMC consensus peaks by first creating a fasta file with 1-kb background padding using the command line utility create_fasta_with_padded_bg_from_bed.sh and scoring using the v.10 SCENIC+[34] motif collection (RRID: SCR_026702; https://resources.aertslab.org/cistarget/motif_collections/v10nr_clust_public/v10nr_clust_public.zip/) and using the command line utility create_cistarget_motif_databases.py. Both utilities are available on https://github.com/aertslab/create_cisTarget_databases.git/ (RRID: SCR_027473). Next, pycisTarget[34] (v.1.1; RRID: SCR_026626) was run on the same cell-type-specific peaks as used for the tfmodisco-lite analysis (see 'Human PBMC motif analysis') using default arguments.

*Human PBMC motif enrichment analysis using pyChromVAR.* The PBMC fragment matrix was processed using pyChromVAR[57] (v.0.0.4; RRID: SCR_027456). Peaks present in fewer than 50 cells and cells with fewer than 2,000 or more than 15,000 detected peaks, or with fewer than 4,000 or more than 40,000 total counts, were removed. Term frequency–inverse document frequency (TF-IDF) count normalization was used and dimensionality was reduced by latent semantic indexing, excluding the first component. Motifs retrieved from the JASPAR 2024 CORE (RRID: SCR_003030) vertebrate collection were identified.

Per-cell deviations (that is, activities) of motifs were computed with pyChromVAR using default parameters. Motif enrichment was computed by Wilcoxon rank-sum testing of the motif deviation scores. Finally, the 'Human PBMC UniBind comparison' analysis was repeated using the motifs identified with pyChromVAR.

*Comparison of contribution-based patterns and enriched motifs from pycisTarget and pyChromVar.* A pairwise similarity matrix was calculated between all identified motifs (both de novo by CREsted and enriched motifs found by pycisTarget and/or pyChromVar) using TomTom from memesuite-lite (v.0.2; RRID: SCR_027429)[91]. *t*-SNE dimensionality reduction was performed on this matrix using Scanpy (v.1.11.4; RRID: SCR_018139)[92].

*Contribution score calculation comparison benchmark.* Contribution scores were calculated on the 171 in vivo-validated enhancers from Ben-Simon et al.[17] for each enhancer's target class using three methods: IGs, EIGs and ISM (considering the maximum absolute effect per position). Pairwise comparisons were evaluated using Spearman and Pearson correlation. Additionally, the 'Human PBMC UniBind comparison', was rerun using ISM instead of EIGs.

*GBM motif analysis.* For DeepCCL, contribution scores for each cell type were calculated on the top 2,000 most cell-type-specific regions (based on the average of the chromatin accessibility and prediction score per region) and tfmodisco-lite was performed using the function: crested.tl.modisco.tfmodisco(window = 1000, max_seqlets = 20000), and the identified patterns were clustered using crested.tl.modisco.process_patterns(sim_threshold = 3.5, trim_ic_threshold = 0.1, discard_ic_threshold = 0.2) and crested.tl.modisco.create_pattern_matrix(normalize = False, pattern_parameter = 'seqlet_count'). The seqlet counts of the A172/M059J and the MM029/MM099 cell lines were averaged and log-transformed (keeping the original sign of the contribution scores). Pattern clusters were manually annotated, disregarding nonmeaningful seqlets and seqlets with zero counts in both GBM and melanoma cell lines.

For DeepGlioma, contribution scores, per topic, were calculated on the top 2,000 regions of topics 8, 21, 25, 20, 14, 18 and 19, and tfmodisco-lite was performed using crested.tl.modisco.tfmodisco(window = 500, max_seqlets = 20,000).

For the combined analysis, contribution scores were calculated for each cell line using DeepCCL on the 1,000 most cell-type-specific regions (based on the average chromatin accessibility and prediction score per region) and for each topic using DeepGlioma on the top 1,000 regions per topic. Additionally, the contribution scores of topic regions were calculated for all cell lines using DeepCCL. The Spearman correlation was used to compare contribution scores for each region across models (taking the middle 500 bp when comparing the DeepCCL and DeepGlioma contributions) and the median correlation coefficient across regions was used to obtain a similarity score between all cell lines and biopsy topics on a nucleotide contribution level. Finally, pattern clustering was performed using crested.tl.modisco.process_patterns(sim_threshold = 3.5, trim_ic_threshold = 0.1, discard_ic_threshold = 0.2) and crested.tl.modisco.create_pattern_matrix(normalize = False, pattern_parameter = 'seqlet_count_log').

*Zebrafish motif analysis.* Contribution scores were calculated for all designed enhancers for the classes: 'Slow muscle cells', 'Fast muscle cells', 'Cardiac muscle' and 'Endothelial' of development stage 72 hpf and 'tfmodisco-lite' was run (crested.tl.tfmodisco using default parameters). Muscle patterns were clustered using the functions crested.tl.modisco.process_patterns and parameters sim_threshold = 4.25, trim_ic_threshold = 0.05 and discard_ic_threshold = 0.15 and visualized in a heat map using the function clustermap_with_pwm_logos using parameter importance_threshold = 4.

**Input size benchmark.** *Model training.* Models with input sizes of 132 bp, 264 bp, 528 bp, 1,057 bp, 2,114 bp and 4,228 bp were trained on the PBMC dataset. First, the data were preprocessed as described under 'CREsted human PBMC peak regression model, DeepPBMC'. The number of dilation layers was adapted based on the input size (where the maximum number of dilation layers was determined using outputwidth = dilationfactor × (kernelsize − 1) + 1; and selecting the maximal number of layers minus 1 that still had output width ≥ 0) or zero-padding (by setting padding parameter to 'same') was used to circumvent convolutional layers with dimensions less than or equal to zero. This resulted in a total of 12 base models that were trained (278,687 consensus peaks; batch size 128) and fine-tuned (51,644 cell-type-specific peaks; batch size of 64; learning rate of $1 \times 10^{-6}$).

*Comparison of seqlet locations and identified patterns.* Contribution scores were calculated for the top 2,000 most cell-type-specific regions per cell type (crested.pp.sort_and_filter_regions_on_specificity on the average of the actual and predicted cell-type accessibility according to each model), and seqlets were identified using the tangermeme[40] (RRID: SCR_026620; v.0.4.0) function recursive_seqlets using default parameters. Finally, patterns were identified and compared across models using TF-MInDi[93] (RRID: SCR_027436; v.1.0.0).

**Zebrafish enhancer design.** *Zebrafish cardiac and body muscle enhancer design.* Cell-type classes of any timepoint annotated a: 'Slow muscle cell', 'Slow muscle cells', 'Fast muscle cells', 'Heart', 'Heart field' or 'Cardiac muscle' were considered and the average over all positive enhancers (see 'CREsted predictions on validated zebrafish enhancers') of the maximum prediction score over all classes was calculated ('avg_pos_prediction_score'). A cost function was defined that minimizes the Euclidean distance between the model's prediction score and a target vector where classes corresponding to 'Cardiac muscle' equals 'avg_pos_prediction_score' and other cell types are zero. Similarly, for body muscle a target vector was used where classes corresponding to 'Slow muscle cell', 'Slow muscle cells' or 'Fast muscle cells' equals 'avg_pos_prediction_score' and the other cell types are zero. Three sets of enhancers were designed to be active in both cell types, with varying levels of prediction scores: $cardiac_1/body_1$, with equal prediction score in cardiac and body muscle cells; $cardiac_1/body_{0.5}$, where the prediction score in body muscle cells is half the prediction score in cardiac muscle cells; and vice versa for $cardiac_{0.5}/body_1$. Two hundred random DNA sequences were initialized with similar nucleotide content across the peak region as consensus peaks and were optimized using ISE (Supplementary Table 1).

*Zebrafish endothelial enhancer design.* The same approach as was used to design enhancers targeted for either cardiac or body muscle was used to design endothelial cells. Classes corresponding to 'Notochord', 'Endothelial' and 'Otic placode' were considered, and a target vector where Endothelial was set to 'avg_pos_prediction_score' and the others to zero was used (Supplementary Table 1).

*Enhancers selection for experimental validation.* From the 200 generated sequences for each task (see above), 5 sequences were sampled at random per task and the nucleotide contributions were visually inspected. Based on this manual inspection, 3 of 5 randomly sampled sequences were tested for each task.

### Human cancer cell lines

**Culture of human GBM cell lines.** Cells were cultured following routine cell culture procedures, as described in, for example, https://doi.org/10.17504/protocols.io.q2jdycn. A172 cells (American Type Culture Collection (ATCC), CRL-1620; RRID: CVCL_0131) were cultured in DMEM (Thermo Fisher, 11965-092) medium supplemented with 10% FBS (Thermo Fisher, 17479-633), and 1% penicillin–streptomycin

(Life Technologies, 15140122). MO59J cells (ATCC, CRL-2366; RRID: CVCL_0400) were cultured in DMEM/F12 (Thermo Fisher, 11320-033) medium supplemented with 10% FBS, 1% penicillin–streptomycin and 0.05 mM non-essential amino acids (Thermo Fisher, 11140-068). LN229 cells (ATCC, CRL-2611; RRID: CVCL_0393) were cultured in DMEM (Thermo Fisher, 11965-092) medium supplemented with 5% FBS and 1% penicillin–streptomycin. All cell lines were passaged twice per week with Trypsin-EDTA 0.05% (Thermo Fisher, 11580-626).

**Bulk ATAC-seq.** Cells were treated with Trypsin-EDTA 0.05% and washed with DPBS (Thermo Fisher, 14190-169). Cell viability and concentration were assessed by the LUNA-FL Dual Fluorescence Cell Counter. Per cell line, 80,000 cells were used to perform ATAC-seq, following the OmniATAC-seq protocol[94]. After centrifugation at 500$g$ for 5 min at 4 °C, the supernatant was replaced with 50 µl ice-cold ATAC-seq lysis buffer (10 mM Tris-HCl, 10 mM NaCl, 3 mM MgCl$_2$, 0.1% IGEPAL CA-630, 0.1% Tween-20, 1% BSA and 0.01% digitonin), and the cells were mixed by pipetting. After incubation for 5 min on ice, 1 ml of wash buffer (20 mM Tris-HCl, 20 mM NaCl, 6 mM MgCl$_2$, 0.1% Tween-20 and 1% BSA) was added to the lysed cells and mixed gently by inverting three times. After centrifugation at 500$g$ for 5 min at 4 °C, supernatant was removed and the nuclei were resuspended in 50 µl ATAC mix (10 mM Tris-HCl, 10% dimethylformamide, 5 mM MgCl$_2$, 1× DPBS, 0.1% Tween-20, 0.01% digitonin and 3.75 ng µl$^{-1}$ Tn5 enzyme) and incubated for 1 h at 37 °C. Transposed DNA was purified using a QIAGEN MinElute purification column and eluted in 15 µl elution buffer (Qiagen). The ATAC-seq library was completed by amplifying the transposed DNA in a total volume of 50 µl PCR mix (15 µl purified DNA, 25 µl NEBNext High Fidelity PCR Master Mix (NEB), 2.5 µl FWD and 2.5 µl REV primer) with the following program: 72 °C for 5 min, 98 °C for 30 s, followed by ten cycles of 98 °C for 10 s, 63 °C for 30 s and 72 °C for 1 min). The final libraries were subjected to 0.4–1.2× double-sided Ampure purification and eluted in 20 µl elution buffer (Qiagen).

**Sequencing.** ATAC-seq libraries were sequenced on an Illumina NovaSeq X system using 51 cycles for read 1 (ATAC paired-end mate 1), 8 cycles for index 1 (sample index 1), 8 cycles for index 2 (sample index 2) and 51 cycles for read 2 (ATAC paired-end mate 2).

**Data processing.** FASTQ files for HepG2 (RRID: CVCL_0027) and GM12878 (RRID: CVCL_7526) were downloaded from ENCODE[53,95] (https://www.encodeproject.org, RRID: SCR_006793) under accession numbers ENCLB324GIU and ENCLB907YRF, respectively. The quality of these reads together with the newly generated ones for A172, M059 and LN229 was evaluated using FastQC (v.0.11.9, RRID: SCR_014583) and any adaptors were trimmed using the fastq-mcf command from ea-utils (v.1.1.2.779, RRID: SCR_005553). Reads were mapped to the hg38 genome using Bowtie2 (ref. [96]; v.2.4.4, RRID: SCR_016368), and duplicates were removed using Picard (v.2.27.1, RRID: SCR_006525). Peaks were called using MACS3 (ref. [87]; v.3.0.0b1, RRID: SCR_013291), and aggregate genome coverage was generated with the bamCoverage command of deepTools[85] (3.5.0, RRID: SCR_016366). The coverage of replicates was averaged using WiggleTools (v.1.2.11, RRID: SCR_001170). For the melanoma cell lines, fragment files were downloaded from the GEO under accession number GSE210745. Pseudobulk ATAC-seq profiles were generated per cell line as shown in the SCENIC+ tutorial[34] and peaks were called for each pseudobulk. From these, we selected MM001, MM029 and MM099 for our analysis. The resulting peaks from all eight cell lines (that is, the melanoma, GBM and ENCODE ones) were merged into a consensus set using the 'get_consensus_peaks' function of pycisTopic[36] and were used together with the eight coverage files to train the DeepCCL model. ChIP–seq peaks for JunB, c-Jun and c-Fos in MM099 were downloaded from the GEO using identifier GSE159965 and for ZEB1 through https://ucsctracks.aertslab.org/papers/enhancer_design/hg38/bb/tf_chip. In addition, we retrieved

ChIP–seq peaks for TEAD1 in MSTO cells and TEAD4 in SK-MEL-147 through ReMap[97] under accession numbers GSE68170 and GSE94488, respectively.

**Authentication of newly generated cell line data.** The identity of all cell lines on which new data were generated in this study was confirmed based on the coverage of single-nucleotide polymorphisms by the ATAC-seq data, compared to publicly available data on those cell lines (downloaded from ENCODE for A172 (ENCSR932KWJ) and M059J (ENCSR000EPG) and from the Sequence Read Archive for LN229 (SRX15782182)) using the tool NGScheckmate[98].

### Enhancer reporter assays

**In vivo validation of zebrafish enhancers.** A detailed protocol describing enhancer reporter plasmid cloning, egg microinjection and imaging is available on protocols.io via https://doi.org/10.17504/protocols.io.4r3l2pk2jg1y/v1.

*Enhancer reporter plasmid.* Synthetic enhancer DNA sequences were ordered from Twist Bioscience with constant flanking DNA sequences: 5′ end 'ATATACCCTCTAGAGTCGAA' and 3′ end 'GATTACCCTGTTAT CCCTAA'. These flanking regions were used for PCR amplification of the sequences using primers containing overhangs that are homologous to the target plasmid.

Fwd: 5′ TTAGGGATAACAGGGTAATCGCGAATTGGGTACCGGGC 3′
Rev: 5′ CTTTCAACAAGCCCGAAAGATCTTCTGGAAGCCTCCA GTGAATT 3′

KAPA HiFi HotStart ReadyMix (Roche) with primer and template concentrations of 300 µM and 0.1 ng µl$^{-1}$, respectively, were used and the PCR setup was: 95 °C for 3 min, 20 cycles of 98 °C for 20 min, 65 °C for 15 min, 72 °C for 15 min, and a final elongation step of 72 °C for 2 min. Next, the target plasmid, Tol2-ISceI-ZSP:EGFP;crya a:mCherry-ISceI-Tol2 (Addgene, 194518), was linearized by restriction digestion using BglII and XhoI (at 0.5 U µl$^{-1}$ and 1 U µl$^{-1}$, respectively, in rCutSmart buffer, with 7 µg plasmid, 2-h incubation at 37 °C), purified (NucleoSpin Gel & PCR cleanup kit, Macherey-Nagel) and combined with purified enhancer DNA fragments in a NEBuilder reaction (NEBuilder enzyme mix from New England Biolabs, 7 fmol linearized plasmid, 65 fmol DNA fragment, 45-min incubation at 50 °C). Then, 2.5 µl of the reaction was transformed into 20 µl of Stellar chemically competent bacteria (thaw cells on ice for 15 min, add plasmid, keep 30 min on ice, 45 s heat shock at 42 °C, 5 min on ice), which were incubated overnight at 37 °C on carbenicillin plates. Single colonies were then grown in 10 ml LB medium, and plasmids were isolated using QIAprep Spin Miniprep Kit (Qiagen) and eluted in 2 × 25 µl. All plasmid preps were sequenced using Sanger sequencing to confirm correct insertion in the target plasmid without sequence alterations. Enhancers C1, B1, F1, G1 and G2 contain point mutations that should not affect enhancer activity according to the model prediction.

*Egg microinjection.* Plasmid DNA together with *Tol2* mRNA were injected (at concentrations of 30 ng µl$^{-1}$ and 40 ng µl$^{-1}$, respectively) in one-cell stage zebrafish eggs (of the wild-type AB strain), and they were grown at 28.5 °C until 48 hpf. Zebrafish with successful injection were selected based on red fluorescence in the eye on an Olympus SZX16 widefield fluorescence microscope. Fish were anesthetized with 0.02% tricaine (MS-222 Ethyl 3-aminobenzoate methanesulfonate, Sigma) and mounted in 1% low-melting-point agarose (Invitrogen) on fluorodish (FD3510-100, World Precision Instruments). All zebrafish breeding was approved by the Ethical Committee for Animal Experimentation of KU Leuven (ECD-000) and all experiments were performed on embryos younger than 5 days after fertilization.

*Imaging.* Zebrafish embryos (48 hpf) were imaged for GFP fluorescence on a spinning-disk confocal microscope consisting of a Nikon Ti2 body

and Crest X-Light V2 spinning disk using a ×10 Plan Apo Lambda lens with numerical aperture of 0.45 and a ×20 Plan Apo Lambda lens with numerical aperture of 0.8. The Lumencor Spectra III using the green channel was used to excite GFP fluorescence, using excitation filter Semrock FF01-378/474/554/635/735-25 and dichroic mirror FF409/4 93/573/652/759-Di01-25×36. Semrock FF01-515/30-25 was used as an emission filter. NIS-element (RRID: SCR_014329) V6 software was used to control the microscope.

*Manual assessment of enhancer cell-type-specific enhancer activity.* For each task, enhancer reporter assays were performed in multiple zebrafish embryos (see figures for numbers) and in three replicates (that is, different injection days and clutches). Cell-type-specific activity of each enhancer was assessed using whole embryo fluorescence microscopy, independently by three researchers, at a single developmental timepoint (48 hpf). Only the cell-type specificity of each enhancer was assessed, not the level of activity, which is not possible due to the nature of our setup (that is, no control for the copy number of enhancer–reporters that are present in each cell—either episomaly or integrated in the genome).

### Reporting summary

Further information on research design is available in the Nature Portfolio Reporting Summary linked to this article.

### Data availability

Analysis data required for reproducing the findings in this paper is available at https://resources.aertslab.org/CREsted/manuscript_data. All CREsted models developed in this paper, and other legacy models, can be loaded through 'crested.get_model' or directly from https://resources.aertslab.org/CREsted. All raw and processed sequencing data generated in this study have been deposited in the GEO and are accessible through accession number GSE292617. This includes the OmniATAC-seq data of three human GBM cell lines, namely A172, M059J and LN229. Raw imaging data of the zebrafish enhancer reporter assays are available on EBI Biostudies via https://doi.org/10.6019/S-BIAD1962. The key resources table[99] (Supplementary Table 2) listing all resources needed to reproduce the results of this paper is available on Zenodo at https://doi.org/10.5281/zenodo.17791463 (ref. 99). The following publicly available datasets were used: Mouse cortex dataset (Zemke et al.[42]) downloaded from the GEO (GSE229169); Human PBMC dataset (De Rop et al.[84]) downloaded from the GEO (GSE194028); Zebrafish developmental dataset (Sun et al.[76]) downloaded from the GEO (GSE243256); Dataset of 100 enhancers tested in zebrafish (supplementary table of Sun et al.[76]; https://doi.org/10.1038/s41556-024-01449-0); Mouse brain pseudobulk BigWig (Zu et al.[19]) downloaded from the GEO (GSE246791); PAX5 ChIP–seq peaks downloaded from ENCODE (ENCFF827VVQ); PAX5 ChIP–seq B cells–BigWig track downloaded from ENCODE (ENCFF914QGY); EBF1 ChIP–seq B cells–peaks downloaded from ENCODE (ENCFF895MHN); EBF1 ChIP–seq B cells–BigWig track downloaded from ENCODE (EBF1 ChIP–seq B cells–BigWig track); POU2F2 ChIP–seq B cells–peaks downloaded from ENCODE (ENCFF934JFA); POU2F2 ChIP–seq B cells–BigWig downloaded from ENCODE (ENCFF803HIP); GATA3 ChIP–seq CD4+ T cells–peaks and BigWig downloaded from ChIP-Atlas (SRX4705120); RUNX1 ChIP–seq CD4+ T cells–peaks and BigWig downloaded from ChIP-Atlas (SRX1492212); ETS1 ChIP–seq CD4+ T cells–peaks and BigWig downloaded from ChIP-Atlas (SRX015825); CEBPA ChIP–seq CD14+ monocytes–peaks and BigWig downloaded from ChIP-Atlas (SRX097095); SPI1 ChIP–seq CD14+ monocytes–peaks and BigWig downloaded from ChIP-Atlas (SRX4001818); PAX5 ChIP–seq direct targets B cells (https://unibind.uio.no/factor/ENCSR000BHD.GM12878_female_B-cells_lymphoblastoid_cell_line.PAX5/); EBF1 ChIP–seq direct targets B cells (https://unibind.uio.no/factor/ENCSR000DZQ.GM12878_female_B-cells_lymphoblastoid_cell_line.EBF1/); GATA3 ChIP–seq direct targets CD4+ T cells (https://unibind.uio.no/factor/GSE76181.Jurkat_T-cells.GATA3/); RUNX1 ChIP–seq direct targets CD4+ T cells (https://unibind.uio.no/factor/GSE76181.Jurkat_T-cells.RUNX1/); ETS1 ChIP–seq direct targets CD4+ T cells (https://unibind.uio.no/factor/EXP000299.Jurkat_E6_1_T-cells.ETS1/); CEBPA ChIP–seq direct targets CD14+ monocytes (https://unibind.uio.no/factor/EXP000946.U937_adult_acute_monocytic_leukemia.CEBPA/); SPI1 ChIP–seq direct targets CD14+ monocytes (https://unibind.uio.no/factor/EXP047756.MDMmonocyte_derived_macrophages.SPI1/); Human genome (hg38; https://hgdownload.cse.ucsc.edu/goldenPath/hg38/bigZips); Mouse genome (mm10; https://hgdownload.cse.ucsc.edu/goldenPath/mm10/bigZips/); Zebrafish genome (danRer11; https://hgdownload.cse.ucsc.edu/goldenPath/danRer11/bigZips).

### Code availability

The CREsted package is available at https://github.com/aertslab/CREsted and https://crested.readthedocs.io and is stored[100] at https://zenodo.org/records/15045960. All computational analyses for the main figures can be found in https://github.com/aertslab/CREsted-paper and are stored[101] at https://zenodo.org/records/17791384. The following packages were used for data analyses; this information is also available in the key resource table[99] (Supplementary Table 2) accompanying this paper (https://zenodo.org/records/17791463): CREsted v.1.4.0, pybigtools v.0.2.0, ChromBPNet v.1.0, anndata v.0.11.3, Enformer, Borzoi, Keras v.3.0, TensorFlow v.2.19.0, PyTorch v.2.6.0, tfmodisco-lite v.2.2.1, tangermeme v.0.4.0, gReLU v.1.0.3, SnapATAC2 v.2.6.4, Harmony v.0.0.10, statsmodels v.0.14.4, Scipy v.1.16, Python Programming Language v.3, NIS-Elements, SCENIC+ v.1.0a, scatac_fragment_tools v.0.1.4, TF-MInDi v.1.0.0, pycisTarget v.1.1, memesuite-lite v.0.2, scanpy v.1.11.4, pyChromVAR v.0.0.4, epiAneufinder v.1.1.3, HyenaDNA, Nucleotide Transformer, Transformers v.4.54.1, PyTorch Lightning v.2.5.2, create_cisTarget_databases, WiggleTools v.1.2.11, MACS v.2 and v.3, pycisTopic v.2 and NGSCheckMate v.1.0.1.

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

## Acknowledgements

We thank the members of the Lab of Computational Biology for discussions and feedback on the CREsted package and paper. We thank early external users of the package for their feedback. We thank the staff of the VIB Data Core and the Flemish Supercomputing Center (Vlaams Supercomputer Centrum - VSC) for their support. An element of Fig. 4a was obtained from BioRender. We thank D. Koldere Vilain for the CREsted logo design. This research was funded in part by Aligning Science Across Parkinson's (ASAP-000430 and ASAP-025179) through the Michael J. Fox Foundation for Parkinson's Research (MJFF); ERC-AdG (101054387); CZI (DI2-0000000068); SBO (S005024N); FWO (G0I2722N EOS ID 40007513, G094121N and G044124N); and Foundation Against Cancer (2020-1396 and 2024-140) to S.A., an FWO PhD fellowship to N.K. (1SH6J24N), an FWO PhD fellowship to S.D.W. (1191323N) and an FWO senior postdoctoral fellowship to V.B. (1267625N). L.V.D.B.'s work was supported by VIB, KU Leuven (C14/22/132, IDN/22/012 and 'Opening the Future' Fund), the 'Fund for Scientific Research Flanders' (FWO-Vlaanderen; G0C1620N, G088523N and G026924N), the Thierry Latran Foundation, the 'Association Belge contre les Maladies neuro-Musculaires – aide à la recherché ASBL' (ABMM), the Muscular Dystrophy Association (MDA), Target ALS and the ALS Liga België (A Cure for ALS) and by the Generet Award for Rare Diseases.

## Author contributions

Conceptualization: N.K. and S.A. Computational analysis: N.K., S.D.W., C.H.B., V.K., S.D. and D.A. Data collection, processing and curation: N.K., S.D.W., V.K. and S.D. Experiments and sample preparation: S.D.W., V.B., K.S. and V.C. Resources: S.A. and L.V.D.B. Software implementation and testing: L.M., N.K., C.H.B., I.I.T., S.D.W., E.C.E., V.K. and G.H. Visualization: N.K., S.D.W., C.H.B. and V.K. Writing—original draft: N.K., S.D.W., C.H.B., V.K. and S.A. Writing—review and editing: N.K., S.D.W., C.H.B., V.K., S.A., E.C.E., L.M., S.D. and V.B.

## Competing interests

L.V.D.B. is head of the Scientific Advisory Board of Augustine Therapeutics (Leuven, Belgium) and is part of the Investment Advisory Board of Droia Ventures (Meise, Belgium). I.I.T. is currently associated with Illumina. The other authors declare no competing interests.

## Additional information

**Correspondence and requests for materials** should be addressed to Stein Aerts.

# Reporting Summary

## Statistics

For all statistical analyses, confirm that the following items are present in the figure legend, table legend, main text, or Methods section.

| n/a | Confirmed | |
|---|---|---|
| ☐ | ☒ | The exact sample size (*n*) for each experimental group/condition, given as a discrete number and unit of measurement |
| ☒ | ☐ | A statement on whether measurements were taken from distinct samples or whether the same sample was measured repeatedly |
| ☐ | ☒ | The statistical test(s) used AND whether they are one- or two-sided<br>*Only common tests should be described solely by name; describe more complex techniques in the Methods section.* |
| ☐ | ☒ | A description of all covariates tested |
| ☐ | ☒ | A description of any assumptions or corrections, such as tests of normality and adjustment for multiple comparisons |
| ☐ | ☒ | A full description of the statistical parameters including central tendency (e.g. means) or other basic estimates (e.g. regression coefficient) AND variation (e.g. standard deviation) or associated estimates of uncertainty (e.g. confidence intervals) |
| ☐ | ☒ | For null hypothesis testing, the test statistic (e.g. *F*, *t*, *r*) with confidence intervals, effect sizes, degrees of freedom and *P* value noted<br>*Give P values as exact values whenever suitable.* |
| ☒ | ☐ | For Bayesian analysis, information on the choice of priors and Markov chain Monte Carlo settings |
| ☒ | ☐ | For hierarchical and complex designs, identification of the appropriate level for tests and full reporting of outcomes |
| ☐ | ☒ | Estimates of effect sizes (e.g. Cohen's *d*, Pearson's *r*), indicating how they were calculated |

*Our web collection on statistics for biologists contains articles on many of the points above.*

## Software and code

Policy information about availability of computer code

| Data collection | *Provide a description of all commercial, open source and custom code used to collect the data in this study, specifying the version used OR state that no software was used.* |
|---|---|
| Data analysis | The CREsted package is available at https://github.com/aertslab/CREsted and https://crested.readthedocs.io and is stored at https://zenodo.org/records/15045960. All computational analyses for the main figures can be found in https://github.com/aertslab/CREsted-paper and are stored at https://zenodo.org/records/17791384. Following packages were used for data analyses, this information is also available in the key resource table accompanying this manuscript (https://zenodo.org/records/17791463). CREsted v. 1.4.0, pybigtools v. 0.2.0, ChromBPNet v.1.0, anndata v.0.11.3, Enformer, Borzoi, Keras v.3.0, TensorFlow v.2.19.0, PyTorch v.2.6.0, tfmodisco-lite v.2.2.1, tangermeme v.0.4.0, gReLU v. 1.0.3, SnapATAC2 v. 2.6.4, Harmony v. 0.0.10, statsmodels v0.14.4, Scipy v1.16, Python Programming Language v3, NIS-Elements, SCENIC+ v.1.0a, scatac_fragment_tools v.0.1.4, TF-MInDi v.1.0.0, pycisTarget v1.1, memesuite-lite v.0.2, scanpy v.1.11.4, pyChromVAR v.0.0.4, epiAneufinder v1.1.3, HyenaDNA, Nucleotide Transformer, Transformers v.4.54.1, PyTorch Lightning v.2.5.2, create_cisTarget_databases, WiggleTools v.1.2.11, MACS v2 and v3, pycisTopic v.2 and NGSCheckMate v.1.0.1. |

For manuscripts utilizing custom algorithms or software that are central to the research but not yet described in published literature, software must be made available to editors and reviewers. We strongly encourage code deposition in a community repository (e.g. GitHub). See the Nature Portfolio guidelines for submitting code & software for further information.

## Data

Policy information about availability of data

All manuscripts must include a data availability statement. This statement should provide the following information, where applicable:
- Accession codes, unique identifiers, or web links for publicly available datasets
- A description of any restrictions on data availability
- For clinical datasets or third party data, please ensure that the statement adheres to our policy

Analysis data required for reproducing the findings in this manuscript is available at https://resources.aertslab.org/CREsted/manuscript_data/. All CREsted models developed in this paper, and other legacy models, can be loaded through crested.get_model, or downloaded directly from https://resources.aertslab.org/CREsted/. All raw and processed sequencing data generated in this study have been deposited in NCBI's Gene Expression Omnibus and are accessible through GEO Series accession number GSE292617. This includes the OmniATAC-seq data of three human glioblastoma cell lines, namely A172, M059J, and LN229. Raw imaging data of the zebrafish enhancer reporter assays is available on EBI Biostudies using doi 10.6019/S-BIAD1962. The key resources table listing all resources needed to reproduce the results of this manuscript is available on Zenodo using the following doi: https://doi.org/10.5281/zenodo.17232414. Following publicly available datasets were used. Mouse cortex dataset (Zemke et al.) downloaded from GEO ( GSE229169); Human PBMC dataset (De Rop et al.) downloaded from GEO (GSE194028); Zebrafish developmental dataset (sun et al.) downloaded from GEO (GSE243256); Dataset of 100 enhancers tested in Zebrafish (Supplemental table of  https://doi.org/10.1038/s41556-024-01449-0); Mouse brain pseudobulk BigWig (zu et al) downloaded from GEO (GSE246791); PAX5 ChIP-seq peaks downloaded from Encode (ENCFF827VVQ); PAX5 ChIP-seq B cells - BigWig track downloaded from Encode ( ENCFF914QGY); EBF1 ChIP-seq B cells - peaks downloaded from Encode (ENCFF895MHN); EBF1 ChIP-seq B cells - BigWig track downloaded form Encode (EBF1 ChIP-seq B cells - BigWig track); POU2F2 ChIP-seq B cells - peaks downloaded from Encode (ENCFF934JFA); POU2F2 ChIP-seq B cells - BigWig downloaded from Encode (ENCFF803HIP); GATA3 ChIP-seq CD4+ T cells - peaks & BigWig downloaded from chip-atlas (SRX4705120); RUNX1 ChIP-seq CD4+ T cells - peaks & BigWig downloaded from chip-atlas (SRX1492212); ETS1 ChIP-seq CD4+ T cells - peaks & BigWig downloaded from chip-atlas (SRX015825); CEBPA ChIP-seq CD14+ monocytes - peaks & BigWig downloaded from chip-atlas (SRX097095); SPI1 ChIP-seq CD14+ monocytes - peaks & BigWig downloaded from chip-atlas (SRX4001818); PAX5 ChIP-seq direct targets B cells (https://unibind.uio.no/factor/ENCSR000BHD.GM12878_female_B-cells_lymphoblastoid_cell_line.PAX5); EBF1 ChIP-seq direct targets B cells (https://unibind.uio.no/factor/ENCSR000DZQ.GM12878_female_B-cells_lymphoblastoid_cell_line.EBF1/); GATA3 ChIP-seq direct targets CD4+ T cells (https://unibind.uio.no/factor/GSE76181.Jurkat_T-cells.GATA3/); RUNX1 ChIP-seq direct targets CD4+ T cells (https://unibind.uio.no/factor/GSE76181.Jurkat_T-cells.RUNX1/); ETS1 ChIP-seq direct targets CD4+ T cells( https://unibind.uio.no/factor/EXP000299.Jurkat_E6_1_T-cells.ETS1/); CEBPA ChIP-seq direct targets CD14+ monocytes (https://unibind.uio.no/factor/EXP000946.U937_adult_acute_monocytic_leukemia.CEBPA/); SPI1 ChIP-seq direct targets CD14+ monocytes (https://unibind.uio.no/factor/EXP047756.MDMmonocyte_derived_macrophages.SPI1/); Human genome (hg38- https://hgdownload.cse.ucsc.edu/goldenPath/hg38/bigZips/);  Mouse genome (mm10- https://hgdownload.cse.ucsc.edu/goldenPath/mm10/bigZips/)l; Zebrafish genome (danRer11 - https://hgdownload.cse.ucsc.edu/goldenPath/danRer11/bigZips/).

## Research involving human participants, their data, or biological material

Policy information about studies with human participants or human data. See also policy information about sex, gender (identity/presentation), and sexual orientation and race, ethnicity and racism.

| | |
|---|---|
| Reporting on sex and gender | NA |
| Reporting on race, ethnicity, or other socially relevant groupings | NA |
| Population characteristics | NA |
| Recruitment | NA |
| Ethics oversight | NA |

Note that full information on the approval of the study protocol must also be provided in the manuscript.

# Field-specific reporting

Please select the one below that is the best fit for your research. If you are not sure, read the appropriate sections before making your selection.

☒ Life sciences        ☐ Behavioural & social sciences        ☐ Ecological, evolutionary & environmental sciences

For a reference copy of the document with all sections, see nature.com/documents/nr-reporting-summary-flat.pdf

# Life sciences study design

All studies must disclose on these points even when the disclosure is negative.

| | |
|---|---|
| Sample size | No statistical method was used to predetermine sample size. Sample sizes were chosen based on the maximum amount of samples that were available for each analysis. For each analysis the sample size was sufficient to derive statistically meaningful results passing multiple testing procedures. |
| Data exclusions | No data was excluded in this study. |

| Replication | For the zebrafish enhancer reporter experiments multiple biological replicates were used. In general observed results could be replicated. We mentioned the number of replicates (and the number of positive samples) in the related figure. |
|---|---|
| Randomization | For the zebrafish enhancer reporter experiments the order in which enhancers were tested was randomized (i.e., enhancers designed for the same cell type were not all injected and imaged at the same experimental session). For the other analyses randomization was not relevant given that this is a study on a new machine learning method and no human judgment / subjective judgment was used to assess these models. Methods were compared using established and objective measurements. |
| Blinding | For the zebrafish enhancer reporter experiments enhancers were given a random name so that researchers were blind for the targetted cell type of each enhancer. For the other analyses performed in this study blinding was not relevant given that this is a study on a new machine learning method and no human judgment / subjective judgment was used to assess these models. Methods were compared using established and objective measurements. |

# Reporting for specific materials, systems and methods

We require information from authors about some types of materials, experimental systems and methods used in many studies. Here, indicate whether each material, system or method listed is relevant to your study. If you are not sure if a list item applies to your research, read the appropriate section before selecting a response.

## Materials & experimental systems

| n/a | Involved in the study |
|---|---|
| ☒ | ☐ Antibodies |
| ☐ | ☒ Eukaryotic cell lines |
| ☒ | ☐ Palaeontology and archaeology |
| ☐ | ☒ Animals and other organisms |
| ☐ | ☐ Clinical data |
| ☒ | ☐ Dual use research of concern |
| ☒ | ☐ Plants |

## Methods

| n/a | Involved in the study |
|---|---|
| ☐ | ☐ ChIP-seq |
| ☐ | ☐ Flow cytometry |
| ☐ | ☐ MRI-based neuroimaging |

## Eukaryotic cell lines

Policy information about cell lines and Sex and Gender in Research

| Cell line source(s) | Commercially available cell lines were used in this study available from ATCC. This is A172 cell line ( CRL-1620); M059J cell line (CRL-1620) and LN229 cell line (CRL-2611). |
|---|---|
| Authentication | We authenticated all cell lines used in this study based on the coverage of SNPs (ATAC-seq) compared to publicly available data of the same cell lines using NGScheckmate (https://github.com/parklab/NGSCheckMate). |
| Mycoplasma contamination | Cell cultures used for experiments providing data to this study were tested for myoplasm contamination and were found to be negative. |
| Commonly misidentified lines (See ICLAC register) | We used A172, as a model system for human glioblastoma as an example of mesenchymal like state. We validated the identity of this cell line using NGScheckmate. |

## Animals and other research organisms

Policy information about studies involving animals; ARRIVE guidelines recommended for reporting animal research, and Sex and Gender in Research

| Laboratory animals | We used Zebrafish (Danio Rerio) of the AB strain. Both male and female animals were used at age 48 hours post fertilization. |
|---|---|
| Wild animals | The study did not use wild animals. |
| Reporting on sex | Sex based analyses were not performed. Sex is not relevant for this study as we report an analysis software as main finding, therefore sex was not considered in the study design. |
| Field-collected samples | the study did not use field-collected samples. |
| Ethics oversight | All animal experiments were conducted according to the KU Leuven ethical guidelines and approved by the KU Leuven Ethical Committee for Animal Experimentation (approved protocol numbers ECD 000). |

Note that full information on the approval of the study protocol must also be provided in the manuscript.

# Clinical data

Policy information about clinical studies

All manuscripts should comply with the ICMJE guidelines for publication of clinical research and a completed CONSORT checklist must be included with all submissions.

| | |
|---|---|
| Clinical trial registration | *Provide the trial registration number from ClinicalTrials.gov or an equivalent agency.* |
| Study protocol | *Note where the full trial protocol can be accessed OR if not available, explain why.* |
| Data collection | *Describe the settings and locales of data collection, noting the time periods of recruitment and data collection.* |
| Outcomes | *Describe how you pre-defined primary and secondary outcome measures and how you assessed these measures.* |

# Plants

| | |
|---|---|
| Seed stocks | *Report on the source of all seed stocks or other plant material used. If applicable, state the seed stock centre and catalogue number. If plant specimens were collected from the field, describe the collection location, date and sampling procedures.* |
| Novel plant genotypes | *Describe the methods by which all novel plant genotypes were produced. This includes those generated by transgenic approaches, gene editing, chemical/radiation-based mutagenesis and hybridization. For transgenic lines, describe the transformation method, the number of independent lines analyzed and the generation upon which experiments were performed. For gene-edited lines, describe the editor used, the endogenous sequence targeted for editing, the targeting guide RNA sequence (if applicable) and how the editor was applied.* |
| Authentication | *Describe any authentication procedures for each seed stock used or novel genotype generated. Describe any experiments used to assess the effect of a mutation and, where applicable, how potential secondary effects (e.g. second site T-DNA insertions, mosiacism, off-target gene editing) were examined.* |

# ChIP-seq

## Data deposition

☐ Confirm that both raw and final processed data have been deposited in a public database such as GEO.

☐ Confirm that you have deposited or provided access to graph files (e.g. BED files) for the called peaks.

| | |
|---|---|
| Data access links<br>*May remain private before publication.* | *For "Initial submission" or "Revised version" documents, provide reviewer access links. For your "Final submission" document, provide a link to the deposited data.* |
| Files in database submission | *Provide a list of all files available in the database submission.* |
| Genome browser session<br>(e.g. UCSC) | *Provide a link to an anonymized genome browser session for "Initial submission" and "Revised version" documents only, to enable peer review. Write "no longer applicable" for "Final submission" documents.* |

## Methodology

| | |
|---|---|
| Replicates | *Describe the experimental replicates, specifying number, type and replicate agreement.* |
| Sequencing depth | *Describe the sequencing depth for each experiment, providing the total number of reads, uniquely mapped reads, length of reads and whether they were paired- or single-end.* |
| Antibodies | *Describe the antibodies used for the ChIP-seq experiments; as applicable, provide supplier name, catalog number, clone name, and lot number.* |
| Peak calling parameters | *Specify the command line program and parameters used for read mapping and peak calling, including the ChIP, control and index files used.* |
| Data quality | *Describe the methods used to ensure data quality in full detail, including how many peaks are at FDR 5% and above 5-fold enrichment.* |
| Software | *Describe the software used to collect and analyze the ChIP-seq data. For custom code that has been deposited into a community repository, provide accession details.* |

# Flow Cytometry

## Plots

Confirm that:

- [ ] The axis labels state the marker and fluorochrome used (e.g. CD4-FITC).
- [ ] The axis scales are clearly visible. Include numbers along axes only for bottom left plot of group (a 'group' is an analysis of identical markers).
- [ ] All plots are contour plots with outliers or pseudocolor plots.
- [ ] A numerical value for number of cells or percentage (with statistics) is provided.

## Methodology

| | |
|---|---|
| Sample preparation | *Describe the sample preparation, detailing the biological source of the cells and any tissue processing steps used.* |
| Instrument | *Identify the instrument used for data collection, specifying make and model number.* |
| Software | *Describe the software used to collect and analyze the flow cytometry data. For custom code that has been deposited into a community repository, provide accession details.* |
| Cell population abundance | *Describe the abundance of the relevant cell populations within post-sort fractions, providing details on the purity of the samples and how it was determined.* |
| Gating strategy | *Describe the gating strategy used for all relevant experiments, specifying the preliminary FSC/SSC gates of the starting cell population, indicating where boundaries between "positive" and "negative" staining cell populations are defined.* |

- [ ] Tick this box to confirm that a figure exemplifying the gating strategy is provided in the Supplementary Information.

# Magnetic resonance imaging

## Experimental design

| | |
|---|---|
| Design type | *Indicate task or resting state; event-related or block design.* |
| Design specifications | *Specify the number of blocks, trials or experimental units per session and/or subject, and specify the length of each trial or block (if trials are blocked) and interval between trials.* |
| Behavioral performance measures | *State number and/or type of variables recorded (e.g. correct button press, response time) and what statistics were used to establish that the subjects were performing the task as expected (e.g. mean, range, and/or standard deviation across subjects).* |

## Acquisition

| | |
|---|---|
| Imaging type(s) | *Specify: functional, structural, diffusion, perfusion.* |
| Field strength | *Specify in Tesla* |
| Sequence & imaging parameters | *Specify the pulse sequence type (gradient echo, spin echo, etc.), imaging type (EPI, spiral, etc.), field of view, matrix size, slice thickness, orientation and TE/TR/flip angle.* |
| Area of acquisition | *State whether a whole brain scan was used OR define the area of acquisition, describing how the region was determined.* |

Diffusion MRI    [ ] Used    [ ] Not used

## Preprocessing

| | |
|---|---|
| Preprocessing software | *Provide detail on software version and revision number and on specific parameters (model/functions, brain extraction, segmentation, smoothing kernel size, etc.).* |
| Normalization | *If data were normalized/standardized, describe the approach(es): specify linear or non-linear and define image types used for transformation OR indicate that data were not normalized and explain rationale for lack of normalization.* |
| Normalization template | *Describe the template used for normalization/transformation, specifying subject space or group standardized space (e.g. original Talairach, MNI305, ICBM152) OR indicate that the data were not normalized.* |
| Noise and artifact removal | *Describe your procedure(s) for artifact and structured noise removal, specifying motion parameters, tissue signals and physiological signals (heart rate, respiration).* |

Volume censoring | *Define your software and/or method and criteria for volume censoring, and state the extent of such censoring.*

## Statistical modeling & inference

Model type and settings | *Specify type (mass univariate, multivariate, RSA, predictive, etc.) and describe essential details of the model at the first and second levels (e.g. fixed, random or mixed effects; drift or auto-correlation).*

Effect(s) tested | *Define precise effect in terms of the task or stimulus conditions instead of psychological concepts and indicate whether ANOVA or factorial designs were used.*

Specify type of analysis: ☐ Whole brain ☐ ROI-based ☐ Both

Statistic type for inference | *Specify voxel-wise or cluster-wise and report all relevant parameters for cluster-wise methods.*

(See Eklund et al. 2016)

Correction | *Describe the type of correction and how it is obtained for multiple comparisons (e.g. FWE, FDR, permutation or Monte Carlo).*

## Models & analysis

n/a | Involved in the study
☐ | ☐ Functional and/or effective connectivity
☐ | ☐ Graph analysis
☐ | ☐ Multivariate modeling or predictive analysis

Functional and/or effective connectivity | *Report the measures of dependence used and the model details (e.g. Pearson correlation, partial correlation, mutual information).*

Graph analysis | *Report the dependent variable and connectivity measure, specifying weighted graph or binarized graph, subject- or group-level, and the global and/or node summaries used (e.g. clustering coefficient, efficiency, etc.).*

Multivariate modeling and predictive analysis | *Specify independent variables, features extraction and dimension reduction, model, training and evaluation metrics.*

