## [Peer Review File · Nature Methods]

CREsted: modeling genomic and synthetic cell type-specific enhancers across tissues and species

Corresponding Author: Professor Stein Aerts

Version 0:

Decision Letter:

16th Jul 2025

Dear Dr Aerts,

Your Article, "CREsted: modeling genomic and synthetic cell type-specific enhancers across tissues and species", has now been seen by 2 reviewers. As you will see from their comments below, although the reviewers find your work of considerable potential interest, they have raised a number of concerns. We continue to be interested in potentially publishing your manuscript in Nature Methods, but would like to evaluate your response to these concerns before we reach a final decision on publication.

We therefore invite you to revise your manuscript to address these concerns. We are committed to providing a fair and constructive peer-review process. Do not hesitate to contact us if there are specific requests from the reviewers that you believe are technically impossible or unlikely to yield a meaningful outcome.

- * include a point-by-point response to the reviewers and to any editorial suggestions
- * please underline/highlight any additions to the text or areas with other significant changes to facilitate review of the revised manuscript
- * address the points listed described below to conform to our open science requirements
- * ensure it complies with our general format requirements as set out in our guide to authors at www.nature.com/naturemethods
- * resubmit all the necessary files by using the link below to access your home page

Link Redacted

Note: This URL links to your confidential account and associated information about manuscripts you may have submitted, or that you are reviewing for us. If you wish to forward this email to co-authors, please delete this link.

We hope to receive your revised paper within 12 weeks. If you are substantially delayed, please let us know. In this event, we will still be happy to reconsider your paper at a later date so long as nothing similar has been accepted for publication at Nature Methods or published elsewhere.

OPEN SCIENCE REQUIREMENTS

REPORTING SUMMARY

When revising your manuscript, please update your reporting summary.

IMAGE INTEGRITY

EXTENDED DATA FIGURES

DATA AVAILABILITY

All novel DNA and RNA sequencing data, protein sequences, genetic polymorphisms, linked genotype and phenotype data, gene expression data, macromolecular structures, and proteomics data must be deposited in a publicly accessible database, and accession codes and associated hyperlinks must be provided in the "Data Availability" section.

For papers containing bioimaging data, we strongly recommend depositing the data to Bioimage Archive (<https://www.ebi.ac.uk/bioimage-archive/>). Associated accession codes and hyperlinks should be provided in the "Data Availability" section.

CODE AVAILABILITY

Please include a "Code Availability" subsection in the Online Methods which details how your custom code is made available.

Only in rare cases (where code is not central to the main conclusions of the paper) is the statement "available upon request" allowed (and reasons should be specified).

SUPPLEMENTARY PROTOCOL

To help facilitate reproducibility and uptake of your method, we ask you to prepare a step-by-step Supplementary Protocol for the method described in this paper. We [encourage authors to share their step-by-step experimental protocols](https://www.nature.com/nature-research/editorial-policies/reporting-standards#protocols) on a protocol sharing platform of their choice and report the protocol DOI in the reference list. Nature Portfolio's protocols.io is a free-to-use and open resource for protocols; protocols deposited onto protocols.io are citable and can be linked from the published article. More details can found at [protocols.io](https://www.protocols.io/help/publish-articles).

ORCID

Please do not hesitate to contact me if you have any questions or would like to discuss these revisions further. We look forward to seeing your revised manuscript and thank you for the opportunity to consider your work.

Best regards,
Lei

Lei Tang, Ph.D.
Senior Editor
Nature Methods

Reviewers' Comments:

Reviewer #1 (Remarks to the Author):

The authors present CREsted, a computational framework designed for modeling, analyzing, and designing cell type-specific enhancers using scATAC-seq data. This integrated tool encompasses preprocessing, model training, enhancer code analysis, and sequence design, and has been validated across diverse biological systems, including mouse cortex, human PBMCs, cancer cell lines, and zebrafish. Key strengths of CREsted include its superior performance compared to existing tools (e.g., ChromBPNet, gReLU), successful in vivo validation of synthetic enhancers, cross-species applicability, and impressive enhancer design capabilities. While the work addresses a critical challenge in regulatory genomics and provides a valuable open-source resource, several aspects require clarification or refinement to enhance its scientific rigor and impact.

Major Comments:

1. For a given peak, the model predicts cell-type-specific accessibility based on the same input sequence. Please clarify how this is achieved.
2. The two-step training process (pre-training on consensus peaks followed by fine-tuning on cell-type-specific peaks) should be explicitly compared to a direct one-step approach. Discuss the advantages in terms of robustness, generalization, and performance.
3. The claim of CREsted's superiority over ChromBPNet and gReLU in peak accessibility prediction requires explicit justification. Please address whether these improvements result from architectural innovations, training protocols, or region-level feature aggregation.
4. The choice of a 2,114 bp prediction window (with a 1,000 bp center) necessitates systematic validation, especially considering that canonical enhancer elements (like the ~200 bp IFNB1 enhanceosome) might be underrepresented. A parametric analysis evaluating the effects of different window sizes (e.g., 200 bp, 500 bp, 1,000 bp) on motif discovery sensitivity and transcription factor binding site (TFBS) localization precision would enhance methodological transparency. This should also assess the framework's ability to resolve nested TFBS arrangements within composite cis-regulatory modules.
5. Tools like chromVAR can identify cell-type-specific motifs. How do the results from CREsted compare with those from these

- existing tools in terms of similarities and differences? Additionally, what specific advantages does CREsted offer?
6. The manuscript would benefit from comparative benchmarking against genomic large language models (LLMs) such as DNABERT2, which are increasingly used for regulatory sequence analysis. A quantitative comparison of performance gains from CREsted's transfer learning approach versus direct applications of these foundational models would strengthen its value proposition. Including metrics on resource utilization (e.g., computational cost, memory requirements) would also inform users about trade-offs between CREsted's specialized models and general-purpose LLMs.
 7. The authors demonstrate that CREsted can accurately predict chromatin accessibility in previously unseen chromosomes, specifically chr9 and chr18. Extending this validation to additional chromosomes would be valuable. If prediction accuracy varies across chromosomes, further investigation into the factors driving these differences would be of great interest.
 8. While the authors attribute reduced nucleotide contribution correlations in Fig. 4g to non-sequence-mediated changes (e.g., copy number variations [CNVs]), this interpretation requires support. To substantiate this claim, the authors should mask amplified/deleted genomic regions in GBM datasets and recalculate nucleotide contribution correlations to control for CNV confounding effects.
 9. The enhancer design aspect of CREsted is particularly intriguing; however, additional information is needed regarding the number of enhancer sequences designed for each task and the extent of their functional validation. Such details are crucial for assessing the model's practical applications.
 10. The peak-scaling normalization procedure assumes that constitutive peaks (selected via a low Gini index across cell types) represent stable accessibility signals for technical bias correction. This approach relies on the assumption that low-variability peaks (e.g., housekeeping promoters) maintain universal accessibility across cell types without exhibiting cell-type-specific regulatory activity. The authors should provide additional empirical evidence or methodological justification for this assumption.

Minor Comments:

1. Define specialized terms (e.g., "seqlets," line 249) for non-specialist readers.
2. In Fig. S3b, clarify the meaning of the "scATAC curve" and its apparent superiority over CREsted.
3. In Fig. S12, define the x-axis and contextualize the description (lines 494–498). Explain the phenomenon observed in this figure.
4. In line 539, is it referring to "Fig. 5d"?
5. Consider moving the technical content (lines 593–603) to the Methods section for better coherence.

Reviewer #2 (Remarks to the Author):

The paper presents CREsted, a Python software package for modeling, interpreting, and designing enhancers from genomic sequences based on scATAC-seq data. It extensively demonstrates applications in various biological contexts (mouse cortex, human immune cells, cancer states, zebrafish development), with several new models like DeepBICCN2 and DeepZebrafish developed and analyzed. Designed tissue-specific enhancers for zebrafish were experimentally validated. The tool offers a comprehensive suite of tools for sequence-to-function modeling, enhancer code interpretation, and synthetic sequence design, with strong practical relevance to regulatory genomics. The code is well documented and provided several tutorials. This is a timely contribution to the field by facilitating such analyses. I have several suggestions for improvement before acceptance.

Main points:

1. CREsted is built around peak-based classification (predict topic probability) and regression (predict peak height) sequence-to-function modeling tasks. The authors should also provide some guidance regarding when classification is recommended and when regression is recommended. In addition, could the authors assess whether training on only peak regions leads to inflated predictions or false positives for sequences that are not accessible?
2. The enhancer code analysis (model interpretation) provides three options: integrated gradient, expected integrated gradient, and ISM. While intuitions are given for each method, there is no empirical evaluation to guide users' choice of these methods. Even though there is no perfect ground truth, the ability to recover known motifs and their active/repressive effects can be quantified systematically. For the expected integrated gradient method, random permutations of the sequence were used. Does it help to preserve dinucleotide frequency in the random permutation to make the sequence more in-distribution?
3. It will improve the usability of CREsted to make it easy for users to incorporate their own custom models. The authors are encouraged to provide a simple, flexible API and a tutorial to allow users to integrate custom architectures.
4. In the enhancer design experiments, the authors validate three designed sequences per condition. Could the authors clarify their criterion or strategy for choosing the three sequences to experimentally validate? Is it based on a quantitative score?

Minor point:

1. For enhancer design, could the authors elaborate on how the target prediction scores were chosen?

Reviewer #2 (Remarks on code availability):

I can install the package, but I have not attempted to reproduce the analysis.

Version 1:

Decision Letter:

Our ref: NMETH-A60436A

14th Nov 2025

Dear Dr. Aerts,

Thank you for your letter detailing how you would respond to the reviewer concerns regarding your revised Article "CREsted: Modeling genomic and synthetic cell type-specific enhancers across tissues and species" (NMETH-A60436A). I'm happy to say that we have decided, in principle, to publish your work in Nature Methods, pending minor revisions to satisfy the referees' final requests and to comply with our editorial and formatting guidelines.

In addition, we do not believe that a comprehensive comparison between CNN-based modeling and motif discovery approaches is necessary. In our view, a discussion at the text level, supported by appropriate citations, should be sufficient.

TRANSPARENT PEER REVIEW

ORCID

Author names using non-Roman characters

Nature Portfolio journals can support presentation of author names using non-Roman characters in the HTML version of the article. If you wish to, please include author names in parentheses after the Roman-character spelling; [see example online here](https://www.nature.com/articles/s44222-024-00258-2). Currently supported scripts are: Arabic, Chinese, Cyrillic, Devanagari, Greek, Hebrew, Hangul, Japanese and Persian. You will be asked to verify the rendering is correct at proof stage.

Best regards,
Lei

Lei Tang, Ph.D.
Senior Editor
Nature Methods

Reviewer #1 (Remarks to the Author):

The authors have addressed our concerns thoroughly and, overall, provide substantial results supporting most of their claims. We particularly appreciate the clearer description of the method's technical details, accompanied by comprehensive benchmarks and comparisons.

We also thank the authors for their detailed discussion of ChromBPNet and gReLU. The explanation of distinct design goals is helpful (ChromBPNet for single-cell-type footprinting, gReLU for large-scale modeling, and CREsted for cross-cell-type peak accessibility prediction). The performance comparison with gReLU on mouse cortex data, together with the clarifications on architectural and loss-function differences, adds valuable context.

That said, the current manuscript does not fully convey this nuanced perspective. The framing risks overstating the foundational novelty of CREsted by not explicitly situating it within the broader ecosystem of related CNN-based frameworks. We strongly recommend incorporating this balanced positioning directly into the manuscript to provide appropriate context.

Additional points:

1, Figure R3: Please extend the analysis to include more cell types to reduce dependence on any single target cell type. Also is it possible to add a training strategy that trains on all peaks (consensus + type-specific) without fine-tuning, and compare its performance to the current approaches?

2, In Section 1.5, all claims about advantages over classical motif discovery methods should be supported by robust analyses. For example, although the manuscript states that CREsted can discover cell type-specific motifs de novo, Fig. R10a shows the lowest sensitivity for CREsted; please reconcile this discrepancy with additional analyses or clearer interpretation. Also, please clarify what is meant by "learning contextual relationships between TFBS and syntactic rules," and provide a concrete validation plan (e.g., controlled synthetic tests, perturbation analyses, or interpretability diagnostics demonstrating learned higher-order dependencies).

Reviewer #2 (Remarks to the Author):

I thank the authors for the revision which have sufficiently addressed my prior comments.

Version 2:

Decision Letter:

4th Mar 2026

Dear Professor Aerts,

We very much appreciate your patience when we evaluate the revised version of your Article, "CREsted: modeling genomic and synthetic cell type-specific enhancers across tissues and species". I am pleased to inform you that this paper has now been accepted for publication in Nature Methods. The received and accepted dates will be 2nd Apr 2025 and 4th Mar 2026. This note is intended to let you know what to expect from us over the next month or so, and to let you know where to address any further questions.

A short editorial summary that will appear in the table of contents is copied below. Please let us know if there is any error in it in 1 working day:

"CREsted is an efficient and user-friendly toolbox for analysis, modeling and design of cell type-specific enhancers across diverse species."

Over the next few weeks, your paper will be copyedited to ensure that it conforms to Nature Methods style. Once your paper is typeset, you will receive an email with a link to choose the appropriate publishing options for your paper and our Author Services team will be in touch regarding any additional information that may be required. It is extremely important that you let us know now whether you will be difficult to contact over the next month. If this is the case, we ask that you send us the contact information (email, phone and fax) of someone who will be able to check the proofs and deal with any last-minute problems.

Authors may need to take specific actions to achieve compliance with funder and institutional open access mandates.

If your research is supported by a funder that requires immediate open access (e.g. according to [a href="https://www.springernature.com/gp/open-science/plan-s-compliance"> Plan S principles](https://www.springernature.com/gp/open-science/plan-s-compliance) or the [a href="https://www.springernature.com/gp/open-science/us-federal-agency-compliance"> NIH public access policy](https://www.springernature.com/gp/open-science/us-federal-agency-compliance)) then you should select the gold OA route, and we will direct you to the compliant route where possible. Because authors warrant under our subscription licensing terms that they haven't committed to licensing any version of their article under a licence inconsistent with the terms of our agreement – including the applicable embargo period – publication under the subscription model isn't suitable for authors whose funders require no embargo.

If you have any questions about our publishing options, costs, Open Access requirements, or our legal forms, please contact

ASJournals@springernature.com

Please feel free to contact me if you have questions about any of these points. Thank you very much for publishing your paper at Nature Methods!

Best regards,

Lin Tang, PhD
Senior Editor
Nature Methods

** Visit the Springer Nature Editorial and Publishing website at http://editorial-jobs.springernature.com?utm_source=ejp_NMeth_email&utm_medium=ejp_NMeth_email&utm_campaign=ejp_Nmeth for more information about our career opportunities. If you have any questions please click [here](mailto:editorial.publishing.jobs@springernature.com). **

to the original author(s) and the source, provide a link to the Creative Commons license, and indicate if changes were made. In cases where reviewers are anonymous, credit should be given to 'Anonymous Referee' and the source. The images or other third party material in this Peer Review File are included in the article's Creative Commons license, unless indicated otherwise in a credit line to the material. If material is not included in the article's Creative Commons license and your intended use is not permitted by statutory regulation or exceeds the permitted use, you will need to obtain permission directly from the copyright holder. To view a copy of this license, visit <https://creativecommons.org/licenses/by/4.0/>

Introduction

We would like to thank the reviewers for their valuable comments and positive feedback, particularly regarding the framework, analyses and *in vivo* validation experiments. Many of the suggestions were a great addition to our manuscript, making our results more robust and clear, and placing them in a broader context. Most of the feedback resulted in textual changes and/or additional analyses. In our responses, we have clearly indicated what elements we added to the manuscript and where these changes were applied.

Reviewer #1 (Remarks to the Author)

The authors present CREsted, a computational framework designed for modeling, analyzing, and designing cell type-specific enhancers using scATAC-seq data. This integrated tool encompasses preprocessing, model training, enhancer code analysis, and sequence design, and has been validated across diverse biological systems, including mouse cortex, human PBMCs, cancer cell lines, and zebrafish. Key strengths of CREsted include its superior performance compared to existing tools (e.g., ChromBPNet, gReLU), successful *in vivo* validation of synthetic enhancers, cross-species applicability, and impressive enhancer design capabilities. While the work addresses a critical challenge in regulatory genomics and provides a valuable open-source resource, several aspects require clarification or refinement to enhance its scientific rigor and impact.

Major Comments

1.1. For a given peak, the model predicts cell-type-specific accessibility based on the same input sequence. Please clarify how this is achieved.

Cell type-specific accessibility prediction is the central aspect of CREsted models. We will therefore discuss both the way the data is preprocessed and how the models are trained to give a clear overview how this is achieved.

During data preprocessing for peak regression, one shared peak set is used to cover all cell types. If a certain genomic region, that is a peak in one or more types, is closed in other types, those will get a zero-value assigned, or a very low value depending on the number of cut sites in that region. This preprocessing allows us to create a 2D matrix of shape `#cell_types x #regions`, with the level of accessibility as values. This sets up a standard machine learning multi-label regression problem, where for one input, here the one-hot encoded list of the nucleotides in the region, multiple outputs, here the peak heights for each cell type obtained from the corresponding BigWig files, are predicted.

During training, the model learns to associate different motifs and enhancer codes for each of the target classes, depending on the given batch of sequences that is being processed. With the fine-tuning step, models are biased more strongly towards cell type-specific enhancer codes rather than ubiquitous ones. Therefore, after training, it is able to analyze a given input sequence using the different enhancer codes of each of the output classes, and form a cell type-specific prediction based on how well that sequence matches the learned enhancer codes. For instance, in fig. S3c-e (Fig. R1) we highlight example region

predictions, with their region contribution scores shown in Fig. 2F, and here for AiE2428m highlighted for the two top scoring classes in (e). We see that the model predicts for both SstChodl and to a lesser extent Sst for this enhancer (c, top), following the accessibility profile (d, top). The similar enhancer code between these types causes this dual prediction (e). In a more exceptional example, we highlight how in the same input sequence the model (DeepBICCN2) correctly predicts peak accessibility in multiple sequences, as that sequence contains multiple overlapping unrelated enhancer codes (Fig. R2).

Fig. R1 (fig. S3c-e): Cell type-specific model predictions.

(c) scATAC peak heights for three example enhancer regions. (d) Predictions for three example enhancer regions. (e) Contribution scores for the AiE2428m SstChodl enhancer.

Fig. R2: Dual enhancer code region example.

(a) Example test set region (chr18:10309537-10311651) from mouse cortex data (Zemke *et al.*, 2023) scored by DeepBICCN2 model. (b) Contribution scores for top two predicted classes, highlighting differently learned enhancer codes.

Since this is a critical aspect of understanding how our models work, we expanded the “Model training” methods section by adding:

“All CREsted models are multi-label models, predicting from one input sequence the accessibility over a range of output classes. During training, the model learns to associate different motifs and enhancer codes for each of the target classes, depending on the given batch of sequences that is being processed. Therefore, after training, it is able to analyze a given input sequence using the different enhancer codes of each of the output classes, and form a cell type-specific prediction based on how well that sequence matches the learned enhancer codes.”

Additionally, this principle has already been repeatedly addressed in the field, and we refer to the relevant literature both in the introduction and in the discussion.

1.2. The two-step training process (pre-training on consensus peaks followed by fine-tuning on cell-type-specific peaks) should be explicitly compared to a direct one-step approach. Discuss the advantages in terms of robustness, generalization, and performance.

We chose the two-step approach with the aim of approaching a more complete and accurate characterization of cell type-specific enhancers. For most applications it is biologically more meaningful to learn what makes a cell type-specific enhancer rather than what makes a genomic region accessible in general. However, the small fraction of cell type specific genomic regions (compared to all consensus peaks) limits the amount of training samples, reducing the ability of the model to learn and making it more prone to overfit on the training data. The two-step approach addresses this challenge by first training on all consensus peaks and later fine-tuning, over a few epochs and a low learning rate, on cell type specific regions to learn what makes cell type-specific enhancers. Furthermore, having the model pretrain on the set of consensus peaks over all cell types has the benefit of it still being functional on non-specific regions. The gene locus plots in Fig. 2e, where promoter and non-peak regions are scored correctly by the model, highlight this functionality. The fine-tuning step ensures increased performance on the cell-type specific enhancers (Fig. 2d, fig. S3b).

To assess the functionality of a direct one-step approach, we performed additional analyses. As this comment relates to comment 1.7, we trained multiple models on the Zemke *et al.* mouse cortex data in different chromosomal training-validation-test splits for increased robustness, with three different strategies: only training on all consensus peaks (“basemodel”), fine-tuning on cell type-specific peaks (“fine-tuned model”), and directly training on cell type-specific peaks without pretraining (“scratch model”). The resulting performances on a held out test set of cell type-specific sequences immediately indicate the expected trend: the scratch models perform significantly worse than the fine-tuned models over the different chromosomal splits, and even than the basemodels, on held out sets of cell type-specific peaks (Fig. R3).

Fig. R3 (fig. S3a): Comparison of model performances from the 10 differentially chromosomally split basemodels, fine-tuned models and scratch models.

We additionally used gene locus scoring plots, where we score entire gene loci along a sliding window, to investigate how the scratch models would handle non-peak and generally accessible regions. We scored the same gene locus from Fig. 2e, and found that for the targeted cell type, L6CT, the scratch model performed surprisingly well (Fig. R4, Table R1). We additionally scored that locus for a cell type (Astrocytes) where the *Chsy3* gene was not expressed. Here we saw that the scratch models are less performant. This suggests that models trained from scratch are less suited to handle off-target and non-peak regions.

Fig. R4 (fig. S3c-d): Gene locus prediction comparison of model training strategies.

(c) Gene locus prediction plots of the *Chsy3* locus, using the average predictions from all differentially chromosomally split basemodels, fine-tuned models, and scratch models, for L6CT (left) and Astrocytes (Astro) (right). (d) Pearson and Spearman correlations between the predicted accessibility tracks of the *Chsy3* gene locus from the average predictions of the basemodels, fine-tuned models and scratch models for L6CT and Astro.

Finally, we highlight an example region where we calculated the average contribution scores from all base-, fine-tuned, and scratch models over their different chromosomal splits. While this is only one example, it shows that the scratch models give more noisy explanations compared to the other two options (Fig. R5).

Fig. R5 (fig. S3e): Contribution score comparison for example region with different model training strategies. Contribution scores were the average over all chromosomal splits per strategy. Region coordinates: chr13:92952861-92953624 (mm10).

These results indicate that models that do not go through a round of pretraining before being fine-tuned on cell type-specific peaks have a loss in predictive performance on both peaks and non peaks, regardless of cell type-specificity. Pretraining on all peaks therefore increases the robustness, generalization and performance of our models. We added these important findings to fig. S3, and refer to them in the main text under the “CREsted provides detailed insights into enhancer codes of mouse cortical cell types” results section:

“To further scrutinize the CREsted model training pipeline, we assessed the performance of CREsted models directly trained on cell type-specific peaks without pretraining on all consensus peaks. Such models have worse performance metrics than both fine-tuned and base models and have worse gene locus predictions (fig. S3).”

1.3. The claim of CREsted's superiority over ChromBPNet and gReLU in peak accessibility prediction requires explicit justification. Please address whether these improvements result from architectural innovations, training protocols, or region-level feature aggregation.

First, we want to highlight that CREsted is a framework that offers many features other than model training; including preprocessing steps and enhancer code analyses that are not provided elsewhere. Furthermore, we want to note that the design goals of the three frameworks are somewhat different: ChromBPNet focuses on chromatin accessibility profile prediction in a single cell type at a time and has a strong focus on footprinting analyses, while CREsted predicts chromatin accessibility levels across peaks and many cell types. gReLU has particular features for large scale models, something that is less present in CREsted, but is overall conceptually more closely related to CREsted compared to ChromBPNet. Because of these differences in design goals and feature sets it is difficult to claim superiority on the level of the frameworks.

Given that CREsted and gReLU are conceptually more similar, we already compared the performance of both frameworks, in terms of Pearson correlation of predicted and actual chromatin accessibility levels across test regions, on mouse cortex data from Zemke *et al.* (Fig. 2d) and observed superior performance from CREsted. However, like the reviewer rightly suggests, we did not address where this performance gain comes from.

Technically, all three frameworks use similar default model architectures that are all based on the dilated convolutional neural network (CNN) architecture as first proposed by ChromBPNet. One feature that does differ compared to both gReLU and ChromBPNet is the fine-tuning on cell type-specific regions. As mentioned in 1.2, we reasoned that applying this step increases our performance on the relevant cell type-specific enhancer codes, which are a key focus and novelty in our work. But even with the non-fine-tuned model, we outperform gReLU, as shown in Fig. 2D.

gReLU uses a similar architecture for multiclass training, but implements a different loss function. We reached out to the developers on GitHub to ask them the best training setup for training a multiclass model, and followed their advice. We provided the exact same (peak-scaled) target values as we used in CREsted, and trained a model following their notebook. Therefore, we are confident to claim that CREsted models outperform gReLU models on multiclass peak regression problems in a default setting. The added fine-tuning further distinguishes CREsted.

To further assess where the performance gains come from, we implemented the gReLU architecture and loss function in CREsted, and trained models using their loss function and architecture, their loss function and our architecture, and their architecture and our loss function, inside of CREsted using the exact same mouse cortex data and training setup. We found that both the architecture and loss function gReLU uses seem to give a worse performance compared to our default settings (Fig. R6).

Fig. R6: Comparison between CREsted and gReLU.

Comparison of the class-wise predictive performance of CREsted (DeepBICCN2_base) and gReLU models on held-out test set sequences for the entire test set (left) and cell type-specific test set peaks (right). Model naming: gReLU (gReLU model trained in gReLU package), gReLU_architecture_gReLU_loss (model with gReLU architecture and loss trained inside of CREsted), gReLU_architecture_CREsted_loss (model with gReLU architecture and CREsted loss trained inside of CREsted), CREsted_architecture_gReLU_loss (model with CREsted architecture and gReLU loss trained inside of CREsted).

We also assessed gReLU's preprocessing, and how it calculates region-level feature aggregation. We did not use this in our previous analyses since we wanted to take the same values used in CREsted for a fair comparison focusing on differences in architecture and loss function. We noticed that gReLU's region-level feature aggregation differs from defaults in CREsted. Importantly, gReLU provides a BigWig generation function that does not produce CPM-normalized tracks. In a multi-class setting, this will result in a model whose predicted values can not be compared across classes. (Fig. R7). Furthermore, CREsted provides additional peak-scaling factors to make values across classes and samples comparable. The combination of both CPM-normalization (to account for number of cells and sequencing depth per cell type (i.e., model class)) and peak scaling (to account for the difference in the number biological peaks per cell type) is important to obtain comparable chromatin accessibility values in a multi-class setting. Therefore, training multi-cell type models in gReLU and fully relying on its preprocessing will not give comparable target labels, as it lacks both CPM-normalization and additional peak normalization.

Fig. R7: Region-level feature aggregation comparison between gReLU and CREsted.

Comparison of region-level feature aggregation using gReLU and CREsted (left), with CREsted peak-scalars added to gReLU peaks (right). Different cell types are highlighted in different colors.

Comparing CREsted to ChromBPNet is more difficult due to the fact that ChromBPNet employs a single output model, meaning that capturing enhancer codes for multiple cell types requires training multiple models (amounting to hundreds of models to study complex biological systems like the brain). In this context it must again be stated that ChromBPNet has an important different design goal. While CREsted-models are designed for multi-class prediction of chromatin accessibility peak *levels*, ChromBPnet is designed for chromatin accessibility *profile* predictions and for that reason employs functionalities that are specifically needed for that outcome, like Tn5 bias correction and non-peak data augmentation. We nevertheless already compared CREsted with ChromBPNet by combining multiple separate ChromBPNet models into one ensemble model in Fig. 4c. We updated these plots to include both the base-model and the fine-tuned DeepCCL model. These results already showed that the ensemble model performed worse on all peaks than the DeepCCL base model, and than the fine-tuned model on specific peaks. To add on to this comparison, we compared those results to a default CREsted model (as DeepCCL used a Poisson loss function and a slightly different architecture that was optimized for the biological system studied in that section), and to a model with the exact ChromBPNet architecture but without chromatin accessibility profile prediction, Tn5 bias correction, and non-peak data augmentation but still using the ChromBPNet loss function (MSE) trained inside of the CREsted framework (Fig. R8). Both the CREsted model with default parameters and the ChromBPNet-like model significantly outperform the ChromBPnet ensemble mode (which still does Tn5 bias corrected chromatin accessibility profile prediction). For this reason, we hypothesize that the improved performance of CREsted, in this task, is due to two factors: first the fact that, due to its multiclass nature, CREsted models see the chromatin accessibility level across multiple cell types (CREsted default vs CREsted ChromBPNet architecture + Loss) and second the fact that ChromBPNet is designed for predicting chromatin accessibility profiles may limit its performance to predict chromatin accessibility levels across cell types (CREsted ChromBPNet Architecture vs ChromBPNet ensemble).

Fig. R8: Comparison between CREsted and ChromBPNet.

Comparison of the class-wise predictive performance of CREsted models (in blue DeepCCL_base, with a poisson loss function and a slightly updated architecture, and in green the default dilated CNN with cosine-MSE loss function) with ChromBPNet ensembles (in orange, an ensemble fully trained inside of ChromBPNet, and in purple, an ensemble of models trained inside of CREsted with the ChromBPNet architecture and loss function).

Furthermore it must be noted that a user with sufficient knowledge could theoretically come up with CREsted architectures, loss functions and training tricks and implement them in gReLU and ChromBPNet and obtain similar performance, and vice versa. This is however not the intent of our benchmark, as we simply followed default settings provided by all packages, for a fair, user-targeted benchmark.

Based on the aforementioned reasoning, we decided not to expand further on this in our manuscript. To do a complete ablation study comparing architectures, loss functions and training procedures, and making strong conclusions from that, is out of scope for the purpose of our work. In that scenario this should be done thoroughly, across multiple datasets, while coordinating with the developers of the other packages.

1.4. The choice of a 2,114 bp prediction window (with a 1,000 bp center) necessitates systematic validation, especially considering that canonical enhancer elements (like the ~200 bp IFNB1 enhanceosome) might be underrepresented. A parametric analysis evaluating the effects of different window sizes (e.g., 200 bp, 500 bp, 1,000 bp) on motif discovery sensitivity and transcription factor binding site (TFBS) localization precision would enhance methodological transparency. This should also assess the framework's ability to resolve nested TFBS arrangements within composite cis-regulatory modules.

We agree that we did not present any validation/evaluation experiments for this parameter choice. The input size choice of 2,114 bp is based on the ChromBPNet model architecture (Pampari & Shcherbina *et al.*, 2025) on which the default CREsted model architecture, in case of peak regression models, is inspired. An in-depth evaluation of this metric would therefore be relevant.

We decided to train six separate models with the following input sizes: 132 bp, 264 bp, 518 bp, 1057 bp, 2,114 bp and 4,228 bp. Given that the default CREsted model makes use of dilated convolution (i.e., a number of bp are skipped for each convolution operation) where the dilation factor (a number indicating the number of bp that are skipped for each convolution operation) increases exponentially with each additional convolutional layer (fig. R9a), smaller input sizes can not be used without altering the model architecture (or performing padding, see further). For instance, an input size of 518 bp would result in a convolutional layer that has zero as the first dimension after eight convolutional layers (which is the default number of convolutional layers used by CREsted regression models). To circumvent this problem we used two solutions. For the first solution we altered the number of convolutional layers so that the dimension of the layers always remained above zero (by selecting the maximal amount of convolutional layers where this constraint was met). This solution changes the number of trainable parameters (fig. R9b). For the second solution we still made use of eight convolutional layers but the input to each layer was padded with zeros in case it was too small. This operation is implemented in Keras and can be activated by using the “*same*” option for the *padding* parameter. This solution does not change the number of trainable parameters (fig. R9b). As such, we trained twelve (six different input sizes all with and without padding) peak regression models to predict chromatin accessibility in PBMC types and fine-tuned all models on cell type-specific peaks.

Across most metrics the model with input size 2,114 bp performed best (fig. R9c). This is not surprising given that the default CREsted model architecture is optimized for this input size. More importantly, all models performed reasonably well with a Pearson correlation coefficient of predicted versus actual cell type-specific chromatin accessibility ranging from 0.43 to 0.63. (fig. R9b). Next, we assessed whether the input size of the model has any influence on the location of predicted transcription factor binding sites (TFBS; i.e., seqlets). As we expected, most seqlets are located within a window of ~300 bp (fig. R9d) around the ATAC peak’s summit. This suggests that many predicted TFBS are biologically relevant, rather than technical artifacts that depend on the choice of the input size. Next, to assess motifs found by the different models we made use of TF-MInDi, a python package that was recently developed in the lab for identification of motifs and their instances from deep learning models (<https://github.com/aertslab/TF-MInDi>). Using TF-MInDi we were able to identify the major transcription factor families for all PBMC types (fig. R9e) and this was independent from the model input size (fig. R9f). Finally, we tested whether model input size influences the explainability of small and compact enhancers. For this purpose, we generated explanations for all input sizes (using the architecture shrinking approach) for the IFNB1 enhanceosome region for the dendritic cells class. All models produce similar explanations, highlighting the known TFBS (fig. R9e). We note that, the MEF TFBS, in the flank of the main enhanceosome region, only appears with input sizes ≥ 528 . In conclusion, we observe a limited effect of altering the input size parameter for training CREsted models and this based on both performance metrics and motif discovery.

Fig. R9 (fig. S9). Model input size benchmark on PBMC dataset.

(a) Schematic representation of convolutional neural network with dilation. (b) Number of trainable parameters versus input size with and without applying padding. (c) Performance metrics versus input size with and without applying padding. (d) Distribution of seqlet locations versus input size without (top) and with (bottom) padding. (e) tSNE of 6,000 seqlets based on seqlet similarities colored based on TF-family calculated using TF-MInDi. (f) tSNE as in (e) split by input size and whether padding was used (padding) or not (param. shrink). (g) Contribution scores of

the IFNB1 enhanceosome (hg38 chr9:21,076,963-21,079,077, zoomed to center 250 bp) for the dendritic cell class for models with different input sizes without using padding. Param. parameters

We have added Figure R9 as a supplementary figure to the manuscript and added following paragraph to the results section “A CREsted human PBMC model captures validated TFBS”:

“In this context, we investigated whether our default model input size of 2,114 bp has an influence on peak accessibility predictions and motif identification. This input size covers most enhancers, given that many fit within a single nucleosome depleted region¹². The user can decide to change this parameter, taking into account that the default architecture has to be modified for different input sizes (Fig. S9a-b). We observed that changing the input sizes has limited effect on model performance (Fig. S9c) and motif identification (Fig. S9d-g).”

1.5. Tools like chromVAR can identify cell-type-specific motifs. How do the results from CREsted compare with those from these existing tools in terms of similarities and differences? Additionally, what specific advantages does CREsted offer?

We agree with the reviewer’s statement that motif enrichment analysis tools, like chromVAR, can also identify cell type-specific motifs. Using deep neural networks has additional benefits compared to classical motif discovery that we would like to highlight. This includes:

- **Finding motif instances.** The discovery of enriched motifs in sets of co-accessible regions can also be performed with classical motif discovery methods, such as pycistarget (developed in our lab) or chromVAR (see below where we compare motif discovery results of such methods with CREsted models). However, after a motif is found significantly enriched, the identification of the true positive instances of this motif, within a critical subset (or leading edge) of the input set, has always been a difficult challenge. CNNs such as CREsted, through seqlet calling, provide a solution to this problem. In addition, CREsted models provide motif instances *before* motif discovery, meaning that TFBS predictions can be obtained even if not enough occurrences are found that would lead to a significant enrichment over the input set.
- **Finding cell type-specific motifs *de novo*.** Even though our current transcription factor motif databases are close to complete (also see below), for organisms that are well studied. For other, more rarely studied organisms that are evolutionarily distant from human, fly or mouse, the *de novo* discovery aspect might be more beneficial.
- **Predicting the effect of non-coding variation.** CREsted can be used to predict the effect of non-coding variation. These effects are harder to detect using position weight matrices, which would require scanning the sequence resulting in many false positives (Wasserman & Sandelin, 2004, Nature Reviews Genetics).
- **Transcription factor binding sites are seen in the context of other binding sites.** Enhancers consist of specific combinations of transcription factor binding sites. Because CREsted processes full enhancer sequences at once, these contextual relationships can be learnt by the model, which again leads to an improved accuracy of TFBS prediction.
- **Learning syntactic rules.** Syntactic rules like distance dependent relationships between individual transcription factor binding sites, binding affinity see (fig. S3e and f) and copy number of individual binding sites can be learnt and extracted.
- **Scoring of unseen sequences/genomes.** As we showed in figure 2e, CREsted can be used to score the genome of other species for which potentially no data is available but still relatively accurately recover chromatin accessibility peaks. Repeating this same analysis with transcription factor binding motifs alone results in large excesses of false positives (Wasserman & Sandelin 2004).

- **Design of cell type specific enhancers.** The rules learnt by the model allows the model to be used as a biological oracle to design cell type-specific enhancers (see Figure 6).

It is important to highlight these advantages and applications that go far beyond classical motif enrichment, and for this reason we thank the reviewer for bringing this to our attention. Accordingly we made the following textual changes in the manuscript (discussion):

“Sequence-to-function models employing deep neural networks have emerged as promising methods to overcome this limitation and have specific advantages over conventional methods making use of motif scanning or motif enrichment. These advantages include: (1) Identifying TFBS locations across peaks, (2) Finding cell type-specific motifs de novo for biological systems where current motif databases are not complete, (3) Predicting the effect of non-coding variation, (4) learning contextual relationships between TFBS and syntactic rules, (5) scoring unseens sequences/genomes with high accuracy and (6) design of cell type-specific enhancers.”

To illustrate that CREsted learns *de novo* motifs underlying cell type-specific enhancers and that those same motifs can also be found using motif enrichment tools we applied pycisTarget (Bravo González-Blas & De Winter *et al.*, 2023) and pyChromVar (Schep *et al.*, 2017) to cell type-specific peaks of peripheral blood mononuclear cell (PBMC) types and compared enriched motifs to those found through CREsted. First, we calculated pairwise similarities between the union of motifs identified using all three techniques, using TomTom (Schreiber *et al.* 2025) and performed tSNE dimensionality reduction on this matrix to visualize the motif similarities. Subsequently we clustered the matrix using the Leiden algorithm and made use of the SCENIC+ motif-to-TF annotations (Bravo González-Blas & De Winter *et al.*, 2023) to annotate each cluster based on TF families (using a majority vote of motifs identified using pycisTarget, for which motif-to-TF annotations are available; Fig. R10a). For all families, similar motifs were found using both CREsted and either of the motif enrichment tools (Fig. R10a). Note that for the homeodomain;POU (HD;POU), CEBP, paired box (PAX) families only CREsted and pycisTarget found motifs and homeodomain (HD) motifs were only found using chromVAR. Notably, two clusters annotated as homeodomain were found, one consisting of motifs solely from chromVAR (HD) and another of motifs from both CREsted and pycisTarget (CEBPB). The second cluster consisted solely of CEBP motifs, for this reason this cluster is most likely mis-annotated based on the SCENIC+ motif-to-TF collection (Fig. R10a). Furthermore, we identified several clusters (e.g., 22, 26 and 27), that were not annotated based on the motif-to-TF annotation database and that were specific to both pycisTarget and chromVAR.

pychromVAR allows for the inference of motif activities per cell type. Next, we compared the resulting motif activity matrix to the pattern contribution score matrix as generated by CREsted (Fig. 3d). The pychromVAR profiles appear noisier and show more binary-like patterns of activity—with monocytes and dendritic cells in one group and natural killer, B and T cells in the other—suggesting lower resolution in distinguishing cell-type-specific effects (Fig. R10b).

Fig. R10 (fig. S14): Motif identification benchmark comparing CREsted and motif enrichment analysis tools on PBMC data.

(a) Left: tSNE of motifs found enriched using pycisTarget and chromVAR and patterns found using TF-MoDISco based on contribution scores from deepPBMC. Motifs and patterns are clustered based on their motif similarity

calculated using TomTom. And annotated at cluster level based on the SCENIC+ motif-to-TF annotation database. Right: Representative motif for each TF family/cluster. **(b)** Heatmap of average motif activity per cell type calculated using chromVAR. **(c)** Precision-recall curve on comparing pyChromVAR identified seqlets and ChIP-seq peaks in the top 1,000 peaks for the corresponding cell type. Thresholding is done on the 'motif match' score. Average precision (AP) and recall (AR) over the thresholds are indicated in the legend. **(d)** Average ChIP peak height of different sets of proposed TFBS for CEBPA in the top 1,000 most-specific CD14+ monocytes identified through CREsted (left) and pyChromVar (right). **(e)** Precision-recall table for a set of TFs comparing CREsted and pyChromVar motif identification overlapping with ChIP-seq peaks at the lowest threshold in both settings.

Following our ChIP-seq analysis from Figure 3c and e, we additionally investigated if hits called by pyChromVar also overlap with ChIP-peaks, and how well that works compared with CREsted. pycisTarget does not provide exact motif instance, or 'seqlet', locations, hence it was left out for the comparison. We found that hits from pyChromVar also overlap with ChIP-seq peaks, but have overall a worse precision and recall than the same analysis through CREsted (Fig R10c-e). Across all tested TFs, CREsted-derived seqlets achieved more than 2x higher recall (0.71 versus 0.33), at similar or higher precision (0.42 versus 0.37).

In conclusion, like classical motif enrichment tools, CREsted can identify TF motifs underlying cell type-specific regions. However, unlike classical tools CREsted has increased performance --- in terms of both precision and recall --- for identifying biologically relevant TFBS. Furthermore, deep learning methods have many other advantages over motif enrichment analysis as we summarised above.

We incorporated this analysis in the manuscript under the section "A CREsted human PBMC model captures validated TFBS"

"Finally, we compared the motifs identified through CREsted to those identified using classical motif enrichment analysis tools (pycisTarget and pyChromVar); CREsted identified similar motifs compared to motif enrichment analysis tools however it has an overall higher precision and recall for identifying TFBS (fig. S14)."

1.6. The manuscript would benefit from comparative benchmarking against genomic large language models (LLMs) such as DNABERT2, which are increasingly used for regulatory sequence analysis. A quantitative comparison of performance gains from CREsted's transfer learning approach versus direct applications of these foundational models would strengthen its value proposition. Including metrics on resource utilization (e.g., computational cost, memory requirements) would also inform users about trade-offs between CREsted's specialized models and general-purpose LLMs.

We thank the reviewer for this suggestion and agree that benchmarking against these models strengthens the presented conclusions. Current genomic language models (gLMs) indeed claim to achieve high scores on tasks like enhancer and promoter detection after fine-tuning. As explained in Della-Torre et al. (2024), there are two primary methods to use gLMs for downstream tasks: probing, which involves training a small model using extracted embeddings from the gLM as input, and fine-tuning, where a new head layer is added at the end of the gLM and the full model is further trained on the specific task.

Recent benchmarks cast doubt on gLMs' performance on relevant tasks in regulatory genomics. Tang, Somia, Yu and Koo (2025) trained small models on embeddings extracted from these self-supervised models to predict regulatory tasks (probing), like MPRA-derived CRE activity and ChIP-seq-derived transcription factor binding. To put their performance into context, they were compared against the same

architectures trained on embeddings from supervised models and on one-hot encoded sequences. The models trained on self-supervised embeddings performed the worst out of these three categories, even being outperformed by the models trained on simple one-hot encoded DNA sequences.

To validate whether these shortcomings still apply when fine-tuning the models rather than probing, we used two prominent gLMs, Nucleotide Transformer (Dalla-Torre *et al.*, 2025) and HyenaDNA (Nguyen *et al.*, 2023) as the basis for fine-tuning in a similar manner as the Borzoi model in figure 5. We also attempted to use DNABERT2, but were unable to run it, as its code has not been updated to be compatible with recent versions of Triton and PyTorch.

The results of fine-tuning these two models to predict chromatin accessibility in the mouse motor cortex can be seen in Fig. R11. We also added this finding to fig. S23. Both models perform worse than the default DilatedCNN or fine-tuned Borzoi models. Strikingly, the Nucleotide Transformer's performance is much worse than all other alternatives, especially on the cell-type specific peaks. This is despite it having over 480 million parameters, as opposed to 6 million for DeepBICCN2 and 170 million for the Borzoi model. HyenaDNA performs better than the Nucleotide Transformer, but still underperforms compared to training from scratch or fine-tuning from a supervised model. Therefore, we conclude that our base and Borzoi-finetuned training approaches presented in CREsted outperform fine-tuning from genomic language models.

As the Nucleotide Transformer is also a much larger model than the Borzoi or CNN/Hyena-based models, using it for fine-tuning or evaluation requires significantly more computational resources than other models. While training or fine-tuning on the full training set for one epoch took between 15 and 30 minutes for the HyenaDNA, DilatedCNN, and Borzoi models, the Nucleotide Transformer took over 24 hours to complete one epoch, owing to a combination of a required lower batch size due to GPU memory limits and slower processing of each batch. The HyenaDNA model is smaller, coming in at 3.3M parameters, and as such took similar time to train for one epoch as the base DilatedCNN models. However, it still required many epochs to fine-tune fully while underperforming the DilatedCNN models trained from scratch, tilting the resource requirements in favor of training from scratch. As such, we conclude that fine-tuning from the unsupervised models required at least as many computational resources, and in some cases much more, than training from scratch or fine-tuning from Borzoi, at no benefit to performance.

Fig. R11 (fig. S23). Language model fine-tuning comparison.

Pearson correlation metric for predicting cell type-specific chromatin accessibility profiles for fine-tuned CREsted model, fine-tuned Borzoi model and fine-tuned genomic language models.

We added this analysis under the section: “CREsted-trained models outperform large, pre-trained models on cell type-specific chromatin accessibility predictions”.

“Next to large supervised models like Borzoi, self-supervised genomic language models (gLMs) have also been trained to serve as foundations to fine-tune for downstream tasks like chromatin accessibility prediction. To assess how these models compare to fine-tuning Borzoi, we fine-tuned two gLMs - HyenaDNA⁸¹ and the Nucleotide Transformer⁸² - to predict cell type-specific chromatin accessibility in the mouse motor cortex, following the same approach as fine-tuning Borzoi. Unfortunately, this resulted in worse-performing models for both gLMs as compared to either fine-tuning Borzoi or training CREsted models from scratch (fig. S23), despite in some cases much higher parameter counts and corresponding computational cost of training.”

1.7. The authors demonstrate that CREsted can accurately predict chromatin accessibility in previously unseen chromosomes, specifically chr9 and chr18. Extending this validation to additional chromosomes would be valuable. If prediction accuracy varies across chromosomes, further investigation into the factors driving these differences would be of great interest.

The choice for the chromosomal split could indeed be seen as a potential biasing factor, as some chromosomes have increased homology and could therefore introduce data leakage between the training

and held out data. Therefore, showing the independence of model performance on chromosomal split is important.

Using the different models trained in comment 1.2 on the Zemke *et al.* mouse motor cortex data using alternating chromosomal splits, we assessed the effect of the split on performance, and on enhancer code interpretation. We observed robust predictive performance across chromosomal splits based on three of our key metrics (Fig. R3).

It could be argued that the different test sets resulting from the different chromosomal splits are not easily comparable one-to-one. We therefore also compared the learned enhancer codes of the models. We investigated a set of 171 *in vivo* validated enhancers (Ben-Simon *et al.*, 2025) and compared the inferred nucleotide-level contribution scores. We obtained the contribution scores for the target cell types with all models, and calculated the Spearman correlation between those scores across models to investigate the extent to which the enhancers are interpreted similarly by models with different chromosomal splits. In Fig. R12, we show a heatmap of these average correlation scores across models, indicating that the 10 fine-tuned and 10 base models show strong correlation regardless of chromosomal split. As a negative control, the worse “scratch” models show a lower correlation. To avoid unwanted biases, this set of validated enhancers was left out of the model training set in all splits.

Fig. R12 (fig. S3b): Nucleotide contribution comparison across chromosomal splits.

Heatmap of nucleotide contribution score Spearman correlations calculated a set of 171 *in vivo* validated enhancers (Ben-Simon *et al.*, 2025) for their target class. A set of 10 chromosomal splits were used per model type.

Moreover, we checked whether our models had a similar performance on regions split by chromosome. We used our set of fine-tuned models with different chromosomal splits and scored cell type-specific regions split per chromosome. We used for each chromosome the model that had that chromosome in its test set. The results indicate that the performance is consistent across chromosomes (Fig. R13), therefore we do not suspect any underlying biological factors influencing our chromosomal splits.

Fig. R13: Chromosome prediction comparison.

Pearson correlation of predicted values versus peaks (log-transformed) on the Zemke *et al.* mouse cortex data. The values for each chromosome were obtained from a model with that chromosome in its test set.

These findings highlight that the choice of chromosomal split does not affect model performance. Regardless of the split, we find robust performance across different model training strategies. We added these findings to fig. S3b, and refer to them in the main text under the “CREsted provides detailed insights into enhancer codes of mouse cortical cell types” results section:

“We additionally investigated the effect of chromosomal train, validation and test split on model performance, and found consistent robustness over different splits (fig. S3b).”

1.8. While the authors attribute reduced nucleotide contribution correlations in Fig. 4g to non-sequence-mediated changes (e.g., copy number variations [CNVs]), this interpretation requires support. To substantiate this claim, the authors should mask amplified/deleted genomic regions in GBM datasets and recalculate nucleotide contribution correlations to control for CNV confounding effects.

We thank the reviewer for raising this important point and we appreciate the opportunity to clarify the interpretation presented in the text:

“To enhance this comparison²⁹ and enable its abstraction away from non-sequence-mediated changes, such as copy number variations, we used DeepGlioma to compare nucleotide contributions with those obtained by DeepCCL focusing on the MES-like topics (Fig. 4g).”

To clarify, our intention was not to attribute the reduced nucleotide contribution correlations themselves to non-sequence-mediated changes (e.g., CNVs). Instead, we used the comparison of nucleotide contributions (derived from the DeepCCL and DeepGlioma models) as a method to enhance the comparison between cell lines and biopsies, specifically to abstract away or mitigate the influence of

features other than local sequence-level determinants that can affect the raw accessibility/prediction scores. We showed in a recent study that this was the most robust comparison method (Hecker & Kempynck et al., 2025). In other words, while the raw accessibility/prediction scores might be susceptible to confounding effects like CNVs, the nucleotide contributions focus on the sequence context that drives model prediction, allowing for a more direct comparison of the sequence-level regulatory logic learned by the models.

To further support our interpretation, we performed an additional analysis to assess the potential impact of CNVs. Specifically, we used *epiAneufinder* (Ramakrishnan et al., 2023) to infer CNVs from our scATAC-seq data and categorized genomic regions as either CNV or neutral, focusing on the two main MES-like topics (Topic8 and Topic21). We then recalculated and visualized the correlations per group (all, neutral and CNV regions) for:

1. Accessibility (A172 coverage vs. Topic8/Topic21 coverage),
2. Prediction scores (DeepCCL A172 vs. Topic8/Topic21 coverage),
3. Contribution scores (DeepCCL A172 contribution scores vs. DeepGlioma Topic8/Topic21 contribution scores).

This analysis revealed that accessibility/prediction scores correlations show a greater variance and often a clear reduction when comparing neutral to CNV regions, while correlations of nucleotide contribution scores between models remain more stable across all groups (Fig. R14). These findings support our initial hypothesis that comparing models via their sequence-driven contribution scores provides a more robust metric to compare cell states between samples. Having said that, we should note that *epiAneufinder* provides an approximate inference of CNVs from sparse scATAC-seq profiles and cannot capture the full complexity of genomic copy number alterations. A comprehensive CNV analysis across modalities is both underpowered and outside the scope of the current work. Nonetheless, this additional test is consistent with our interpretation that nucleotide-level contribution scores provide a robust, sequence-centric measure of cell state similarity that abstracts away variability introduced by non-sequence-mediated changes.

Fig. R14 (fig. S17): Impact of inferred CNVs on correlation analysis in MES-like Topics and the A172 cell line. Spearman correlations are shown for three comparisons across all regions (light colors), neutral regions (medium colors), and CNV regions (dark colors). The three comparisons are: (1) A172 cell coverage accessibility vs. Topic accessibility, (2) DeepCCL A172 prediction scores vs. Topic accessibility, and (3) DeepCCL A172 contribution scores

vs. DeepGlioma Topic contribution scores. The pie chart shows the relative proportion of All, Neutral, and CNV windows used in the analysis.

We have revised the text under the section “CREsted identifies high similarity between mesenchymal-like enhancer codes in cancer” to clarify this point and include the new supporting evidence using *epiAneufinder*:

“To enhance this comparison²⁹ and reduce the influence of non-sequence-mediated changes, such as copy number variations (CNVs), we used DeepGlioma to compare nucleotide-level contribution scores with those obtained by DeepCCL, focusing on the MES-like topics (Fig. 4g). We reasoned that this approach provides a more sequence-centric measure of similarity between models and cell states. Consistent with this interpretation, analysis of scATAC-inferred CNV regions using epiAneufinder (Ramakrishnan et al., 2023) showed that contribution score correlations between models remain stable across CNV and neutral regions, whereas accessibility/prediction correlations vary more strongly with CNV state (fig. S17).”

1.9. The enhancer design aspect of CREsted is particularly intriguing; however, additional information is needed regarding the number of enhancer sequences designed for each task and the extent of their functional validation. Such details are crucial for assessing the model's practical applications.

We thank the reviewer for this positive feedback.

Regarding the number of enhancer sequences designed for each task: For each task we designed 200 sequences (this number was reported in the legend of Fig. 6). From these, 5 sequences were sampled at random per task and the nucleotide contributions were visually inspected. Based on this manual inspection we tested 3 out of 5 randomly sampled sequences for each task.

Based on this comment (and the comment from reviewer 2) we added the following paragraph to the material and methods section

“Enhancers selection for experimental validation

From the 200 generated sequences for each task (see above), 5 sequences were sampled at random per task and the nucleotide contributions were visually inspected. Based on this manual inspection 3 out of 5 randomly sampled sequences were tested for each task.”

Regarding the extent of functional validation: For each task, the enhancer reporter assays were performed in multiple zebrafish embryos (numbers are reported in figure 6) and in three replicates (i.e., different injection days and clutches). The cell type-specific activity of each enhancer was assessed using whole embryo fluorescence microscopy at a single developmental time point (48 hours post fertilization). We were not able to assess differences in strengths between enhancers, due to the fact that we did not control for the copy-number of enhancer-reporters that are present in each cell (either episomally or integrated in the genome).

Based on this comment we added the following paragraph to the material and methods section:

“Manual assessment of enhancer cell type-specific enhancer activity

For each task, enhancer reporter assays were performed in multiple zebrafish embryos (see figure for numbers) and in three replicates (i.e., different injection days and clutches). Cell type-specific activity of each enhancer was assessed using whole embryo fluorescence microscopy, independently by three researchers, at a single developmental time point (48 hpf). Only the cell-type specificity of each enhancer was assessed, not the level of activity, which is not possible due to the nature of our setup (i.e., no control for the copy-number of enhancer-reporters that are present in each cell --- either episomally or integrated in the genome).”

1.10. The peak-scaling normalization procedure assumes that constitutive peaks (selected via a low Gini index across cell types) represent stable accessibility signals for technical bias correction. This approach relies on the assumption that low-variability peaks (e.g., housekeeping promoters) maintain universal accessibility across cell types without exhibiting cell-type-specific regulatory activity. The authors should provide additional empirical evidence or methodological justification for this assumption.

The selection of peaks for peak-scaling normalization inherently uses low-variability peaks based on the low Gini index filtering, to avoid the scenario of including regions with cell-type specific regulatory activity. In other words, we use a data-driven approach to find strongly accessible regions across all cell types, under the assumption that these should be of equal peak height. By taking a large set of regions for this purpose, the effect of a potential more cell type-specific region will be negligible. This approach was previously validated in Johansen & Kempynck *et al.*, 2025, where we first showed that this peak-scaling improves the prediction of cell type-specific enhancer activity, and second that it nearly perfectly matches other peak normalization methods, such as the 'ReadsInTSS' option in ArchR.

To further empirically validate our peak scalars, we compare the CREsted-obtained peak scalars to the average accessibility of a set of housekeeping promoters in fig. S1D. Even though the CREsted peak-normalization has no notion of the chosen regions' regulatory functions, it still nearly perfectly follows the expected trend of these housekeeping promoters. To further quantify this, we added a comparative scatter plot showing a near-perfect correlation (Pearson correlation of 0.99) for normalization weights obtained from regions selected based on their low Gini index and normalization weights obtained on housekeeping promoters (Fig. R15). Additionally, we observed that 84.33% of the housekeeping promoter set (n=3,970) overlaps (with a minimum of 50 % of bp) with a region in the peak set used in CREsted normalization (n=48,308). And vice versa, 6.84% of CREsted normalization peaks overlap with the housekeeping promoter set. Thus, using filtering based on the Gini index regions can be selected with housekeeping-like features, that is regions with low cell type-specific chromatin accessibility. Regarding whether these regions confer cell type-specific activity, we assessed the expression of genes regulated by the housekeeping promoters and observed constant expression levels across cell types (fig. S2c). This shows that our data-driven method does capture housekeeping promoters in its set, as well as a much larger set of other generally accessible housekeeping-like regions and that these regions in general do not regulate cell type-specific activity. We are therefore convinced that our peak-scaling method is robust and not biased towards cell type-specific regulatory systems. We added Fig. R15 to fig. S1e for further validation of our peak scalars.

Fig. R15 (fig. S1e): Comparison of CREsted peak-normalization weights and the weights obtained from looking at housekeeping promoter regions.

Minor Comments

m1.1. Define specialized terms (e.g., "seqlets," line 249) for non-specialist readers.

We moved the introduction of the "seqlet" term to the next results section since it added unnecessary confusion here. We added "motif instances" to the new introductory sentence to clarify the meaning of the term. We went over the manuscript to find additional unexplained specialist terms but did not find any.

m1.2. In Fig. S3b, clarify the meaning of the "scATAC curve" and its apparent superiority over CREsted.

In Fig. S3b, we compare different methods on their predictive capabilities of *in vivo* validated enhancer function. To achieve this, we started from a 2D matrix with predicted values, where the rows each represent one of the 171 validated enhancers, and the columns the different cell types. First, we used the predictions of our models to make this 2D matrix, simply by generating predictions for the enhancer

sequences. We then also took the actual peak height values, both from mouse and human scATAC-seq data (Zemke *et al.*, 2023), to make this 2D matrix, both for the regular and the peak-scaled data. This is following the method used in a previous study where we looked at a much larger set of enhancers, also including non-functional ones (Johansen & Kempynck *et al.*, 2025).

From these matrices, we made the precision-recall curves per method. The scATAC curves represent how well solely using accessibility values over the cell types predicts enhancer activity. Indeed, both the regular and peak-scaled differential ATAC signal outperforms CREsted models. In the previously mentioned study, we showed that for single modality methods, on-target, functional enhancers, are best predicted by scATAC data, while CREsted models show their strength in identifying non-functional enhancers, even with an accessibility peak. Therefore, the scATAC curves being better was expected and previously observed. The models aim to approach those values, as they are the ground truth the models are trained on. We added the scATAC curve to put the performance of CREsted models into perspective, and additionally to further validate our peak scaling approach.

We updated our methods section on “CREsted predictions on validated mouse cortex enhancers” by adding:

“The scATAC curves represent how well solely using accessibility values over the cell types predicts enhancer activity. In the previously mentioned study, we showed that for single modality methods, on-target, functional enhancers, are best predicted by scATAC data, while CREsted models show their strength in identifying non-functional enhancers, even with an accessibility peak. CREsted models aim to approach those values, as they are the ground truth the models are trained on. We added the scATAC curve to put the performance of CREsted models into perspective, and additionally to further validate our peak scaling approach.”

m1.3. In Fig. S12, define the x-axis and contextualize the description (lines 494–498). Explain the phenomenon observed in this figure.

On the x-axis there are several types of fine-tuned Borzoi models, which were trained using different strategies than the main strategy presented in the paper. For all strategies, they are compared on their held-out test set sequences (on both the full and the cell type-specific set).

In panel a, we extended on Fig. 5b of the manuscript. We compared our reference, the double fine-tuned Borzoi, to a Borzoi model fine-tuned once solely on the specific peaks, and to two Borzoi models with different architectures (CNN tower only), both single and double fine-tuned. We observe that none of these alternatives reach the performance level of the reference.

In panel b, we investigated whether the difference in train-validation-test set splits between the original Borzoi model and CREsted for mouse cortex data influenced our findings. We observed the same results with the Borzoi splits in the fine-tuning steps as we did in the main figure using the CREsted split, indicating that potential train-test set leakage is not affecting our observed performance.

Fig. R16 (fig. S19): Model performance of additional Borzoi transfer learned models.

Violin plots showing Pearson correlation values between the log_{1p}-transformed test sets (all consensus peaks and cell type-specific peaks respectively) and corresponding predictions, calculated across the peaks for each cell type. **(a)** Comparison of the best-performing transfer learned model (Double fine-tuned Borzoi) with other transfer learning approaches (Specific fine-tuned: only fine-tuning on specific peaks; Borzoi CNN: CNN-tower only pre-trained Borzoi model) on the r per cell type ($n=19$). **(b)** Correlation values per cell type ($n=19$) for models trained and evaluated analogously to fig. 5b, but using a train/validation/test split based on the Borzoi folds instead (same model terminology from Fig. 5b).

We updated the text in the main manuscript to further clarify this by moving and extending the following sentence:

“We additionally observed that fine-tuning solely on the cell type-specific peaks showed worse performance compared to the double fine-tuning approach, as was the case for a more compact CNN tower-only version of Borzoi as a pre-trained section (Fig. S19a).”

The results of panel b were already described adequately in the discussion. We additionally increased the font size in this figure (Fig. R16) to make it more readable and extended the figure caption to clearly describe the different model types.

m1.4. In line 539, is it referring to "Fig. 5d"?

We thank the reviewer for noticing this mistake. We updated it in the manuscript.

m1.5. Consider moving the technical content (lines 593–603) to the Methods section for better coherence.

We agree that this content fits better in the Methods section and moved it to the subsection: “Enhancer design - Optimization function”.

Reviewer #2 (Remarks to the Author):

The paper presents CREsted, a Python software package for modeling, interpreting, and designing enhancers from genomic sequences based on scATAC-seq data. It extensively demonstrates applications

in various biological contexts (mouse cortex, human immune cells, cancer states, zebrafish development), with several new models like DeepBICCN2 and DeepZebrafish developed and analyzed. Designed tissue-specific enhancers for zebrafish were experimentally validated. The tool offers a comprehensive suite of tools for sequence-to-function modeling, enhancer code interpretation, and synthetic sequence design, with strong practical relevance to regulatory genomics. The code is well documented and provided several tutorials. This is a timely contribution to the field by facilitating such analyses. I have several suggestions for improvement before acceptance.

Main points

2.1. CREsted is built around peak-based classification (predict topic probability) and regression (predict peak height) sequence-to-function modeling tasks. The authors should also provide some guidance regarding when classification is recommended and when regression is recommended. In addition, could the authors assess whether training on only peak regions leads to inflated predictions or false positives for sequences that are not accessible?

Topic classification vs peak regression

Topic classification and peak regression each have their appropriate use cases. In most settings, where a user has access to annotated BigWigs per cell type, we recommend peak regression. The modeling of peak heights instead of binary classifying regions is more closely related to biology and avoids running topic modeling. Not only does that require annotating your topics to relevant cell types/groups, but it can also get quite memory intensive, especially on large datasets.

Topic classification is appropriate to use on scATAC datasets where cell type definitions are not easy to acquire. For example, in our DeepGlioma models on the Wang *et al.* biopsy data containing many different cancer states. In this case, it is biologically more meaningful for a more continuous representation of the data rather than binary cell type labels. For this reason, topic modeling and topic classification was more appropriate in this case. We further clarified this in the “CREsted identifies high similarity between mesenchymal-like enhancer codes in cancer” results section, by adding to the final paragraph:

“Moreover, we highlight a relevant use-case for CREsted topic classification models with the DeepGlioma model, trained on scATAC-seq data that, due to its more continuous nature of biological cell states, is not easily annotated by cell type and therefore benefits from topic modeling.”

Non-peak false positives

The reviewer raises a valuable point here concerning the lack of non-ATAC-seq peak regions in our training set, and a concern for false positive predictions on closed regions. Due to the multiclass nature of our models and the fact that for each cell type only a fraction of all consensus peaks are accessible, the CREsted models are in fact already trained on closed regions (although not closed across all cell types in the dataset). We can already observe in the gene locus scoring plots of Fig. 2e that our models handle closed regions well. We additionally investigated more quantitatively how CREsted models predict closed peaks (that can still be open in other types, we took peaks from cell types with aggregate signal equal to zero) in the Zemke *et al.* mouse cortex dataset with the DeepBICCN2 base and fine-tuned model (Fig. R17b and c). We observe that these models on average predict values that are close to zero with a mean

absolute error (MAE) of 0.45 and 0.32 for base and fine-tuned respectively) and therefore do not make strong false positive predictions.

To assess whether the addition of non-peak regions to our training set would benefit model performance in general and on closed regions, we trained a CREsted model on the consensus peak set of +/- 550k peaks in mouse motor cortex cell types and extended this peak set with 550k randomly selected non-exonic non-peaks (“Nonpeakextended” models). We then further finetuned that model to the same set of cell type-specific regions used for DeepBICCN2. Overall, we do not see an improvement of performance for both predicting peaks (Fig. R17a) and non-peak regions (Fig. R17b and c).

Fig. R17 (fig. S4): Non-peak region model benchmarking.

(a) Scatter plot of predictions vs peak heights over all classes for the DeepBICCN2 base model, DeepBICCN2 model, Non-peak extended base model, and Non-peak extended fine-tuned model. Regions are held-out test set regions from the filtered cell type-specific test set. (b) Closed region predictions over all classes on closed regions from the different models. MAE: mean absolute error. (c) Density distributions of predictions over all classes for closed and open regions from the different models.

We therefore show that, if a dataset has a substantially large peak set and enough cell types, CREsted models see enough closed regions during training by taking in peaks that are open in only a subset of the cell types. In such a scenario, adding non-peak regions does not help to improve general model performance and performance on closed regions. There are however scenarios where adding non-peaks is a relevant strategy (e.g. both ChromBPNet and gReLU do this). For example, in cases of a limited peak

set, for instance in organisms with small genomes, or a limited set of cell types, adding non-peak regions can extend the training set and potentially improve performance.

We added this important analysis to the result section titled: “CREsted provides detailed insights into enhancer codes of mouse cortical cell types” and added Fig. R17 as fig. S4:

“Lastly, we doubled our peak set with a set of non-peak regions to assess whether this would increase general and non-peak predictive performance, but report no improvements (fig. S4). Even within the set of consensus peaks, a large fraction of target values are near zero because most peaks are open only in a subset of all types, this seems to suffice and not necessitate the need of adding additional non-peak regions.”

2.2. The enhancer code analysis (model interpretation) provides three options: integrated gradient, expected integrated gradient, and ISM. While intuitions are given for each method, there is no empirical evaluation to guide users’ choice of these methods. Even though there is no perfect ground truth, the ability to recover known motifs and their active/repressive effects can be quantified systematically. For the expected integrated gradient method, random permutations of the sequence were used. Does it help to preserve dinucleotide frequency in the random permutation to make the sequence more in-distribution?

We thank the reviewer for this remark. We had indeed not extensively studied the similarities and differences between our model interpretation methods. To first compare their similarities, we took a set of 171 *in vivo* validated enhancers (Ben-Simon *et al.*, 2025) and scored them with DeepBICCN2 for their relevant target classes, using the three different interpretation methods. We then compared the importance score of each nucleotide across interpretation methods (Fig. R18) For ISM, we chose to take the inversed maximal absolute effect over all mutations per nucleotide. We observed that expected integrated gradients (EIG) and integrated gradients (IG) give near-identical results, an interesting observation as IG is 8x faster and can therefore potentially be used for bulk sequence contribution score calculations. The ISM scores also significantly correlates with both EIG and IG, but not as strongly as EIG and IG.

Fig. R18 (fig. S11a): Comparison of per-nucleotide contribution scores for different explanation methods of 171 *in vivo* validated mouse cortical enhancers (Ben-Simon *et al.*, 2025) using the DeepBICCN2 model.

Next, we investigated whether using dinucleotide shuffling instead of random shuffling would result in different explanations using the EIG approach. For this purpose we made use of the *dinucleotide_shuffle* function from tangermeme. In this case we observed a perfect correlation with our default setting (Fig. R19).

Expected integrated gradients vs Expected integrated gradients (dinuc shuffle)

Fig. R19: Comparison of per-nucleotide contribution scores for EIG and EIG using dinucleotide shuffles using the DeepBICCN2 model on a random set of 100 regions from the Zemke *et al.* mouse cortex dataset for random target classes.

To analyze the EIG-ISM differences more in depth, we examined which of the two approaches (gradients vs ISM) is more appropriate for motif recovery. We reran the PBMC Unibind site overlap calculation on the set of 1000 most specific regions per cell type using ISM instead of EIG (as was done in Fig. 3f in the manuscript) (Fig. R20). We found that the performance using ISM drops drastically, further indicating that gradient-based methods seem most appropriate for motif recovery. ISM does have its appropriate use in variant scoring applications (Avsec *et al.*, 2021; Linder *et al.*, 2025).

Fig. R20 (fig. S11b): Recall of found Unibind hits for a set of different TFs in the top 1000 most specific regions per PBMC cell type using ISM.

We therefore added the following to the section “A CRESTed human PBMC model captures validated TFBS”:

“Here, we also compared gradient-based versus ISM nucleotide contribution calculation methods, and observed that gradient-based methods are more appropriate for motif recovery (fig. S11).”

2.3. It will improve the usability of CRESTed to make it easy for users to incorporate their own custom models. The authors are encouraged to provide a simple, flexible API and a tutorial to allow users to integrate custom architectures.

We thank the reviewer for their suggestion. We added a tutorial where users are shown how to make their own architectures, loss functions, and metrics, available at: https://crested.readthedocs.io/en/latest/tutorials/custom_models.html.

2.4. In the enhancer design experiments, the authors validate three designed sequences per condition. Could the authors clarify their criterion or strategy for choosing the three sequences to experimentally validate? Is it based on a quantitative score?

For each task we designed 200 sequences. From these, 5 sequences were sampled at random per task and the nucleotide contributions were visually inspected. Based on this manual inspection we tested 3 out of 5 randomly sampled sequences for each task.

Based on this comment (and the comment from reviewer 1) we added the following paragraph to the following material and methods section:

“Enhancers selection for experimental validation

From the 200 generated sequences for each task (see above), 5 sequences were sampled at random per task and the nucleotide contributions were visually inspected. Based on this manual inspection 3 out of 5 randomly sampled sequences were tested for each task.”

Minor point

m2.1. For enhancer design, could the authors elaborate on how the target prediction scores were chosen?

The following was already mentioned in the results section:

“CREsted designs enhancers specifically active in targeted cell types of a developing zebrafish [...] In this case, we aimed for high levels in the target cell type and no accessibility in other cell types. We set the target predicted level of chromatin accessibility to the average level of predicted chromatin accessibility of the positive validated enhancers.”

And the following in the material and methods section:

“Zebrafish cardiac and body muscle enhancer design

Cell type classes of any time point annotated as either: “Slow muscle cell”, “Slow muscle cells”, “Fast muscle cells”, “Heart”, “Heart field” or “Cardiac muscle” were considered and the average over all positive enhancers (see “CREsted predictions on validated zebrafish enhancers”) of the maximum prediction score over all classes was calculated (avg_pos_prediction_score). To design enhancers specific for cardiac muscle a cost function was defined that minimizes the euclidean distance between the models prediction score on the set of considered cell types and a target vector where classes corresponding to “Cardiac muscle” equals avg_pos_prediction_score and other cell types are zero. Similarly, for body muscle a target vector was used where classes corresponding to either “Slow muscle cell”, “Slow muscle cells” or “Fast muscle cells” equals avg_pos_prediction_score and the other cell types are zero.”

In other words, the prediction score of an already validated set of positive enhancers was used to set the target prediction score.

To make this more clear we now added a reference to Fig. 6b:

“We set the target predicted level of chromatin accessibility to the average level of predicted chromatin accessibility of the positive validated enhancers (Fig. 6b).”

Reviewer #2 (Remarks on code availability):

I can install the package, but I have not attempted to reproduce the analysis.

Reviewer #1

Remarks to the Author:

The authors have addressed our concerns thoroughly and, overall, provide substantial results supporting most of their claims. We particularly appreciate the clearer description of the method's technical details, accompanied by comprehensive benchmarks and comparisons.

We also thank the authors for their detailed discussion of ChromBPNet and gReLU. The explanation of distinct design goals is helpful (ChromBPNet for single-cell-type footprinting, gReLU for large-scale modeling, and CREsted for cross-cell-type peak accessibility prediction). The performance comparison with gReLU on mouse cortex data, together with the clarifications on architectural and loss-function differences, adds valuable context.

We thank the reviewer for their positive comments about our improved descriptions and comparisons to other tools.

That said, the current manuscript does not fully convey this nuanced perspective. The framing risks overstating the foundational novelty of CREsted by not explicitly situating it within the broader ecosystem of related CNN-based frameworks. We strongly recommend incorporating this balanced positioning directly into the manuscript to provide appropriate context.

We believe we already position ourselves sufficiently within the ecosystem, both in the introduction and discussion section we summarize how CREsted is positioned relative to other frameworks and in the result section we provide explicit comparisons. We do not believe that we overstate the foundational novelty of CREsted.

Please find examples below illustrating the positioning of CREsted.

Introduction

Revision 1:

*"[...] some modeling approaches have focused on using **large sequence contexts**. These methods **often predict multiple genomic assays** to achieve a global view of gene regulation, including **bulk or scRNA-seq**. These studies have mostly focused on **predicting the effect of variants on gene expression**, but have **not been used extensively to decode enhancer logic**. Other approaches **have focused on local enhancer contexts**, with the aim of capturing the sequence features that define their functionality at high resolution. Such models have been used to **describe enhancers across tissues and species**, deciphering their inherent codes. Additionally, with the aim of **synthetically designing enhancers**, these **local enhancer models have been the modeling option of choice**"*

Final version:

*"**Sequence-to-function models** take in genomic sequences and predict a variety of genomic assays, including **TF binding, chromatin accessibility, enhancer activity and gene expression**"*

Revision 1:

*"To facilitate and standardize the application of deep learning in regulatory genomics, software packages have recently been introduced with the aim of streamlining the process of data processing, model training and sequence design across different datasets and architectures. Packages such as **Selene** and **EUGENE** offer frameworks for **various predictive tasks and model architectures**. Other methods like **ChromBPNet** and **scPrinter** focus on predicting **TF footprinting** to obtain insights into enhancer codes. Lastly, **gReLU** is a comprehensive toolkit covering many steps in DNA sequence modeling such as **data processing, model training, variant effect prediction, and model-guided sequence design**. While all these frameworks in general share some features, they can be complex to integrate into modern scATAC-seq data analysis pipelines; they **have not been validated on large scale and complex scATAC-seq datasets** in different biological systems; and they often **lack comprehensive global enhancer code analysis tools**."*

Final version:

“Software packages have been introduced with the aim of streamlining the process of data processing, model training and sequence design. For example, **Selene** and **EUGENE** offer frameworks for **various predictive tasks**; **ChromBPNet** and **scPrinter** focus on predicting **TF footprinting** and **Ledidi** is a toolkit to **design edits to biological sequences**. Lastly, **gReLU** covers many steps in sequence modeling such as **data processing, model training, variant effect prediction, and model-guided sequence design**. However, these frameworks **are not tailored to modeling enhancer codes across cell types, nor have they been validated on large scale and complex scATAC-seq atlases in different biological systems and they often lack comprehensive cell type-specific enhancer code analysis tools.**”

Result section: “CREsted provides detailed insights into enhancer codes of mouse cortical cell types”

Revision 1:

“We compared the predictions of our base- and fine-tuned model against a default (6 million parameter) and large (22 million parameter) model trained with the **gReLU** framework. Both CREsted predictions on all test peaks and predictions on cell type-specific test peaks significantly outperformed both the base- and the fine-tuned gReLU models (P -value < 0.001) (Fig. 2d).”

Final version:

“We compared the predictions of our base- and fine-tuned model against a default (6 million parameter) and large (22 million parameter) model trained with the **gReLU** framework. The predictions from CREsted are significantly (P -value < 0.001) more accurate both on all and on cell type-specific test peaks compared to the gReLU models (Fig. 2d).”

Result section: “A CREsted human PBMC model captures validated TFBS”

Revision 1:

“[...] For both enhancers, TFBS were previously identified and experimentally validated. (Fig. 3b). Indeed, CREsted finds the validated TFBS [...] CREsted also proposes additional TFBS, [...] **These additional TFBS were also found for relevant classes of the Borzoi model** (fig. S8a and b). Next, we investigated whether DeepPBMC is also capable of capturing the TFBS of highly complex enhancers, such as the classical Interferon- β (IFNB1) enhanceosome [...] the model retrieves a large part of the enhanceosome’s complexity, [...] **Compared to two classes in Borzoi [...] DeepPBMC identifies substantially more (34%) important nucleotides within this enhanceosome** (fig. S8c).”

Final version:

“Next, we assessed two enhancers for which TFBS were previously experimentally validated [...] DeepPBMC could recover all TFBS [...] (Fig. 3b) but also identifies extra TFBS [...] that **were also confirmed using Borzoi** (fig. S8a and b). Next, we investigated the dendritic cell-specific Interferon- β (IFNB1) enhanceosome. [...] DeepPBMC retrieves a large part of the enhanceosome’s complexity, [...] (Fig. 3c). **Compared to Borzoi, DeepPBMC identifies substantially more (34%) important nucleotides** (fig. S8c).”

Result section: “CREsted identifies high similarity between mesenchymal-like enhancer codes in cancer”

Revision 1:

“**To confirm our model findings, we added an ensemble of ChromBPNet models** [...] The ChromBPNet ensemble illustrated a **similar grouping** of the MES-like states across the GBM and melanoma cell lines (fig. S15a and b), despite its **lower performance on test data** [...]. We also examined the previously validated IRF4 melanocyte-like enhancer, for which the **CREsted-based ISM achieved a higher Spearman correlation (0.75) with in vitro mutagenesis values, compared to ChromBPNet (0.67) and DeepMEL2 (0.66) models** (fig. S15c).”

Final version:

“We confirmed these findings using an ensemble of ChromBPNet models (fig. S15a and b) although this model has a **lower predictive and explanatory performance** (Fig. 4c, fig. S15C).”

Result section: “CREsted-trained models outperform large, pre-trained models on cell type-specific chromatin accessibility predictions”

This entire section positions CREsted within the context of large (“foundation”) models and is present in both the first version of the manuscript as well as the final version.

Discussion

Revision 1:

“CREsted is positioned within the rapidly expanding domain of enhancer modeling tools. One aspect of this domain aims at a complete understanding of gene regulation at a **large scale**, by **predicting bulk or single-cell RNA-seq data and other genomic assays for large input sequences** (hundreds of kilobases long). Models like **Basenji** and **Enformer** have **pioneered** this field, followed by successors like **Borzoï**, **Scooby** and **Decima**. While these models have shown to be **powerful and can provide a global model of gene regulation** across a wide variety of tissues, they may be **overly complex for the task of decoding local enhancer regions**. Furthermore, these large models are **not optimized for enhancer design**. Therefore, another direction entails **local enhancer modeling**, optimized for scrutinization of enhancer logic. Complete software tools for modeling, analyzing and designing enhancers, however, are not as readily available. **gReLU** is to our knowledge the only **comprehensive package** providing all these options. Compared to **gReLU**, **CREsted provides similar features, with additional peak normalization options and extensive enhancer code analyses**. **gReLU** has a larger emphasis on the incorporation and utilization of the previously described **large context models**. We showed that **multi-output task-specific models from CREsted outperform those trained in gReLU** for a mouse motor cortex dataset. Other comprehensive packages, like **tangermeme**, **EUGENE** and **Selene**, either **do not provide training or design options**. Unlike **ChromBPNet** and **scPrinter**, we decided not to pursue **footprinting** analysis, and to predict accessibility aggregated at a region level instead of base-pair level. This alleviated the need for **Tn5 bias correction**, while maintaining **on-par enhancer code learning**. For example, in previous work, we have seen that results from both approaches result in **very similar nucleotide contribution** scores. Moreover, we have extensively shown that **CREsted infers TFBS with strong precision and recall through their contribution scores**, both here and in previous studies. We also validate **CREsted models on large and complex scATAC-seq datasets**, both here and in previous work, **an aspect that is often lacking in the aforementioned enhancer modeling packages.**”

“CREsted models are **multi-output**, while **footprinting models are trained on single scATAC tracks**. The choice for a multi-output model has the benefit that it easily **scales to large scATAC datasets** [...], while **training and evaluating many single-output models in parallel for such a large atlas is impractical**. For our human cancer DeepCCL, we also observed that it **outperformed a combined ChromBPNet model trained on separate cell lines**. This could indicate that using the **contrast of other cell types improves model performance**, as has been observed in a previous study. However, we have seen here and in past studies that **multi-output models tend to misuse positive features**, or motifs, of another class to negatively influence predictions for a given class.”

“[...] we investigated the option of **transfer learning our enhancer models from Borzoï**. We showed that this technique results in **near-identical results as training from scratch and outperforms predictions from the base Borzoï model**. [...] Although **transfer learning from a pre-trained model could have a larger benefit on datasets that are harder to model**, e.g. due to a small amount of informative sequences to train on, this overall indicates that there is limited benefit in transfer learning from large pre-trained models for enhancer modeling and could potentially suggest that both approaches reach a performance plateau dependent on the data quality.”

Final version:

“CREsted is positioned within a rapidly expanding domain of enhancer and gene-locus modeling tools, targeting different aspects of the cis-regulatory code. One major direction focuses on the prediction of **gene expression data** typically using **multi-class models covering hundreds of tissues and cell lines with large input windows** that can integrate information from **multiple CREs**. Such models can be used for inference and variant effect prediction and be fine-tuned to unseen cell types. Packages

such as **gReLU facilitate their usage**. Although these approaches provide global models of gene regulation across diverse tissues, they are **computationally expensive and not suited for decoding enhancers at cell type-specific resolution**, as we showed for mouse cortex cell types, **and enhancer design**. Another direction entails **local enhancer modeling using smaller-scale accessibility-based models**. For example, **ChromBPNet** and **scPrinter**, can be trained to accurately predict accessibility at **basepair resolution** and allow for **TF footprinting analyses** of a **single cell type/line at a time**. Such models reveal important TFBS underlying chromatin accessibility, but **multiple models need to be trained to model enhancers across cell types**. The **CREsted** package is optimized for training **multi-class models encompassing many cell types from scATAC-seq atlases**. Compared to general-purpose packages like gReLU, where similar models can be trained (as we illustrated on mouse cortex), CREsted focuses specifically on modeling of **cell type-specific enhancers** through optimized **data preprocessing** and **track normalization**, training modalities with **fine-tuning on cell type-specific peaks**, and an **improved loss function** to focus on cell type-specific signals. Likewise, it contains **optimized downstream functions for enhancer design and to link TFBS to candidate TFs**. The multiclass paradigm used in CREsted also enables **straightforward scaling to large scATAC-seq datasets** across entire tissues and organisms, as we illustrated here on whole-zebrafish development. A **downside** of multiclass models, however, is that they may **misuse positive sequence features** (e.g., TF activator binding sites) of one class to negatively influence predictions of another, leading to certain motifs being assigned incorrect negative contributions (thus appearing as TF repressor sites), as exemplified in Fig. 2G. Regardless, multi-output models can still identify repressive chromatin factors.”

Additional points:

1, Figure R3: Please extend the analysis to include more cell types to reduce dependence on any single target cell type. Also is it possible to add a training strategy that trains on all peaks (consensus + type-specific) without fine-tuning, and compare its performance to the current approaches?

We apologise that the caption of Figure R3 was not detailed enough, and may have led to confusion.

On the first remark regarding the ‘dependence on any single target cell type’: in fact, the Results shown in Figure R3 do not represent a single cell type, but all 19 cell types in the mouse cortex (model trained on Zemke et al. scATAC-seq atlas). This includes all non-neuronal types, various excitatory and various inhibitory neuron types. Each dot in the figure represents **the average value of the relevant metric across these 19 cell types** for one model with a distinct train/validation/test split. As such, this analysis does not depend on a single cell type. To address this confusion, (1) we modified the figure caption to make this clear, as follows: “(a) Comparison of model performances from the 10 differentially chromosomally split basemodels, fine-tuned models and scratch models. Each dot in the figure represents the average value of the relevant metric across these 19 cell types.”, (2) we included a figure that shows a breakdown of these metrics across these cell types below (**fig. R1b**)

On the second remark about ‘training on all peaks without fine-tuning’: in fact the red boxplot named ‘basemodel’ represents exactly this - training on all peaks without any fine-tuning. **The consensus peaks are ‘all’ peaks, thus also containing the specific peaks**. Here are the explanations on the three training strategies presented in the figure:

- Basemodel: Training on all consensus peaks (non-specific + type-specific) without fine-tuning
- Fine-tuned: Training on all consensus peaks (non-specific + type-specific) followed by fine-tuning on type-specific peaks
- Scratch: Training directly on type-specific peaks, without any prior training (pretraining)

Fig. R1 (fig. S2a): Comparison of model performances from the 10 differentially chromosomally split base-models, fine-tuned models and scratch models. Each dot in the figure represents the average value of the relevant metric across these 19 cell types. Pairwise differences between models were assessed using two-sided Welch’s t-tests on the metrics, and the resulting P values were adjusted for multiple comparisons using the Benjamini–Hochberg false discovery rate procedure; exact adjusted P values for each model–model comparison are reported in the figure. * < 0.05, ** < 0.01, *** < 0.001. (from Rebuttal 1)

Fig. R1b: Per cell type comparison (Pearson Correlation Log) of model performance using three different model strategies over the different chromosomal splits.

2, In Section 1.5, all claims about advantages over classical motif discovery methods should be supported by robust analyses. For example, although the manuscript states that CREsted can discover cell type–specific motifs *de novo*, Fig. R10a shows the lowest sensitivity for CREsted; please reconcile this discrepancy with additional analyses or clearer interpretation. Also, please clarify what is meant by “learning contextual relationships between TFBS and syntactic rules,” and provide a concrete validation plan (e.g., controlled synthetic tests, perturbation analyses, or interpretability diagnostics demonstrating learned higher-order dependencies).

Thank you for bringing this up. In section 1.5 of the Rebuttal, we made the following general claims, in the context of the initial Reviewer’s question to explain “what specific advantages CREsted offer[s] [versus classical motif enrichment analysis]”

- Finding motif instances.
- Finding cell type-specific motifs *de novo*.
- Predicting the effect of non-coding variation.

- Transcription factor binding sites are seen in the context of other binding sites.
- Learning syntactic rules.
- Scoring of unseen sequences/genomes.
- Design of cell type-specific enhancers.

These claims reflect the differences between deep learning based enhancer modeling and motif discovery, in general (not specifically CREsted), and they are supported by a mix of our own analyses that are presented in the manuscript and analyses performed by colleagues in the field. We fully agree with the reviewer's remark that these claims should all be supported by robust analyses and we believe that this is mostly the case already. It was our mistake that, while making these claims in both the rebuttal and especially the manuscript, we did not refer clearly to either our own work or work of others. We have added a supplementary note to the manuscript (note S4) detailing the advantages of deep learning based enhancer modeling over classical motif enrichment analysis. Below we provide a table, that for each claim, indicates whether it is the result from analyses performed in this work and/or by work of others where we refer to relevant literature.

Claim	Supported by analyses in this work?	Supported by analyses by others?
Finding motif instances.	Yes:  • Figure 3 b, e-g • Figure 4 d • Figure 5 e • Figure 6 n 	 • Minnoye and Taskiran et al. 2020. Genome research. • Avsec et al. 2021. Nature Methods. • Bravo Gonzalez-Blas et al. 2024. Nature Cell Biology. • Kosicki et al. 2025. Nature. These studies show that explainability methods on trained CNN-based models can be used to correctly interpret the effect of mutations to functioning enhancers. The mutations that affect enhancer activity significantly correspond to transcription factor binding sites and these are accurately (with high precision and recall) found by CNN-based models.
Finding cell type-specific motifs de novo .	Yes:  • Figure 2 g • Figure 3 d • Figure 4 h-i • Figure 6 o-p • Figure R10 a 	 • Minnoye and Taskiran et al. 2020. Genome research.: Finds TF binding motifs underlying differential chromatin accessibility across melanoma cell states as convolutional filters learned by the CNN-based model. • Avsec et al. 2021. Nature Genetics.: Finds CTCF motifs de novo using TF-MoDISco at topologically associating domain (TAD) boundaries. • Janssens et al. 2022. Nature.: Finds TF binding motifs de novo using TF-MoDISco underlying differential accessibility in Kenyon cells and T-neurons of the fly brain and links those to TF expression. • Sahu et al. 2022. Nature Genetics.: Finds TF binding motifs de novo using TF-MoDISco underlying enhancer activity of random DNA sequences. • Brennan et al. 2023. Developmental cell. Finds TF binding motifs de novo using TF-MoDISco for the fly TFs: Zelda, GAF, Dorsal, Twist, Bicoid and Caudal using a model trained to predict TF binding of those TFs. • Mannens et al. 2024. Nature. Finds TF binding motifs de novo using TF-MoDISco underlying differential accessibility of neuron subtypes in the developing human brain.

		 • Bravo Gonzalez-Blas et al. 2024. Nature Cell Biology. Finds TF binding motifs de novo using TF-MoDISco underlying differential accessibility and activity of enhancers specific for hepatocyte zonation. • Hecker and Kempynck et al. 2025. Science. Finds TF binding motifs de novo using TF-MoDISco underlying differential accessibility of cell types in the mouse and chicken brain. • Linder et al. 2025. Nature Genetics. Finds TF binding motifs de novo using TF-MoDISco underlying differential gene expression in blood and liver cell types. • Nair et al. 2025. bioRxiv. Finds TF binding motifs de novo using TF-MoDISco underlying differential accessibility during induced pluripotent stem cell reprogramming and suggests TF binding site affinity related effects on chromatin accessibility. • Cochran et al. 2025. bioRxiv. Identifies TF binding motifs de novo using TF-MoDISco underlying transcriptional initiation.
Predicting the effect of non-coding variation.	No	 • Avsec et al. 2021. Nature Methods.: Test non-coding SNVs by comparing predicted expression for reference and alternative alleles, validating the effects against eQTL and MPRA data. • Lal et al. 2024, Nature Methods: Find that dsQTLs tend to alter TFBS identified by CNN. • Pampari et al. biorXiv, 2025: Enable robust prediction of causal regulatory variants across caQTLs, bQTLs, and disease-associated loci.
Transcription factor binding sites are seen in the context of other binding sites.	Yes:  • Figure 2 f • Figure 3 b-c, h-i • Figure 4 d • Figure 5 e • Figure 6 n For example, in Fig. 3 h-i we show two example regions: one where the model infers only an EBF1 binding site and another where the model infers binding sites for EBF1 (2x), PAX5 and MEF2. The model correctly predicts that the former region has a lower chromatin accessibility compared to the latter in line with the prediction that the former region contains less binding sites. More importantly, when EBF1 binding sites are mutated in both regions the former region is predicted to be closed while the latter is predicted to reduce in chromatin	 • Kim et al. 2021, Nature Genetics: Assess the effect of heterotypic pairs of transcription factor binding sites and show that most pairs, according to the model, have a multiplicative effect. • Taskiran et al. 2024, Nature: Show that the introduction of ZEB2 binding sites in functional enhancers using a CNN brings down the in vitro activity. Adding Mamo repressor sites to Kenyon Cell enhancers abolishes enhancer activity in vivo. • McAnany et al. 2025, bioRxiv: Here a new tool is developed that allows for the inference of the contribution of each nucleotide to the chromatin accessibility of each nucleotide of a chromatin accessibility peak and it is shown that multiple TF binding sites can affect the chromatin accessibility of a single nucleotide.

	accessibility. Again, in line with the prediction that the latter region contains multiple transcription factor binding sites. These results are confirmed by an EBF1 TF degradation experiment.	
Learning syntactic rules.*	No	 • Avsec et al. 2021. Nature Genetics.: Identifies distance dependent cooperativity of TF binding for the TFs Oct4, Sox2, Nanog and Klf4 in embryonic stem cells. • Kim et al. 2021, Nature Genetics: In silico assessments on how the scrambling of one motif influences binding site functionality of another motif, effect of homotypic motif density, relationship between motif affinity and motif density, the relationship between motif affinity and relative position to the ATAC-seq peak summit. • de Almeida et al. 2022. Nature Genetics: Identifies distance dependent effect of pairs of TF binding sites on enhancer activity based on model predictions. • Pampari et al. 2025, biorXiv: Identify TF-motif pairs that show strict syntactical constraints from CNN (like FOS-TEAD heterodimer). • Seitz et al. 2024, Nature Mal/...;c.hine Intelligence: Quantify epistatic interactions within and between cis-regulatory elements from NN, as well as global explanations of cis-regulatory mechanisms across sequence contexts. • Taskiran et al. 2024, Nature: Identification of preferential spacing between motifs based on model predictions.
Scoring of unseen sequences/genomes.	Yes:  • Figure 2 e 	 • Hecker and Kempynck et al. 2025. Science: Score regions from one species with models trained on other species to highlight conservation of enhancer codes. • Zemke et al. 2023, Nature: Use one species as a test set to evaluate the performance of models trained on other species. • Minnoye et al. 2020, Genome Research: Score different species' genomes with DeepMEL to identify melanocytic enhancers.
Design of cell type-specific enhancers.	Yes  • Figure 6, including in vivo validations 	 • Taskiran et al. 2024, Nature. • de Almeida et al., 2024, Nature • Gosai et al. 2024, Nature • Castillo-Hair et al. 2025, biorXiv All mentioned papers use CNN-based approaches to design enhancers and validate them in vivo or in vitro.

*Extracting and interpreting syntactic rules and assessing to what extent such rules are important for enhancer functioning is still an active field of research. We believe that DNA-sequence based deep learning models are

and will be invaluable for this. As such, currently there are no robust analyses supporting the claim that those models can be used for “learning syntactic rules”, only anecdotal findings. We are happy to either remove this claim from the manuscript or give appropriate context, for example by stating that ‘CNN models are well placed / can be used to discover syntactical rules (cfr. papers cited in table) and that we foresee more comprehensive/better/more formalized discovery of cis-regulatory rules in the future, either by directly extracting them from CNN models (e.g., sparse autoencoders), or by training surrogate biophysical or probabilistic models with the help of trained CNNs, or hybrid models.

To the specific comment: “[...] although the manuscript states that CREsted can discover cell type-specific motifs de novo, Fig. R10a shows the lowest sensitivity for CREsted; please reconcile this discrepancy with additional analyses or clearer interpretation.”

We want to clarify that this qualitative analysis was not meant to assess the sensitivity of motifs identified *de novo* using CREsted (through TF-MoDISco). The analysis clearly shows that *all* motifs extracted from the CREsted model match with known motifs, and that these motifs (from motif databases) are enriched in CREsted-predicted enhancers. This shows that the TF-MoDISco patterns are not false positives or statistical artifacts. These patterns are actually enriched when performing motif discovery on the same set of top-predicted (by CREsted) enhancers. Moreover, in a previous study (Avsec et al., 2021, Nature Genetics), CNN-inferred motif instances from TF-MoDISco pipelines showed improved accuracy of identification over traditional motif discovery methods. They state: “*The advantage of BpNet over classical methods is that it detects motifs and their syntax in a fundamentally different way. Classical methods for motif discovery rely on motifs being over-represented compared to background sequences. Similarly, existing approaches to infer syntax rules use summary statistics of over-represented co-occurrence patterns. These methods have limited statistical power to test individual features present in complex cis-regulatory sequences (Supplementary Note). By contrast, BpNet’s vast network capacity allows it to learn complex predictive rules agnostically based on their ability to accurately predict relevant experimental profiles, without explicitly defining features a priori. This allows the discovery of relatively rare but nonetheless predictive motifs (for example, Oct4–Oct4), as well as predictive syntax features, such as helical periodicity or the direction of TF cooperativity, that were not known to be relevant for these TFs.*”

We realize now, thanks to this comment, that this comparison is also useful to investigate ‘missing’ motifs: certain PWMs that are enriched by pycisTarget or chromVAR, but not identified by TF-MoDISco, may reflect a threshold problem at the level of seqlet calling or pattern clustering (i.e., the lower ‘sensitivity’), which in turn may stimulate the user to adjust these thresholds. For instance, the motifs of cluster 27 which are found by pycisTarget and chromVAR but not the deepPBMC model are similar to C2H2 ZF motifs (**fig. R10a-b**), which are in fact found by deepPBMC, but just not annotated to any TF based on our motif-to-TF annotation database. Similarly, the motifs of cluster 26 are similar to nuclear receptor motifs which are found by deepPBMC, however they reflect a dimer motif in this case (**fig. R10a-b**). Simply adjusting the clustering resolution for the analysis shown in figure R10a would have caused these motifs to co-cluster with motifs also identified by deepPBMC. This leaves the motifs represented by cluster 22 which are found by both pycisTarget and chromVAR but not deepPBMC and the homeodomain motifs that are only found by chromVAR. Notably, chromVAR misses motifs for three TF families (homeodomain; POU, Paired box and CEBP) which are found by both pycisTarget and deepPBMC (**fig. R10a-b**).

We added the following sentence to supplementary note S4: “Sequence-to-function deep learning models learn TF binding motifs *de novo* in order to make biologically relevant predictions. Indeed, all motifs learned by DeepPBMC match to known motifs obtained through motif enrichment analysis (Fig. S6a).”

To the specific comment: “[...] Also, please clarify what is meant by “learning contextual relationships between TFBS and syntactic rules,” and provide a concrete validation plan”

With contextual relationships we mean the following: CREsted evaluates DNA sequences of, by default 2114 bp, at a time. Functional enhancers often consist of multiple transcription factor binding sites and CREsted is able to infer the combined effect of all binding sites together on cell type-specific chromatin accessibility. In other words, the context of all transcription factor binding sites is used to make a prediction. Using classical motif scanning approaches alone, this is not the case. Here, the score of each motif is an independent feature and additional modelling is needed to combine the effect of multiple features.

With syntactic rules we mean the following: relative spacing and orientation of transcription factor binding sites and their copy number and affinity might be important for cell type-specific enhancer activity and the CREsted

models might implicitly learn such rules. In the field it is currently believed that such syntactic features are important but that the rules are not strict (i.e., soft syntax (Zeitlinger 2021 Current Opinion in Systems Biology)). The extent to which such rules are important for different cell types, whether they are properly modelled by DNA sequence based deep learning models and, if so, how to extract them from the model are currently still open research questions and fully addressing this point is beyond the scope of this manuscript. We clarified these points in a new supplementary note added to the manuscript (note S4).

Reviewer #2

Remarks to the Author:

I thank the authors for the revision which have sufficiently addressed my prior comments.

We thank the reviewer for their positive feedback.